# The Brazilian Earth System Model version 2.5: Evaluation of its CMIP5 Historical Simulation

**Sandro F. Veiga[1], Paulo Nobre[2], Emanuel Giarolla[3], Vinicius Capistrano[4], Manoel Baptista Jr.[2], André L. Marquez[2], Silvio Nilo Figueroa[2], José Paulo Bonatti[2], Paulo Kubota[2], Carlos A. Nobre[5]**

[1]Earth System Science Center-CCST, National Institute for Space Research (INPE), São José dos Campos 12227-010, São Paulo, Brazil

[2]Center for Weather Forecasting and Climate Studies-CPTEC, National Institute for Space Research (INPE), Cachoeira Paulista 12630-000, São Paulo, Brazil

[3]Center for Weather Forecasting and Climate Studies-CPTEC, National Institute for Space Research (INPE), São José dos Campos 12227-010, São Paulo, Brazil

[4]Amazonas State University (UEA), Manaus 69005-010, Amazonas, Brazil

[5]CN Research, São José dos Campos 12544-590, São Paulo, Brazil

*Correspondence to*: Sandro F. Veiga (sandro.veiga@inpe.br)

**Abstract**

The performance of the coupled ocean-atmosphere component of the Brazilian Earth System Model version 2.5 (BESM-OA2.5) was evaluated in simulating the historical period 1850−2005. After a climate model validation procedure in which the atmospheric and oceanic main variabilities were evaluated against observed and Reanalysis datasets, the evaluation specifically focused on the mean climate state and the most important large-scale climate variability patterns simulated in the historical run, which was forced by the observed greenhouse gas concentration. The most significant upgrades in the model's components are also briefly presented here. BESM-OA2.5 could reproduce the most important large-scale variabilities, particularly over the Atlantic Ocean (e.g., the North Atlantic Oscillation, the Atlantic Meridional Mode and the Atlantic Meridional Overturning Circulation), and the extratropical modes that occur in both hemispheres. The model's ability to simulate such large-scale variabilities supports its usefulness for seasonal climate prediction and in climate change studies.

# 1. Introduction

Climate models, which have recently been expanded into Earth system models via inclusion of biogeochemical cycles, are key tools for investigating climate phenomena that significantly influence human societies (e.g., von Storch, 2010; Flato, 2011). Since 2008, the Brazilian climate community has been engaged in setting up the Brazilian Earth System Model (BESM; Nobre et al., 2013; Giarolla et al., 2015). This major scientific task has been carried out by Brazilian scientific institutions and highlights the critical need to produce reliable future climate projections and to understand their potential impact, particularly over South America. The primary objective of this effort was to assemble the scientific expertise capable of developing and maintaining a state-of-the-art Earth system model. Such an achievement would represent a significant step forward in establishing a scientific tool that can be used in different types of research activities. The importance of such an undertaking lies in the need to understand the physics of the Earth system to produce and lend credibility to studies that explore the impacts of climate change on different areas of great importance, such as food and water security, tropical ecosystems, natural disasters. One of the fundamental aims of the BESM project is to participate in the Coupled Model Intercomparison Project's sixth phase (CMIP6; Meehl et al., 2014).

BESM has been set up at the Brazilian National Institute for Space Research (INPE). Currently, it consists of a land-ocean-atmosphere coupled model in which the coupling is achieved via the Flexible Modeling System (FMS) coupler, a tool developed at the Geophysical Fluid Dynamics Laboratory (GFDL) of the National Oceanic and Atmospheric Administration (NOAA). The inclusion of aerosols (as read-in fields) and

atmospheric chemistry components are in the implementation and testing phases. Currently, work has been completed to activate the biogeochemical model (TOPAZ) within the MOM5 to simulate biogeochemical cycles in future simulations.

The previous version of BESM (BESM-OA2.3) was first evaluated by Nobre et al. (2013). This version showed a significant bias against precipitation in the tropical region, as it showed a deficient representation of the precipitation in the Amazon region. To improve these aspects, studies were conducted to ameliorate cloud parameterizations over the tropics, and the resulting changes improved the representation of the precipitation over the same region and the representation of Convergence Zones over both the Atlantic and Pacific Ocean basins (Bottino and Nobre, 2018). The main changes in the current version of BESM relate to its atmospheric model, which now incorporates modifications in the surface wind field and its parameterizations as described in Capistrano et al. (2018). The updated version presented in this manuscript is BESM-OA2.5.

Operationally, BESM-OA2.3 is already being used for extended weather forecasting (10−30 days) and for seasonal climate prediction (three months), as well as for producing global climate change scenarios (Nobre et al., 2013) and providing atmospheric and oceanic boundary conditions to regional climate models for dynamical downscaling of climate change scenarios (Chou et al., 2014).

This overview paper describes the most important developments and improvements in the model's components, and presents the simulation of recent-past mean climate conditions and major large-scale climate phenomena. In section 2, the BESM-OA2.5 components and experimental design are briefly described; section 3

presents the methodology and the observed data used to evaluate the model; section 4

presents the evaluation of the historical simulation, which evaluated the most important

atmospheric and oceanic variables related to their climatological fields and the

prominent large-scale phenomena of the climate system; and, finally, section 5 provides

a summary.

## 2 Model description and simulation experimental design

### 2.1 BESM-OA2.5

The atmospheric component of BESM-OA2.5 is the Brazilian Global

Atmospheric Model (BAM; Figueroa et al., 2016), which was developed at the Center

for Weather Forecasting and Climate Studies (CPTEC/INPE). The BAM is a primitive

equation model with spectral representation with triangular truncation at the wave

number 62 (corresponding to a grid resolution of approximately $1.875\,° \times 1.875\,°$) and

28 sigma levels in the vertical, with uneven increments between the levels, i.e., T62L28

resolution. As mentioned before, it is in the atmospheric component that underlies the

primary differences between BESM-OA2.5 and BESM-OA2.3 (Nobre et al., 2013). The

new version includes a key improvement in the energy balance at the top of the

atmosphere via reduction of the mean global bias from $-20$ W m$^{-2}$ in version BESM-

OA2.3 to $-4$ W m$^{-2}$ in the current version (Capistrano et al., 2018). BESM version 2.5

incorporates the formulation presented in Jiménez et al. (2012) for representing the

wind, humidity and temperature in the surface layer. The model runs without flux

correction or adjustment. The physics parameterizations for the continental processes

are based on the Simplified Simple Biosphere Model (SSiB) land surface model (Xue et

al., 1991) the shortwave radiation is based on the Clirad scheme (Chou and Suarez,
1999; Tarasova and Fomin, 2000), the longwave radiation is based on Harshvardhan
scheme (Harshvardhan et al., 1987), the Cloud microphysics uses Ferrier scheme
(Ferrier et al., 2002), the vertical diffusion is the modified MY2.0 scheme (Mellor and
Yamada, 1982), the gravity wave drag scheme is based on Webster et al. (2003) , the
deep convection module is based on Grell and Dévényi (2002), and the shallow
convection module is based on Tiedtke (1983). More details can be found in Figueroa et
al. (2016) and Capistrano et al. (2018).

9        The oceanic component of BESM-OA2.5 is the Modular Ocean Model version

4p1 (MOM4p1; Griffies, 2009) developed at GFDL, which includes the Sea Ice
Simulator (SIS) built-in ice model (Winton, 2000). There were no changes in the
physics parameterizations used in BESM-OA2.3. The horizontal grid resolution in the
zonal direction is $1°$, and in the meridional direction it varies uniformly from $1/4°$
between $10°S$ and $10°N$ to $1°$ of resolution at $45°$ and to $2°$ of resolution at $90°$ (in
both hemispheres). The vertical resolution has 50 levels, with approximately 10 m
resolution in the upper 220 m and increasing gradually to about 370 m resolution at
deeper levels. The oceanic model spin-up was performed in a manner similar to that of
Nobre et al. (2013) and Giarolla et al. (2015), in which the spin-up run begins from rest,
and the ocean T-S structure is that of Levitus (1982). The initial stage of the ocean
model spin-up was performed over a 13-year period, forced by climatological
atmospheric fields (winds, solar radiation, air temperature and humidity, and
precipitation). The model spin-up was then integrated by an additional 58-year period,
forced by interannually varying atmospheric fields from Large and Yeager (2009),

while maintaining the river discharges and the sea ice variables at their respective monthly mean climatological values. The forced ocean model run was used to save the oceanic dynamical and thermodynamical structures to be used for initiating future coupled model experiments.

The atmospheric and oceanic models were coupled via the Flexible Modeling System (FMS) coupler, which was also developed at GFDL and incorporated into MOM4p1. The atmospheric model receives SST and ocean albedo data from the ocean and sea ice models at hourly time increments. The oceanic model receives information about freshwater (liquid and solid precipitation), momentum fluxes (winds at 10 m), specific humidity, heat, vertical diffusion of velocity components and surface pressure, also at hourly time increments. The wind stress fields were computed in MOM4p1 using the Monin-Obukhov scheme (Obukhov, 1971). In the coupled simulations, the ocean temperature and salinity restoration options were set to off.

**2.2 Experimental design**

A set of numerical experiments were performed with the coupled ocean-atmosphere version of BESM-OA2.5 following the CMIP5 experimental design protocol (Taylor et al., 2012), and they are shown schematically in Figure 1. Out of the experiments listed below, only the historical simulation is evaluated in this paper. The following experiments were performed:

- Historical: the simulation ran over the period 1850−2005 (156 years), forced by the observed historical atmospheric equivalent $CO_2$ concentration (greenhouse gas only) over this period, based on the CMIP5 protocol.

- piControl: ran for 1140 years, forced by the invariant pre-industrial atmospheric $CO_2$ concentration level (280 ppmv).

- Abrupt 4×CO2: ran for 1000 years, consisting of an abrupt quadruplication of the atmospheric $CO_2$ concentration level from the piControl simulation.

- RCP4.5: ran over the period 2006−2105 (100 years), forced by the time-dependent changes in greenhouse gas levels projected by the Representative Concentration Pathways 4.5 (RCP4.5), based on the CMIP5 protocol. This simulation continued the historical simulation through the 21[th] century, reaching a radiative atmospheric forcing of 4.5 W m$^{-2}$ in 2100.

- RCP8.5: the same as the RCP4.5 simulation, but forced by the time-dependent changes in greenhouse gas levels projected by the Representative Concentration Pathways 8.5 (RCP8.5), based on the CMIP5 protocol, i.e., reaching a radiative atmospheric forcing of 8.5 W m$^{-2}$ in 2100.

The ocean stand-alone ran for 71 years (a 13-year period of ocean model spin-up forced by climatological atmospheric fields plus a 58-year period forced by interannually varying atmospheric fields). Next, a spin-up of the fully coupled model was performed for 100 years. The oceanic and atmospheric states at the end of this 100-year-long integration were used as the initial conditions for the piControl simulation. The versions of the model differ slightly in the 100-year spin-up and the piControl run, in the parameterizations of the land ice albedo and in the cloud microphysics. For its initial conditions, the historical simulation used information about the 14[th] year provided by the piControl simulation. The piControl simulation showed stable conditions following a fast adjustment over the first 13 years of simulation (figure not shown). Therefore, it is assumed that the historical simulation had a spin-up of 113

years. The ocean stand-alone runs for 71 years (13 years period of ocean model spin-up
forced by climatological atmospheric fields plus 58 years period forced by interannually
varying atmospheric fields). Then a spin-up of the fully coupled model is done for 100
years. The ocean and atmosphere states at the end of this 100 years long integration are
used as the initial condition for the piControl simulation. The piControl simulation
shows stable conditions after a fast adjustment over the first 13 years of simulation
(figure not shown). The analyses of the piControl and $4\times CO2$ simulations are described
in Capistrano el al. (2018) and Nobre et al. (2018, in preparation). Capistrano et al.
(2018) estimated that BESM-OA2.5 has an equilibrium climate sensitivity of 2.96 ℃ in
the abrupt $4\times CO2$ experiment. This value is within the range of 2.07 to 4.74 ℃ that has
been computed for 25 CMIP5 models and is close to the ensemble averaged value
(3.30 ℃).
## 3. Methods and data
To evaluate the output of the BESM-OA2.5 historical simulation, comparisons
were made against the observed datasets and reanalysis products. The atmospheric
fields and sea ice concentration were from the Twentieth-Century Reanalysis dataset
version 2 (20CRv2; Compo et al., 2011) with a global horizontal resolution of $2\degree \times 2\degree$
and                    24                    vertical                    levels
(https://www.esrl.noaa.gov/psd/data/gridded/data.20thC_ReanV2.html);              the
precipitation dataset was obtained from the Global Precipitation Climatology Project
version 2.2 Combined Precipitation Dataset (GPCP; Adler et al., 2003; Huffman et al.,
2009)    with    a    global    horizontal    resolution    of    $2.5\degree$    $\times$    $2.5\degree$
(http://rda.ucar.edu/datasets/ds728.2/#!description) and from the CPC Merged Analysis

of Precipitation (CMAP; Xie and Arkin, 1997), with a global horizontal resolution of 2.5 ° × 2.5 ° (https://www.esrl.noaa.gov/psd/data/gridded/data.cmap.html). To compare the global average air surface temperature, the Hadley Centre-Climate Research Unit Temperature Anomalies version 4 (HadCRUT4, Morice et al., 2012), which provides a time series of the globally averaged air temperature anomaly at 2 meters, (https://crudata.uea.ac.uk/cru/data/temperature/) was used. The cloud cover was compared to data from The International Satellite Cloud Climatology Project (ISCCP D2; Rossow and Schiffer, 1999), which has a global horizontal resolution of 2.5 ° × 2.5 ° (https://isccp.giss.nasa.gov/products/onlineData.html). Finally, for the sea surface temperature (SST) comparisons, the Extended Reconstructed Sea Surface Temperature version 4 (ERSSTv4, Huang et al., 2015), which is available at a grid resolution of 2 ° × 2 °, was used (https://www.esrl.noaa.gov/psd/data/gridded/data.noaa.ersst.v4.html).

To identify the main modes of climate variability, all of the analyses presented in the paper were performed using detrended data set anomalies. Detrended data sets were obtained by removing the linear trend based on a least-squares regression. For the analyses using monthly data sets, the annual cycle was removed by subtracting the climatological monthly means from the respective individual months. Prior to performing the analyses, the model's data sets were interpolated to the grid resolution of the respective observation or the reanalysis data sets used for comparison.

The Empirical Orthogonal Function analysis (EOF; Hannachi et al., 2007) was used to analyze the model's ability to simulate major modes of climate variability and to compare them with observations. Prior to performing the EOF calculations, the data were weighted by the square root of the cosine of latitude. The results of the EOF maps

are shown as the original data anomalies regressed onto the normalized principal component (PC) time series, i.e., by the standard deviation.

In this paper, to evaluate the periodicity of the phenomena, the power spectrum technique based on Fourier analysis of the normalized time series was applied, in which the normalization was based on the long-term monthly standard deviation.

To gain better insight into the performance of BESM-OA2.5 in relation to the global average near-surface air temperature and the average SST in the equatorial regions of the Pacific and Atlantic Ocean, a comparison with 11 CMIP5 models was performed (Table 1). Because the BESM-OA2.5 historical simulation is forced only by the observed $CO_2$ equivalent concentration, for this comparison the historical simulation forced only by greenhouse gas (historical GHG) was chosen.

## 4. Results

### 4.1 Mean climate state

In this section, the most important atmospheric and oceanic variables are evaluated in relation to their climatological fields, either globally or over regions in which their representation are key elements of the climate system.

### 4.1.1 Mean surface air temperature

The evolution of the global surface air temperature during the industrial era is a key element for analyzing the long-term model behavior while being forced by the observed conditions. The HadCRUT4 observation and BESM-OA2.5 time series of the globally averaged air temperature anomaly at 2 meters are shown in Figure 2. The time

series are the annual mean anomalies relative to the period from 1850–1879. The
BESM-OA2.5 simulation of the global average surface air temperature evolution
closely followed the observed time series. However, since BESM-OA2.5 does not
incorporate the representation of aerosols, and consequently their cooling effects, the
surface air warming rate should be higher, similar to the remaining models (the grey
shadow in Figure 2). To compare BESM-OA2.5 with the selected CMIP5 models, the
grey shadow represents the spread of the minimum and the maximum values of the
yearly anomalies from the 11 models (Table 1). In this comparison, the historical GHG
simulation was used, in which the models are only forced by well-mixed greenhouse
gases (mainly carbon dioxide, methane, and nitrous oxide), without the cooling
resulting from the direct and indirect effects of aerosols, volcanos and effects of land
use change. Thus, the CMIP5 models show a warmer tendency compared with the
observations (see Jones et al., 2013 for more details). Although BESM-OA2.5 has the
same forcing conditions, it does not show the warming tendency seen in the remaining
models. With exception of GFDL-ESM2M (1861−2005) and HadGEM2-ES
(1860−2005), all of the remaining CMIP5 models encompass the period from
1850−2005, and their respective anomalies are from the period 1850−1879. For GFDL-
ESM2M and HadGEM2-ES, the anomalies are computed relative to the periods
1861−1890 and 1860−1889, respectively.
The net radiation at the top of atmosphere (TOA) has a negative bias and the net
ocean/atmosphere heat flux has a positive bias (Fig. 3). The net TOA radiation has a
mean value of $-4.20$ W m$^{-2}$, and the net ocean/atmosphere heat flux has a mean value
of 1.16 W m$^{-2}$ over the first 50 years. The net radiation imbalance at the TOA is related
to a significant loss of energy at the TOA both from outgoing long-wave radiation and
outgoing short-wave radiation. Throughout the simulation, the net radiation at the TOA
becomes less negative due to the increasing $CO_2$ in the atmosphere and the
consequential increase in the atmospheric heat content. Part of this heat is transferred
into the ocean, as indicated by the positive net increase in the ocean/atmosphere heat
flux. The negative net radiation at the TOA and the positive ocean/atmosphere heat flux
are likely the reasons for the weak warming observed in the historical simulation (Fig.
2), as the atmosphere loses heat to outer space and into the oceans during the
simulation.
**4.1.2 Mean precipitation**

11       One of the key points in evaluating a climate model is to gauge its ability to

simulate the hydrological cycle, as this cycle is critical for maintaining the energy
balance of the climate system. Figure 4 shows the spatial distribution of annual mean
precipitation for (a) BESM-OA2.5, (b) the GPCP dataset, the spatial distribution of
annual mean precipitation bias (c) for BESM-OA2.5 relative to the GPCP dataset, and
(d) for BESM-OA2.5 relative to the CMAP dataset. The spatial annual mean
precipitation values represent averaged values over the periods 1971–2000 and 1979–
2008 for BESM-OA2.5, and the GPCP and CMAP datasets, respectively. The global
model's mean biases are similar for GPCP (0.3 mm day$^{-1}$) and CMAP (0.4 mm day$^{-1}$).
In the case of the global model's root-mean-square-error (RMSE) biases, they are also
similar for GPCP (1.4 mm day$^{-1}$) and CMAP (1.5 mm day$^{-1}$). BESM-OA2.5 was able
to reproduce the globally observed precipitation patterns and showed a slight
improvement in the global mean precipitation simulation over the previous version
(BESM-OA2.3). The spatial average biases were 0.3 mm day$^{-1}$ and 0.5 mm day$^{-1}$, and
the RMSE biases were 1.4 mm day$^{-1}$ and 1.7 mm day$^{-1}$ for BESM-OA2.5 and BESM-
OA2.3, respectively. The improvements are particularly clear in the Pacific and Atlantic
Ocean areas, where BESM-OA2.5 reduced the positive bias that extends into
subtropical southeast Pacific and into both north and south Atlantic subtropics that was
observed in BESM-OA2.3 (see Fig. 6a of Nobre et al., 2013). Despite these
improvements, BESM-OA2.5 still generated a strong negative bias over the Amazon
region. This is a particular concern since an important aim is related to the model's
usefulness for future climate projections in that region. Based on the progress observed
from BESM-OA2.3 to BESM-OA2.5, work on cloud parametrizations to improve the
precipitation simulation over the Amazon is still needed. Nevertheless, some state-of-
the-art models show deficiencies in generating precipitation over the Amazon region.
This is the case of the IITM-ESM (Fig. 5; Swapna et al., 2018), although the bias is
more confined to the north of the Amazon, and for the NESM, which has a more
distributed bias over the region (Fig. 9; Cao et al., 2018). The Indian subcontinent
region also has a significant negative bias, and a strong positive bias appears over the
Indian Ocean and in the South Pacific Convergence Zone (SPCZ). Such strong positive
biases over the Indian Ocean (near the African coast) are also identified in different
versions of the CCSM model (Fig. 5; Gent et al., 2011).

20          To understand the global atmospheric circulation associated with the

precipitation deficiencies over both the Amazon and Indian regions, the global
anomalies of the velocity potential and the divergence of the wind at the 200 hPa
pressure level were computed and are shown in Figure 5. The velocity potential and

divergent wind anomalies were averaged over the period 1971−2000 for the BESM-

OA2.5 outputs (Fig. 5a), the Reanalysis (Fig. 5b) and for the difference BESM-OA2.5

minus Reanalysis (Fig. 5c, 5d and 5e). Figure 5c shows anomalous convergence over

the Amazonian and Indian regionsresulting on the model's poor capacity in creating

convection and, consequently, to generate precipitation. Figures 5d and 5e show the

velocity potential and wind divergence separated by season. For the Amazonian rainfall

season, which occurs during MAM, it is possible to observe anomalous convergence at

the high levels of the atmosphere (Fig 5d). An equivalent result was observed for the

Indian region during the JJA season (Fig. 5e).

Figure 6 shows the zonally averaged precipitation during the four seasons. For

this comparison, the results of the BESM-OA2.3 analysis performed by Nobre et al.

(2013) are also shown. Both versions could reproduce the maximum peaks of

precipitation in both the tropical and subtropical regions. BESM-OA2.5 showed a

negative bias from around $40\,°$ latitude poleward in both hemispheres. In the seasons

DJF, JJA and SON, BESM-OA2.5 had a positive bias on the peak of maximum

precipitation corresponding to the ITCZ. During the MAM season, the model still failed

to perform the interhemispheric transition of the ITCZ. However, the JJA season

showed that BESM-OA2.5 could completely perform the transition, whilst BESM-

OA2.3 showed a double ITCZ in the JJA and SON seasons. The double ITCZ problem

is one of the most significant biases that persists in climate models (e.g., Hwang and

Frierson, 2013; Li and Xie, 2014; Tian, 2015). With the exception of the MAM season,

BESM-OA2.5 yielded zonal precipitation values that were identical to the observed

values, although with a generally positive bias. It should be noted that BESM-OA2.5

has a rapid precipitation decline at high latitudes. Compared to the GPCP dataset, the
model showed peaks of precipitation at the mid-latitudes, which are related to the storm
tracks, and less precipitation in the subtropics.

4          Figure 7 shows the general characteristics of cloudiness over the globe simulated

by the model. In particular, Figure 7a shows that the model underestimated cloudiness
in most parts of the globe, with significant exceptions in the high latitudes of the boreal
hemisphere and in the southern subequatorial regions of the Pacific and Atlantic Oceans
upon comparison with observations. Globally, BESM-OA2.5 has a cloudiness negative
bias of −13.9 % with a RMSE of 19.9 %. The periods used were 1971−2000 and
1984−2009 for BESM-OA2.5 and ISCCP, respectively. The model failed to generate
clouds in the high latitudes of the austral hemisphere, as can be observed in Figure 7b,
where the percentage of cloud cover is negligible. The reason for this lack of simulated
cloudiness in this region is not yet clear. However, Figure 7b shows that it is possible to
see that the meridional variation in cloud cover simulated by the model is similar to the
observation.
**4.1.3 Zonal atmospheric mean state**

17          Figures 8 and 9 present the analysis of the zonally averaged vertical profiles of

air temperature and zonal wind for all seasons as simulated by BESM-OA2.5 and the
respective bias relative to the 20CRv2 Reanalysis dataset, in which all of the data are
time averaged over the period 1971−2000. BESM-OA2.5 had a large positive air
temperature bias that appears above the 250 hPa pressure level (Fig. 8) in the subpolar
and polar regions during all of the seasons. This result indicates that the model warms
abnormally in the tropopause and the lower stratosphere in the polar regions. The warm
bias is stronger during the DJF and MAM seasons over the northern polar region,
reaching a maximum bias of 20 ℃ during the DJF season. Such a bias is a matter of
concern since other models, despite showing strong bias in the polar regions, do not
show such a strong bias. BNU-ESM presents positive biases up to 10 ℃ in the austral
hemisphere during the season JJA (Fig. 3a; Ji et al., 2014) and NorESM1-M presents
negative biases (∼−10 ℃) during the DJF and JJA seasons (Fig. 9; Bentsen et al., 2013).
In the lower and middle troposphere, BESM-OA2.5 showed a negative temperature bias
that is stronger in the lower troposphere over the polar region in the respective winter-
spring seasons in both hemispheres, i.e., during DJF and MAM over the North Pole, and
JJA and SON over the South Pole. This negative bias reached its maximum of −10 ℃
over the South Pole during SON. Such a negative bias in the troposphere has already
been reported by many CMIP5 models (see Charlton-Perez et al., 2013; Tian et al.,

13    2013).

14        Concerning the zonal wind, BESM-OA2.5 simulated a much weaker wind speed

in the tropopause and stratosphere over the boreal hemisphere, mainly during the DJF
season, which has a maximum negative bias of $-26$ m s$^{-1}$ at 50−30 hPa (Fig. 9a). This
bias is out of the range ($-10$ m s$^{-1}$ to 10 m s$^{-1}$) presented by some other models,
including NorESM1-M (Fig. 10; Bentsen et al., 2013) and NESMv3 (Fig. 10d; Cao et
al., 2018). The tropospheric jets and their seasonal migration were reasonably well
simulated, although the eastward wind was stronger in the subtropics, with a maximum
positive bias of 12 m s$^{-1}$ at 300−100 hPa during the MAM season.
**4.1.4 Ocean mean state**

23        The global distribution and the range values of the sea surface temperature

(SST) are important characteristics of the mean climate state. Figure 10 shows a spatial
map of the annual mean SST values for (a) BESM-OA2.5 and (b) ERSSTv4 as well as
the (c) the bias for BESM-OA2.5 relative to the ERSSTv4 dataset. BESM-OA2.5
showed a warm SST bias that spread throughout all of the oceans, in contrast with the
negative biases shown by most of the CMIP5 models over the North Pacific and North
Atlantic Oceans (see Wang et al., 2014). However, the extreme values found in the
south of Greenland and in the North Pacific, where they reached ~6 ℃, are well within
the range of the biases reported by other models, including NorESM1-M (Fig. 12b;
Bentsen et al., 2013) and IITM-ESM (Fig. 3; Swapna et al., 2018). Such warm biases do
not appear in the tropical and subtropical regions in the BESM-OA2.3 simulation (Fig.
5a of Nobre et al., 2013), where there are instead cold SST biases. The spatial average
biases are 1.5 ℃ and 0.9 ℃, and the RMSEs are 1.9 ℃ and 2.1 ℃ for BESM-OA2.5
and BESM-OA2.3, respectively. A notable feature of BESM-OA2.5 is its strong warm
SST bias in the North Pacific and off the California coast and south of Greenland. The
model still overestimated the SSTs in the major eastern coastal upwelling regions. This
feature is a systematic error observed in different state-of-the-art models that could be
caused by the simulation of weaker-than-observed alongshore winds, which
consequently leads to an underrepresentation of the upwelling and alongshore currents
(e.g., Humboldt, California and Benguela Currents), and/or the under-predicted effects
of shortwave radiation due to deficient simulation of stratocumulus clouds over cold
waters (see Richter, 2015). Nevertheless, the bias was negligible over the north
equatorial Pacific and in large parts of tropical western Atlantic.
Figure 11a shows the mean SSTs in the equatorial Pacific for BESM-OA2.5 and
ERSSTv4, averaged over the period 1971−2000. The equatorial region is defined as the

region lying between the latitudes 2 °S and 2 °N. The model simulated a warmer mean SST over the western and extreme eastern parts of the equatorial Pacific Ocean. This positive bias was most notable in the western part of the Pacific, where it was about 1.5−2 ℃ warmer than the observed values and was warmer than the values from the CMIP5 models (shown by the shaded grey area in Figure 11a). However, for the extreme eastern part of the basin, the model showed a lower bias compared with those of the CMIP5 models. For most of the central Pacific Ocean, BESM-OA2.5 yielded a very good representation of the SSTs, with a RMSE of 0.14 ℃ between 160 ℉ and 120 ℉. The annual cycle of the equatorial Pacific SST anomalies for BESM-OA2.5 and ERSSTv4 are shown in figure 11b and 11c, respectively. BESM-OA2.5 simulated the marked annual cycle that occurs on the eastern Pacific reasonably well, although the negative SST anomalies between July and December are up to 1 ℃ colder than the observed values. The propagation of the SST anomaly patterns from the eastern to the western parts of the Pacific Ocean that occurs throughout the year was not well captured by the model. BESM-OA2.5 showed an annual cycle in the western part of the Pacific Ocean, where the observations show a semiannual pattern of SST anomalies. The same methodology was used for the tropical Atlantic. Figure 12a shows that in the Atlantic basin there was a significant bias of about 3 ℃ in the eastern part of the basin. This bias started in the central Atlantic and was higher than the biases of the CMIP5 models (shown by the shaded grey area in Figure 12a). However, it should be noted that the CMIP5 models also have a warm bias in the eastern part of the tropical Atlantic, which is a problem discussed in previous studies (e.g., Richter et al., 2014 and references therein). Despite this warm bias, the tropical Atlantic seasonal SST variation was well simulated by BESM-OA2.5, in particular on the eastern side of the basin, as can be seen

in Figures 12b and 12c.

2         To evaluate how the global ocean profile evolves throughout the simulation,

depth-time Hovmöller diagrams of global mean ocean salinity and temperature
departures from their respective initial conditions were calculated (Fig. 13a and 13b) in
the historical simulation. Here, "initial condition" indicates the value of the first year of
the simulation, in this case, 1850. The ocean salinity slightly increased below a depth of
1000 m and from 1935 on, the increase reached 0.04 PSU between depths of 1500 and
3000 m compared with the initial values (Fig. 13a). Above a depth of 1000 m, there was
a significant freshening of the ocean waters, with the surface water salinity decreasing
up to 0.18 PSU by the end of the simulation. Concerning ocean temperature, prominent
warming occurred from the surface up to a depth of 400 m (Fig. 13b). This warming
was more significant at the end of the simulation (~0.6 ℃ compared with the initial
conditions) and was mostly caused by the ocean warming drift in the model. Fig. 13c
shows the same diagram for a piControl simulation (during the period in which both
simulations were performed in parallel), which also shows the ocean drift feature.
However, the ocean temperature anomalies above 600 m reach approximately 0.6 ℃ in
the historical simulation, whereas they only reached approximately 0.4 ℃ in the
piControl. This difference of 0.2 ℃ between the two simulations is likely due to the
global warming of the planet and consequential increasing heat flux from the
atmosphere into the ocean (Fig. 13d). In deeper waters, from 1500 m down to the ocean
floor, there was weaker warming, indicating that the ocean is gaining heat mainly in its
upper layers (Fig. 13b). Between the depths of 500−1500 m, a cooling tendency was
observed relative to the initial conditions. Such a tendency could indicate that the ocean
is still drifting from its initial conditions in the historical simulation.

The meridional overturning circulation (MOC) plays an important role in transporting heat from the tropics to the higher latitudes in both hemispheres. This is particularly important in the North Atlantic, where the Atlantic Meridional Overturning Circulation (AMOC) has a profound impact on the climate of the surrounding continents (see Buckley and Marshall, 2015). The AMOC in the BESM-OA2.5 historical experiment showed the typical structure described in Lumpkin and Speer (2007), with the upper layer of the upper cell, which is the northward flux, depicted at the appropriate depth, from the surface down to ~1000 m (Fig. 14a). However, the upper cell simulated by BESM-OA2.5 was too shallow compared with the RAPID measurements (McCarthy et al., 2015). The depth of the upper cell was 2500 m in the model, whereas the measurements show its depth at ~4500 m. This shallow upper cell of the AMOC is a common feature of state-of-the-art climate models (see Menary et al., 2018). In the deep ocean, the model accurately simulated the Antarctic Bottom Water flowing northwards over the Atlantic Ocean floor. The annual mean maximum AMOC strength simulated by BESM-OA2.5 is about 15 Sv (1 Sv $\equiv 10^6$ m$^3$ s$^{-1}$) between 25 °N and 30 °N at a depth of about 850 m (Fig. 14a). This maximum value is within the 17.2 $\pm$ 4.6 Sv mean strength (with a 10-day filtered root-mean-square variability of 4.6 Sv) observed at 26.5 °N by the RAPID project (McCarthy et al., 2015). This value is also in the range of the maximum volume transport strength simulated by other state-of-the-art CMIP5 models (Weaver et al., 2012; Cheng et al., 2013). Figure 14b shows the maximum annual mean AMOC strength time series for the historical period at the 30 ° N. For comparison, Figure 14c shows the AMOC maximum volume transport strength measured by the RAPID project over the period April/2004 to October/2015 (http://www.rapid.ac.uk/rapidmoc/rapid_data/datadl.php).

1        After averaging the maximum AMOC strength over the first and the last 30

years of the time series, i.e., over the periods 1850−1879 and 1976−2005, respectively,
the result shows a decrease of 11.2 %, from 16.9 Sv to 15.1 Sv during each period.
Modeling results indicate that the AMOC has a multidecadal cycle; however, the power
spectrum of its strength time series did not show a multidecadal oscillation (not shown).
The standard deviation of the detrended maximum AMOC strength time series is 1.4
Sv.

8        Figure 15 shows the mean sea ice concentration simulated by BESM-OA2.5 for

the end of the winter and the summer seasons for each hemisphere over the period
1971−2000. The thick black lines represent the 15 % climatological values for the
period 1971−2000 given by the 20CRv2 Reanalysis. The sea ice concentration at the
end of the Arctic winter was overestimated in the Atlantic, specifically north of
Scandinavia (Fig. 15a). However, at the end the Arctic summer, the sea ice
concentration was underestimated (Fig. 15b). At the end of the Antarctic summer, the
model showed a significant underestimation of the sea ice concentration (Fig. 15c),
whereas at the end of the Antarctic winter, the model generally overestimated the
extension of the sea ice concentration over the Southern Ocean (Fig. 15d). Such
seasonal sea ice concentration variations are likely related to the radiative net bias
inherent in the model at high latitudes, which results in the generation of higher sea ice
extensions during the winter season in each hemisphere compared with those from the
Reanalysis dataset and excessive sea ice melting during the summer season in each
hemisphere.

**4.2 Climate variability**

In this section, we evaluate the most prominent global climate variability patterns. This evaluation allows us to understand the ability of the model to correctly simulate atmospheric internal and ocean-atmosphere coupled variabilities in the climate system.

**4.2.1 Tropical variability**

**4.2.1.1 El Niño-Southern Oscillation**

The El Niño-Southern Oscillation (ENSO) in the equatorial Pacific Ocean is one of the most prominent climate variability phenomena on interannual time scales (Dijkstra, 2006), and it has strong effects on regions surrounding the Indian and Pacific Ocean and regions that are influenced by its teleconnections (see McPhaden et al., 2006 and references therein). There are many methods to evaluate the ENSO variability. In the present study, the EOF was applied to detrended monthly SST anomalies over the tropical Pacific Ocean (30 ° S−30 ° N; 240 °−70 ° W) for the period 1950−2005 for both the BESM-OA2.5 historical simulation and the ERSSTv4 data. Figures 16a and 16b show the leading EOF patterns associated with the El Niño/La Niña variability. The model was ineffective at simulating the El Niño/La Niña variability, with lower amplitudes in the SST variability and with the center of maximal variability confined to the eastward part of the basin. The model's leading EOF explains 17.9 % of the total variance, substantially less than the 45 % explained by observations. The lower amplitude of the simulated El Niño/La Niña can be verified in the power spectrum of the leading principal component (PC) shown in Figures 16c and 16d. Even though the

simulation shows two significant peaks between 2−4 years cycle (Fig. 16c), which is within the range of the period cycle given by the leading PC of the observations (3−7 years cycle; Fig. 16d), the amplitude of the simulated variance is lower than that of the observations.

Figure 17 shows the spatial correlation between the detrended monthly anomalies of the Niño-3 index (defined inside the black rectangular area, bounded by 5°S−5° N, 90°−150° W) and detrended monthly anomalies of global SSTs computed by BESM-OA2.5 and ERSSTv4 over the period 1900−2005. The model did not show strong correlation at grid points inside the Niño-3 area, which is a signal that the El Niño/La Niña spatial pattern is weakly simulated. The horseshoe pattern of negative correlation observed over the Pacific Ocean is also weakly simulated by the model, particularly in the westward equatorial region. The positive correlation between the observed SSTs over the Indian Ocean and the Niño-3 index was absent in the model's simulation. It is worth mentioning that the model simulated the observed correlation pattern of SST anomalies over the Atlantic Ocean with the Ninõ-3 index, although it is not so robust (Fig. 17a).

**4.2.1.2 Atlantic Meridional Mode**

The leading modes of coupled ocean-atmosphere variability over the Tropical Atlantic Ocean are the zonal mode, also referred as equatorial mode (Zebiak, 1993; Lutz et al., 2015), and the meridional mode, also referred as the interhemispheric mode (Nobre and Shukla, 1996). The first is an ENSO-like phenomenon that emerges in the Gulf of Guinea mainly during the boreal summer and has a strong impact on West African precipitation (Zebiak, 1993; Lutz et al., 2015). The second is characterized by a

cross-equatorial SST gradient associated with meridional wind stress toward the warmer SST anomalies. The maximal amplitude of the meridional mode occurs during the boreal spring and influences the precipitation in Northeast Brazil and West Africa (Nobre and Shukla, 1996; Chang et al., 1997; Chiang and Vimont, 2004). The Atlantic Meridional Mode (AMM) has an interannual and decadal temporal scale of variability and results from a thermodynamic coupling between wind speed, the sea surface evaporation induced by the wind stress, and the SST, a mechanism known as wind-evaporation-SST feedback (WES feedback, Xie and Philander, 1994; Chang et al., 1997; Xie, 1999). To evaluate the AMM simulations, a joint EOF of SST and wind stress (Taux and Tauy) fields was computed, as such variability is intrinsic to the coupled ocean-atmospheric system. Figure 18 shows the AMM simulated by BESM-OA2.5 and that obtained via observed data. The AMM pattern simulated by the model is similar to that obtained from observations, regardless of the weaker gradient pole in the South Atlantic. Nevertheless, the variance explained by the model (10.7 %) is very close to the observed value (11.8 %). The patterns shown in Figure 18 are defined as a positive phase of the AMM, with the inter-hemisphere cross-equatorial wind from the south to the north and with corresponding negative SST anomalies over the southern pole and positive SST anomalies over the northern pole (the negative phase of the AMM is the reverse pattern). Over the second half of the twentieth century, the AMM showed a predominant decadal periodicity of 11−13 years. Figures 18c and 18d show the power spectra of the PC of the AMM patterns simulated by the model and based on observed data, respectively. It is possible to see that the pattern simulated by BESM-OA2.5 shows, similar to that derived from the observed data, a predominant periodicity on decadal timescales.

**4.2.1.3 South Atlantic Convergence Zone**

The South Atlantic Convergence Zone (SACZ) is characterized by an intense NW-SE oriented cloud band that extends from the Amazon Basin to the South Atlantic subtropics, mainly during the austral summer (Nogués-Paegle and Mo, 1997; Carvalho et al., 2004; de Oliveira Vieira et al., 2013). The formation of the SACZ has a strong influence on the precipitation over southeast South America and is considered, together with the convection activity over the Amazon Basin, the main component of the South American Monsoon System (Jones and Carvalho, 2002). The southern part of the SACZ normally lies over cooler SSTs (Grimm, 2003; Robertson and Mechoso, 2000). Chaves and Nobre (2004) suggest that the cloud cover resulting from the formation of the SACZ over the ocean tends to block solar radiation, thus leading to cooler SSTs beneath. AGCM are unable to simulate the precipitation over the cooler SSTs caused by the SACZ (Marengo et al., 2003; Nobre et al., 2006; Nobre et al., 2012), since such models tend to increase the precipitation over warmer SSTs as a hydrostatic response. Nobre et al. (2012) showed that coupled AOGCMs can simulate SACZ formation over colder SST anomalies, as this class of models incorporates atmosphere-ocean surface thermodynamic coupling. Following Nobre et al. (2012), a correlation exists between the seasonal precipitation and SST anomalies during the austral summer (DJF) over the tropical South Atlantic ($40°S-10°N; 70°W-20°E$) over the period 1979−2010 for observations and over the period 1971−2002 for the model; therefore, 32 years were used. Figure 19 shows the rainfall-SST anomaly correlation maps for both BESM-OA2.5 and the observations. BESM-OA2.5 could simulate an inverse correlation between the precipitation and SST in the southeast of Brazil (near $20°S$), indicating it's

capacity in simulating precipitation over cooler SSTs, a feature related to the formation

of SACZ (which results in cooler SSTs). Its noteworthy in Figure 19 that BESM-OA2.5

could generate both positive and negative SSTA-rainfall correlations over the equatorial

Atlantic (positive, thermally direct driven circulation over the equatorial region, and

negative, thermally indirect driven atmospheric circulation over the SW tropical

Atlantic, Figure 19a), a feature also present in the observation correlation map shown in

Figure 19b.

**4.2.1.4 Madden-Julian Oscillation**

The Madden-Julian Oscillation (MJO) is the primary intraseasonal variability

(30−90 days) over the eastern Indian and western Pacific tropical regions and consists

of deep convection events coupled to atmospheric circulation that propagate together

eastward through the equatorial region (Madden and Julian, 1971, Madden and Julian,

1972; Zhang, 2005). Influence of MJO events on large-scale phenomena has been

reported, as in the case of the evolution of ENSO (e.g., Takayabu et al., 1999),

formation of tropical cyclones (e.g., Liebmann et al., 1994) and in the North Atlantic

Oscillation (e.g., Lin et al., 2009). To evaluate the MJO simulated by the model,

wavenumber-frequency power spectrum analyses were performed for tropical (10 °S–

10 °N) averaged daily outgoing long-wave radiation (OLR) and for the daily zonal wind

component at the 850 hPa pressure level (U850) during the boreal winter (Nov–Apr)

over the period 1971–2000. To compute and plot the wavenumber-frequency power

spectra the MJO Simulation Diagnostic package was used (details in Waliser et al.,

2009).

Figures 20a and 20b show the wavenumber-frequency power spectra for the OLR from BESM-OA2.5 and 20CRv2, respectively. Although BESM-OA2.5 yielded an eastward propagating disturbance with wavenumber 1, it was characterized by a lower frequency (> 80 days) compared to the maximal peak within the 30–80 day frequency band shown by the 20CRv2 data, despite its spread over frequencies less than 80 days. This observed peak has more energy for wavenumber 2. A westward propagating disturbance (negative frequencies) with weaker energy than the eastward propagating counterpart appears in the 20CRv2 data sets, with a peak for wavenumber 2. Similarly, BESM-OA2.5 also showed a westward propagating disturbance with weaker energy for wavenumbers 1–3. The wavenumber-frequency power spectrum for U850 in 20CRv2 showed an eastward propagating disturbance that peaked at the 30–80-day frequency band with wavenumber 1 (Fig. 20d). In the case of BESM-OA2.5, there was an eastward propagation with a periodicity slightly higher than 80 days for wavenumber 1, but this disturbance spread over different frequencies outside of the 30–80-day frequency band (Fig. 20c). BESM-OA2.5 also presented a westward propagating disturbance that is absent in the Reanalysis. BESM-OA2.5 poorly simulated the MJO and underestimated its amplitude. However, the MJO has been highlighted as a phenomenon that climate models struggle to properly simulate, especially via underestimation of the OLR and representation of a coherent eastward propagation (Kim et al., 2009; Ahn et al., 2017).

**4.2.2 Extratropical variability**

**4.2.2.1 North Atlantic Oscillation**

The North Atlantic Oscillation (NAO) is a major atmospheric variability pattern

that occurs in the North Atlantic that is characterized by oscillations in the sea level pressure (SLP) differences between Iceland and Portugal (Wanner et al., 2001; Hurrel et al., 2003). The NAO has a robust impact in the Euro-Atlantic region (Hurrell et al., 2003; Hurrell and Deser, 2009), and the notable work of Namias (1972) connected the droughts in Northeast Brazil to NAO variations. Recent studies show that it has teleconnections to East Asia (e.g., Yu and Zhou, 2004; Wu et al., 2012). The NAO's influence on rapid climate changes in the Northern Hemisphere has been highlighted in Delworth et al. (2016, thus making its correct simulation more critical. Since the NAO's largest amplitude of variation occurs mainly during the boreal winter, the analyses presented here are centered on this season, and the period used to perform these analyses was 1950−2005. The leading EOF of the SLP averaged over the boreal winter season (DJF) in the Euro-Atlantic region showed that the NAO is well simulated by BESM-OA2.5 (Fig. 21a), as it simulations of the NAO dipole centers and their amplitudes were very similar to the observed pattern (Fig. 21b). The variances explained by the leading EOF were also similar, 50.2 % and 44 % for BESM-OA2.5 and the Reanalysis, respectively. The spectral analysis of the leading PCs showed that BESM-OA2.5 captures the ~2.5-year cycle in the time variability, but failed to capture the ~8-year cycle (Fig. 21c and 21d). It is interesting to note that BESM-OA2.5 simulated a NAO spatial pattern without capturing its low-frequency variability. Based on an analysis of the NAO variability, we propose that it is not necessary to analyze the Northern Annular Mode (NAM), since both are manifestations of same mode of variability (Hurrell and Deser, 2009).

**4.2.1.2 Pacific-North America pattern**

1       Together, the NAO and the Pacific-North American pattern (PNA) are the

dominant atmospheric internal modes over the boreal hemisphere. The PNA is
characterized by four centers of the 500 hPa geopotential heightanomalies in the North
Pacific and over North America, each center located over Hawaii, in the south of the
Aleutian Islands, in the intermountain region of North America, and in the Gulf Coast
region of the U.S.A., representing the centers of action of a stationary wave train
extending from the tropical Pacific into North America (Wallace and Gutzler, 1981).
The PNA exerts a significant influence on surface temperature and precipitation over
North America (Leathers et al., 1991). Some studies have shown that although the PNA
is an internal atmospheric variability phenomenon, it is influenced by other climate
variabilities, including the ENSO and the Pacific Decadal Oscillation (PDO) (see Straus
and Shukla, 2002; Yu and Zwiers, 2007).

13       Similar to the NAO, the PNA has its largest variation of amplitude during the

boreal winter; therefore, the present analyses were performed for this season. Following
Wallace and Gutzler (1981), we constructed one-point correlation maps for BESM-
OA2.5 and the 20CRv2 Reanalysis to evaluate the capacity of the model to reproduce
the PNA pattern. The one-point correlation maps correlate the 500 hPa geopotential
height at the reference point (45 $^{\circ}$N, 165 $^{\circ}$W) with all of the other grid points on the
map domain (0 $^{\circ}$–80 $^{\circ}$N; 240 $^{\circ}$–70 $^{\circ}$W). The time series used to perform the correlations
were an averaged boreal winter seasonal (DJF) dataset over the period 1950−2005. The
time series were departed from their long-term means and normalized at each grid point
prior the correlation computation. Figure 22 shows the one-point correlation maps for
BESM-OA2.5 (Fig. 22a) and 20CRv2 (Fig. 22b). In this figure, it is possible to observe
the four geopotential height centers simulated by the model, which show a stronger
correlation  when compared with the Reanalysis correlation maps shown in Figure 22b.
**4.2.1.2 Pacific-South America patterns**
The second and third EOF of the 500 hPa geopotential height over the Southern
Hemisphere (20°–90° S) shares a notable resemblance to the Pacific-South America
(PSA) teleconnection pattern (Mo and Peagle, 2001). PSA patterns are stationary
Rossby wave trains that extend from the central Pacific to Argentina, in which the PSA1
(EOF2) is a response to the ENSO and the PSA2 (EOF3) is associated with the quasi-
biennial component of the ENSO (Karoly, 1989; Mo and Peagle, 2001). These patterns
have a significant impact on rainfall anomalies over South America (Mo and Peagle,
2001). Figure 23 shows the PSA patterns simulated by BESM-OA2.5 and from the
Reanalysis. As the explained variance of EOF2 and EOF3 are similar, the EOFs seem to
be degenerate for both the Reanalysis and the model simulation. To relax the
orthogonality constraint, a rotated EOF (REOF) retaining the first 10 modes was
performed. The REOF2 and REOF3 resembled the EOF2 and EOF3, respectively,
implying that they are independent modes. The PSA pattern was well simulated by
BESM-OA2.5, although the model changed the order of the EOF patterns. BESM-
OA2.5 showed an anomaly south of South Africa (Fig. 23c) that does not appear in the
Reanalysis (Fig. 23b). PSA patterns have significant interannual and decadal
variabilities (Zhang et al., 2016). The PSA patterns simulated by BESM-OA2.5 had
significant variability only on the interannual scale, with no decadal variability (figure
not shown).
**4.2.1.4 Southern Annular Mode**
The Southern Annular Mode (SAM) is the dominant atmospheric variability in
the Southern Hemisphere, and it occurs in the extra-tropics and in the high latitudes
(Kidson,1988). It is also referred to as the Antarctic Oscillation (AAO; Gong and Wang,
1999). SAM variability is characterized by anomalous variations in the polar low-
pressure and in the surrounding zonally high-pressure belt. The SAM can be captured
via the first EOF applied to different atmospheric variables, such as the sea level
pressure, different geopotential height levels and the surface air temperature (Kidson,
1988; Rogers and van Loon, 1982; Thompson and Wallace, 2000). To evaluate the
capacity of BESM-OA2.5 to simulate this atmospheric mode of variability, EOF
analysis was applied to the monthly mean 500 hPa geopotential height field from 20 °S
to 90 °S over the period 1950−2005, for both the model and Reanalysis. The SAM
pattern simulated by BESM-OA2.5 strongly resembled the observed pattern, showing
mid-latitude 500 hPa geopotential height variation centers at the same longitudes as
those observed, although it showed differences in the amplitude values (Fig. 24).
However, the explained variance is higher compared with the observation. The
explained variances of BESM-OA2.5 and 20CRv2 are 34.1 % and 21.0 %, respectively.
The SAM is a quasi-decadal mode of variability (see Yuan and Yonekura, 2011);
however, the BESM-OA2.5 power spectrum reveals a SAM with a markedly
interannual variability, without the peak between 8 and 16 years contained in the
Reanalysis (figure not shown).
**4.2.1.5 Pacific Decadal Oscillation**

22       The observed SST anomalies over the North Pacific have shown an oscillatory

pattern in the central and western parts in relation to the tropical part and along the

North American west coast. This oscillatory shift in SST anomalies with interdecadal periodicity was termed the Pacific Decadal Oscillation (PDO), and it is defined as the leading EOF of the monthly SST anomalies over the North Pacific (Mantua et al., 1997). The positive phase of the PDO is defined when negative SST anomalies are predominate over the central and western parts of North Pacific and positive SST anomalies predominate over the Tropical Pacific and along the North American west coast. The negative phase is a reversal of this pattern. Many studies have connected the PDO with variations in precipitation regimes in different regions around the world, including the South China monsoon (e.g., Wu and Mao, 2016), the Indian monsoon (e.g., Krishnamurthy and Krishnamurthy, 2016) and, together with the ENSO, in the precipitation regime in North America (see Hu and Huang, 2009). There are different mechanisms that modulate the PDO, among which is the response of the Northern Pacific SST to the ENSO variability via the "atmospheric bridge" (for a detailed review, see Newman et al., 2016).

Following its definition (Mantua et al., 1997), the spatial pattern of the PDO was obtained by regressing the SST anomalies onto the leading normalized PC time series, as shown in Figure 25, which in this case shows the positive phase of the PDO. The EOF was applied to monthly SST anomalies over the North Pacific ($20°-60°$ N; $240°-110°$ W) over the period $1900-2005$. BESM-OA2.5 was not capable of reproducing this pattern in the leading EOF. The PDO pattern only appeared on the second EOF (Fig. 25a), with an explained variance of 14.0 % compared with 20.5 % for the observations. Although the EOF2 resembles the PDO mode, the tropical part has weaker variation compared with the observed variation. The basis for the model's

deficiency in reproducing the PDO as the leading mode of variability is probably the model's simulation of weaker ENSO variability, both on spatial and temporal scales. These deficiencies may impact the mechanisms that reproduce the PDO, mainly via the "atmospheric bridge" referred to earlier. Figures 26a and 26b show the normalized PC2 and PC1 time series of BESM-OA2.5 and ERSSTv4, respectively. It is possible to note that both time series present a multidecadal periodicity, but on different time scales, as confirmed by the power spectrum (Fig. 26c and 26d). The power spectra show that both time series possess interannual periodicity (~5−6 years), with the strongest multidecadal variability spectrum around 15 years for BESM-OA2.5, a higher frequency compared with the observed frequencies (~22 and ~40−45 years).

## 5. Summary

The ability of Earth system models to project future climate parameters based on conditions given by future scenarios of atmospheric greenhouse gas concentrations can be assessed by how accurately the models can reproduce observed climate features. Therefore, evaluation of how these models perform over historical periods for which there are observations that can be compared with model's calculations represents a key part of Earth system modelling. In this study, the BESM-OA2.5 historical simulation was evaluated for the period 1850−2005 following the CMIP5 protocol (Taylor et al., 2012), with a focus on simulations of the mean climate and key large-scale modes of climate variability.

BESM-OA2.5 is an updated version of BESM-OA2.3 (Nobre et al. 2013; Giarolla et al. 2015), which now incorporates the new Brazilian Global Atmospheric

Model (BAM; Figueroa et al., 2016). This new version reduced a mean global bias of
the energy balance at the top of the atmosphere from $-20$ W m$^{-2}$ to $-4$ W m$^{-2}$.
Moreover, systematic errors were reduced in wind, humidity and temperature in the
surface layer over oceanic regions via the inclusion of formulations presented by
Jiménez et al. (2012).
The analysis of the mean climate showed that the model can simulate the general
mean climate state. Nevertheless, some significant biases appeared in the simulation,
such as a double ITCZ over the Pacific and Atlantic Oceans, some notable regional
biases in the precipitation field (e.g., over the Amazon and Indian regions) and in the
SST field (e.g., south of Greenland). Nevertheless, the model has shown an
improvement in simulating the ITCZ and a reduction in the global precipitation RMSE
compared with that of BESM-OA2.3. BESM-OA2.5 shows a nearly globally positive
SST bias that was absent in version 2.3; however, the SST RMSE was slightly reduced
in the newer version of the model.
The most relevant climate patterns on interannual to decadal time scales
simulated by BESM-OA2.5 were compared with the ones obtained from observations
and Reanalysis. Over the Pacific, the ENSO was simulated with a lower amplitude of
variability than that recorded from the observations, and this weak ENSO seems to
impact other Pacific variability patterns, such as the PDO. Conversely, the major
phenomena over the Atlantic basin were well represented in BESM-OA2.5 simulations.
This was the case for the Tropical Atlantic mode of interhemispheric variability
(AMM), which was very well simulated by the model in terms of the spatial pattern and
temporal variability. It is worth noting that this mode is considered to be poorly

simulated by the models used in the Intergovernmental Panel on Climate Change (IPCC) fifth assessment report (AR5) (Flato et al., 2013). It is also relevant to highlight the ability of BESM-OA2.5 to represent the enhanced rainfall over the cooler waters of the SW Tropical Atlantic that are associated with the South Atlantic Convergence Zone (SACZ). The ability of the model to simulate the AMM and SACZ is an important result, since one of our main aims is to represent the modes that directly impact the precipitation over South America. The AMOC reproduced by BESM-OA2.5 has a meridional overturning structure comparable with the ensemble AMOC simulated by the CMIP5's models. BESM's maximum AMOC strength average value was slighter lower than the average value observed by the RAPID project, but well within the range of the observed mean-square-root variability. Although the averaged maximum strength AMOC simulated by the CMIP5 models is within the observed mean range square root variability, most models tend to simulate a strong AMOC, with a maximum strength above 20 Sv, which is outside of the range (Zhang and Wang, 2013). The NAO atmospheric variability, which is well simulated by the CMIP5 models (Ning and Bradley, 2016), is also very well simulated by BESM-OA2.5. In the extra-tropics, BESM-OA2.5 could reproduce major variabilities in both hemispheres, such as the PNA, PSA, and the SAM teleconnection patterns, relatively well compared to the CMIP5 models, which reproduces the PNA (Ning and Bradley, 2016) and SAM (Zheng et al. 2013).

Similar to Nobre et al. (2013), this study aimed to evaluate BESM-OA2.5 by comparing the most important features of the climate system simulated by the model with observations and Reanalysis. The next version of the model (BESM-OA2.8) is

already under development. In this new version, the MOM4p1 ocean model has been replaced by the MOM5. Regarding the atmospheric model, new developments have been carried out to improve BAM's capacity, the most important being the inclusion of a humidity scheme at the planetary boundary layer, a new dynamic core and new cloud cover scheme (Figueroa et al., 2016). This new BESM version confronts the challenge of improving the precipitation simulation, in particular alleviating the deficit over the Amazon. The ENSO is a large-scale phenomenon that will be scrutinized to understand the reasons for weak variability. The other feature of the model is the weaker warming when the $CO_2$ equivalent is used as the only forcing compared with the warming predicted by other CMIP5 models that do not consider the direct and indirect effects of atmospheric aerosols. If BESM-OA2.5 performs consistently with CMIP5 models, then it would underestimate the warming observed over the last decades. Because models can respond in different ways to external forcing, an aim in the near future is to carry out a numerical experiment in which the model is forced with observed aerosol concentration estimates (as a read-in field) to address to what extent BESM is affected. In the future, a study comparing BESM-OA versions 2.5 and 2.8 is planned to fully explore and report the advances made in the modeling work over the last couple years. Such a study will provide a broader perspective on the technical challenges overcame throughout this project and will assess the improvements achieved in each version of the model for better simulating the climate system.

## Code and data availability

The BESM-OA2.5 source code is freely available after signature of a license agreement. Please contact Paulo Nobre to obtain the BESM-OA2.5 source code and data.

## Competing interests

There are no competing interests of which the authors are aware.

## Author contributions

SFV conducted the analyses and wrote the manuscript, under the supervision of PN. PN, EG, VC, MBJ, ALM, SNF, JPB, PK worked in the development of the new version of the model. VC and MBJ conducted the experiments. All of the authors contributed the revision of the manuscript.

## Acknowledgements

This research was partially funded by FAPESP (2009/50528-6), FAPESP (2008/57719-9) and by the National Institute of S&T for Climate Change (CNPq 573797/2008-0). SFV is supported by a Ph.D. grant funded by CAPES. MBJ is supported by a grant funded by FAPESP (2018/06204-0). The authors would like to acknowledge Rede CLIMA, FAPESP and INPE for the use of their supercomputer facility, which made this work possible. The Twentieth Century Reanalysis Project data sets (20CRv2) were provided by the U.S. Department of Energy, Office of Science Innovative and Novel Computational Impact on Theory and Experiment (DOE INCITE) program, the Office of Biological and Environmental Research (BER), and the National Oceanic and

Atmospheric Administration Climate Program Office. The GPCP combined
precipitation data sets were developed and computed by the NASA/Goddard Space
Flight Center's Mesoscale Atmospheric Processes Laboratory. The HadCRUT4 data
sets were provided by the Met Office Hadley Centre and the University of East
Anglia/Climatic Research Unit. The ISCCP D2 data sets were provided through the
International Satellite Cloud Climatology Project, maintained by the ISCCP research
group at the NASA/Goddard Institute for Space Studies. The Extended Reconstructed
Sea    Surface    Temperature    (ERSSTv4)    data    were    provided    by    the
NOAA/OAR/ESRL/PSD. The data from the RAPID-WATCH MOC monitoring project
were funded by the Natural Environment Research Council. The authors acknowledge
the World Climate Research Programme's Working Group on Coupled Modelling,
which is responsible for CMIP, and we thank the climate modeling groups (listed in
Table 1 of this paper) for producing and making available their model output. For
CMIP, the U.S. Department of Energy's Program for Climate Model Diagnosis and
Intercomparison provides coordinating support and led the development of the software
infrastructure, in partnership with the Global Organization for Earth System Science
Portals. This work is part of the Ph.D. dissertation of SFV under the guidance of CN
and PN.

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

## 1   **List of Figures**

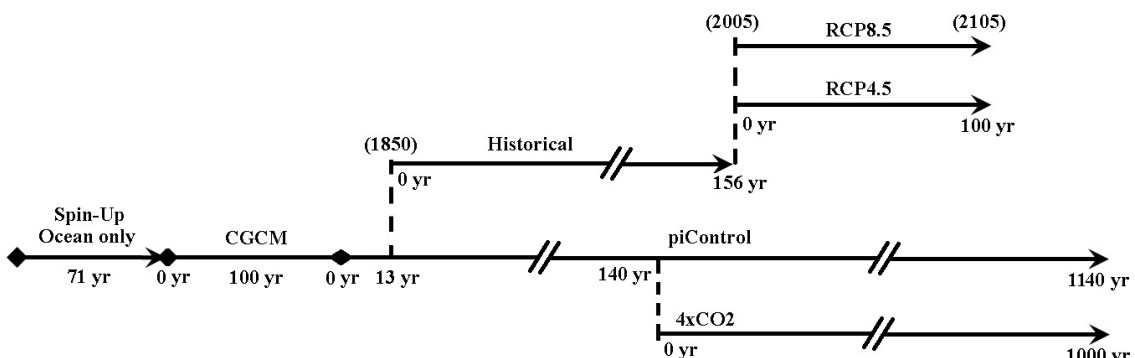

4   Figure 1 – A schematic of the principal simulations carried out by BESM-OA2.5 using

5   different forcing conditions based on the CMIP5 protocols. The dates for the Historical

6   and RCP simulations are from the actual calendar years.

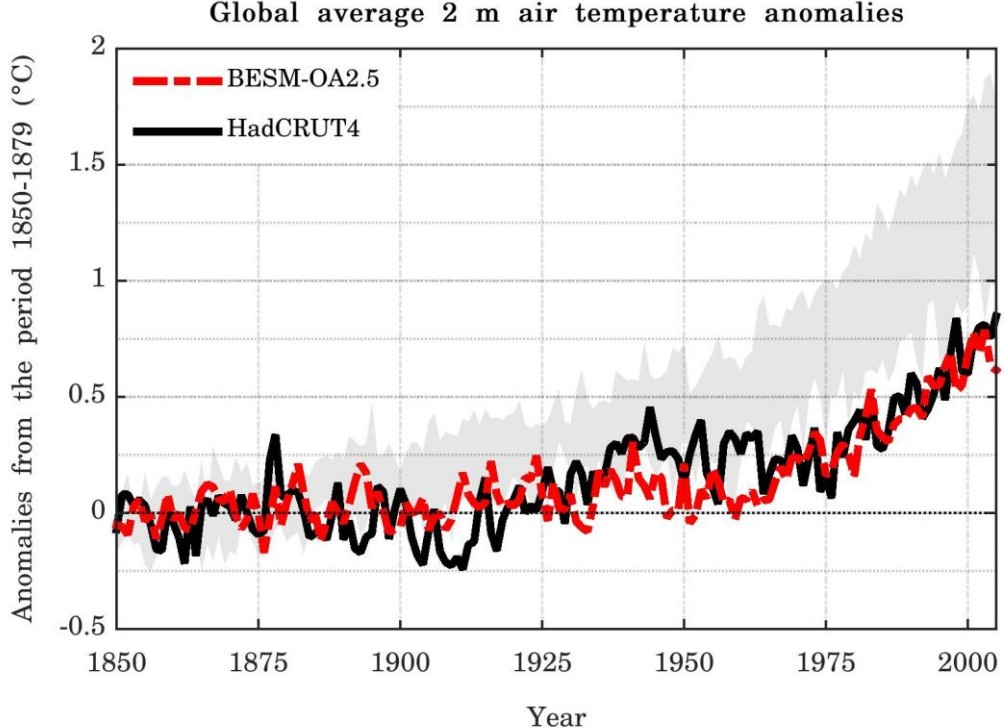

Figure 2 – Global averaged 2-m annual mean air temperature anomalies relative to the

period 1850–1879 as simulated by BESM-OA2.5 (dashed red line) and observed (solid

black line). The grey shadow represents the spread of 11 CMIP5 models (historical

GHG simulations). The CMIP5 model anomalies were also computed relative to the

period 1850–1879, with exception of GFDL-ESM2M and HadGEM2-ES, whose

anomalies were computed relative to the periods 1861−1890 and 1860−1889,

respectively. Units are in °C.

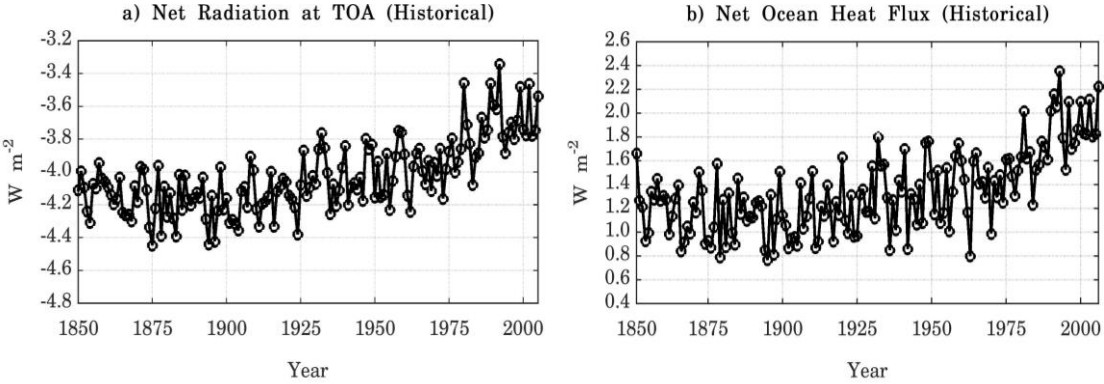

Figure 3 – Annual average time series for the global average (a) net of the radiation at the TOA (positive values indicate that the atmosphere is warming) and (b) net of the ocean/atmosphere heat flux (positive values indicate that the ocean is warming), as simulated by the Historical run over the period 1850–2005 (156 years). Units are in W $m^{-2}$.

**a) Annual mean precipitation (BESM-OA2.5)**

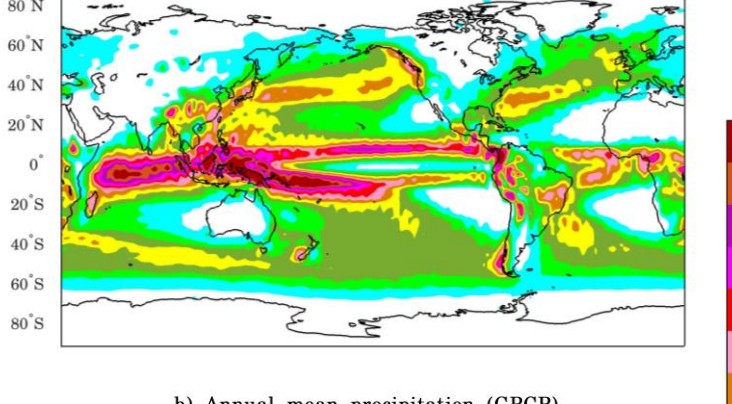

**b) Annual mean precipitation (GPCP)**

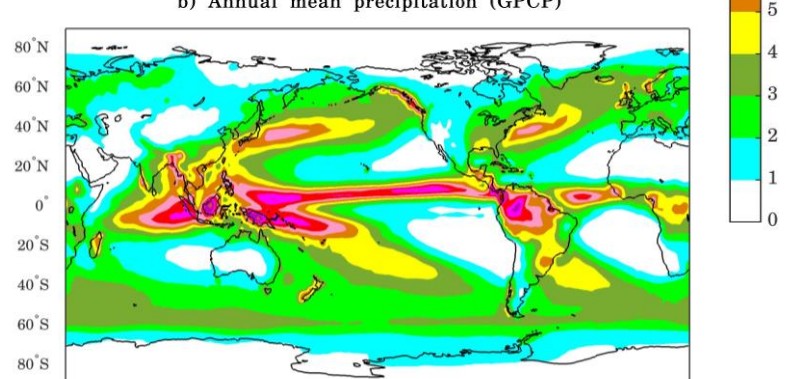

**c) BESM-OA2.5 - GPCP    mean: 0.3 mm/day    rmse: 1.4 mm/day**

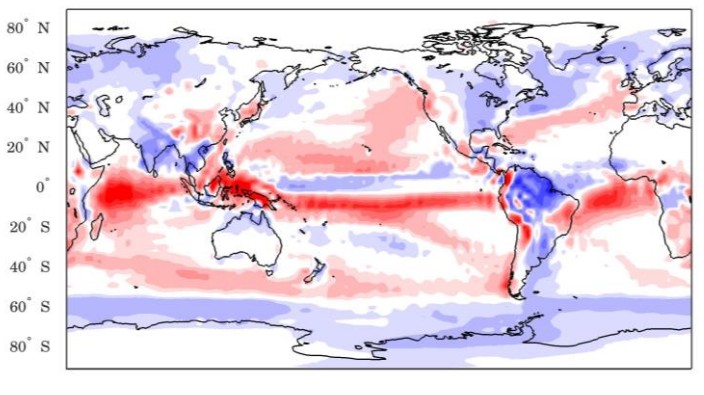

**d) BESM-OA2.5 - CMAP    mean: 0.4 mm/day    rmse: 1.5 mm/day**

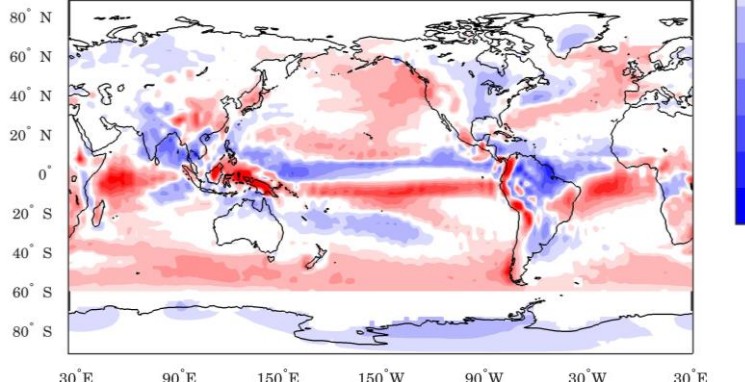

Figure 4 – Spatial maps of the annual mean precipitation for (a) BESM-OA2.5, for (b)
GPCP, (c) the bias of BESM-OA2.5 relative to GPCP and (d) the bias of BESM-OA2.5
relative to CMAP. The average values were computed over the periods 1971–2000 (for
BESM-OA2.5) and 1979–2008 (for GPCP and CMAP). Units are in mm day$^{-1}$.

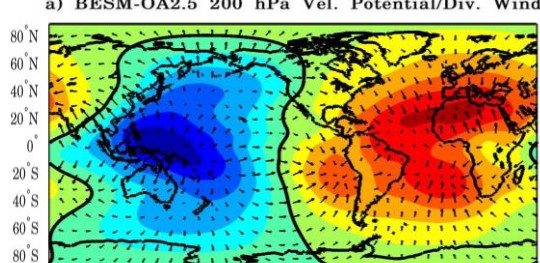

a) BESM-OA2.5 200 hPa Vel. Potential/Div. Wind

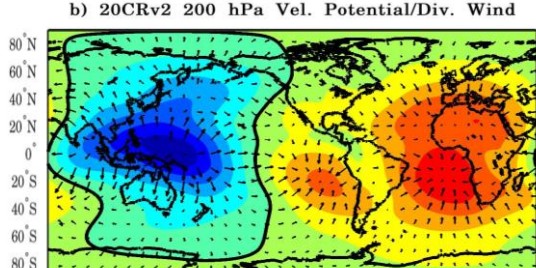

b) 20CRv2 200 hPa Vel. Potential/Div. Wind

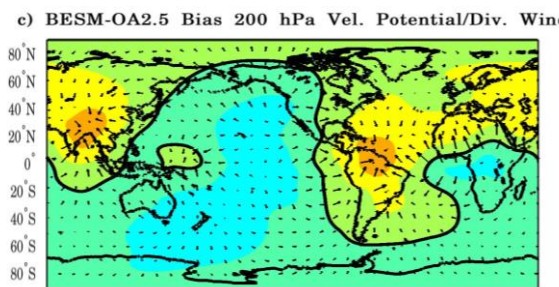

c) BESM-OA2.5 Bias 200 hPa Vel. Potential/Div. Wind

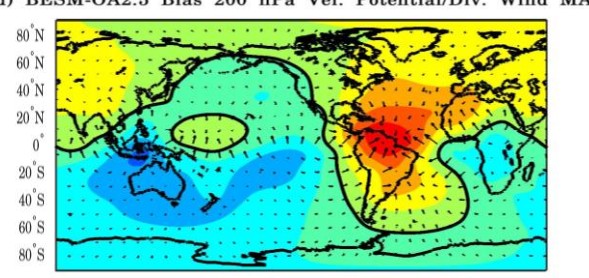

d) BESM-OA2.5 Bias 200 hPa Vel. Potential/Div. Wind MAM

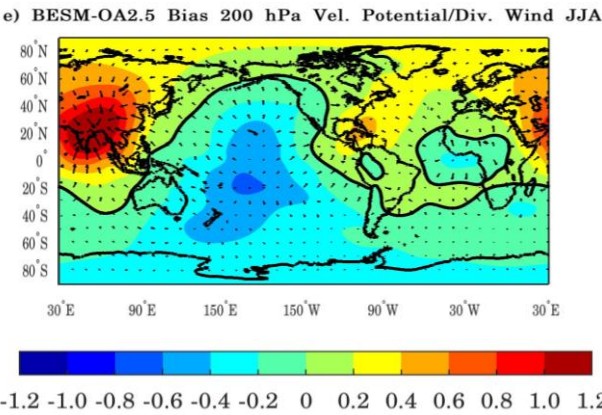

e) BESM-OA2.5 Bias 200 hPa Vel. Potential/Div. Wind JJA

-1.2 -1.0 -0.8 -0.6 -0.4 -0.2 0 0.2 0.4 0.6 0.8 1.0 1.2

$m^2 \ s^{-1}$

1 Figure 5 – Spatial maps showing the averaged global anomalies in velocity potential

2 and wind divergence at the 200 hPa pressure level for (a) BESM-OA2.5 and (b)

3 reanalysis. (c) The bias of the model relative to the reanalysis, (d) and (e) are the biases

4 for the MAM and JJA seasons, respectively. The averages were computed over the

5 period 1950–2005. Units are in m s$^{-1}$.

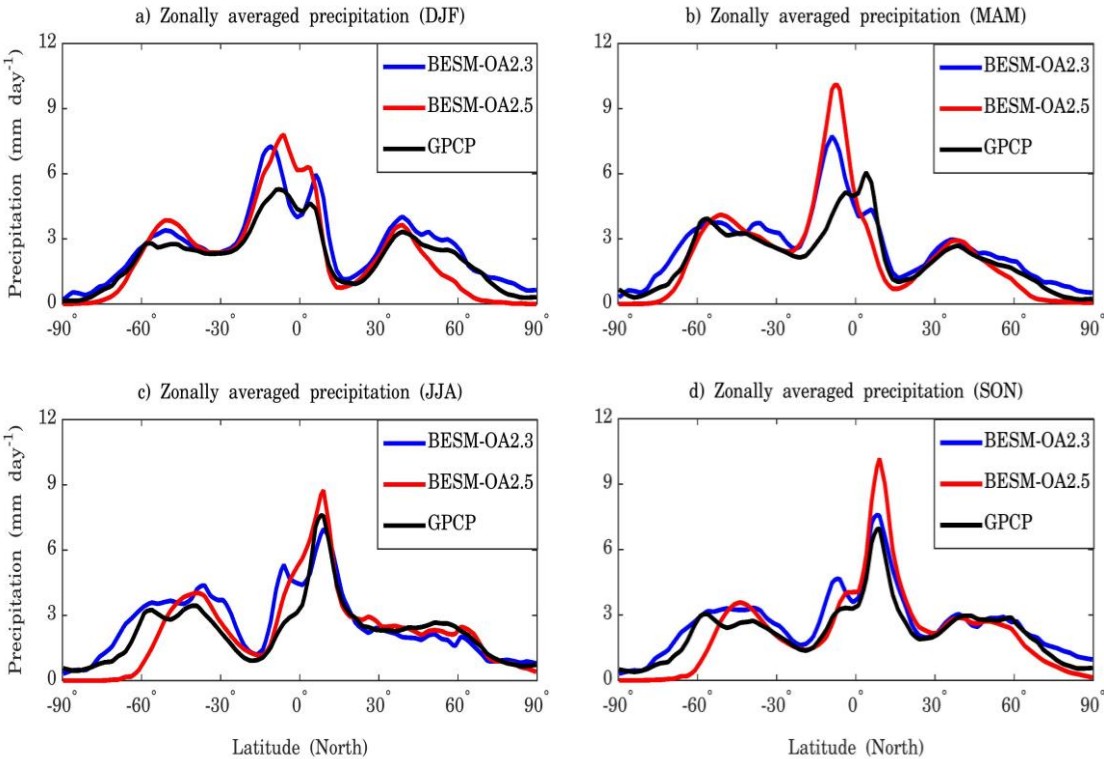

Figure 6 – Zonally averaged annual mean precipitation for the BESM-OA2.5, BESM-OA2.3 and GPCP datasets relative to the seasons DJF, MAM, JJA and SON. The zonally averaged values were computed over the periods 1971–2000 and 1979–2008, for BESM-OA2.5 and GPCP, respectively. Units are in mm day$^{-1}$.

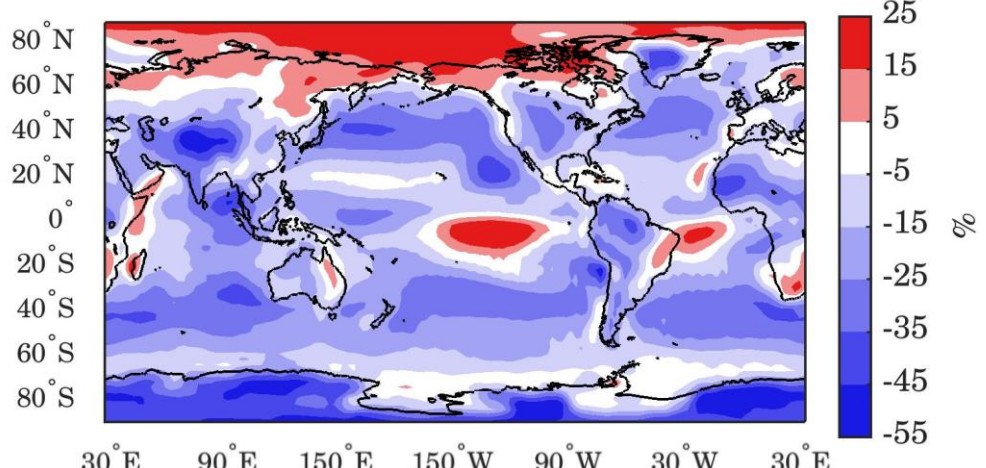

**a) Total cloud fraction (BESM-OA2.5 - ISCCP)**

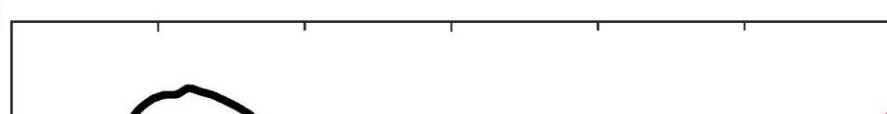

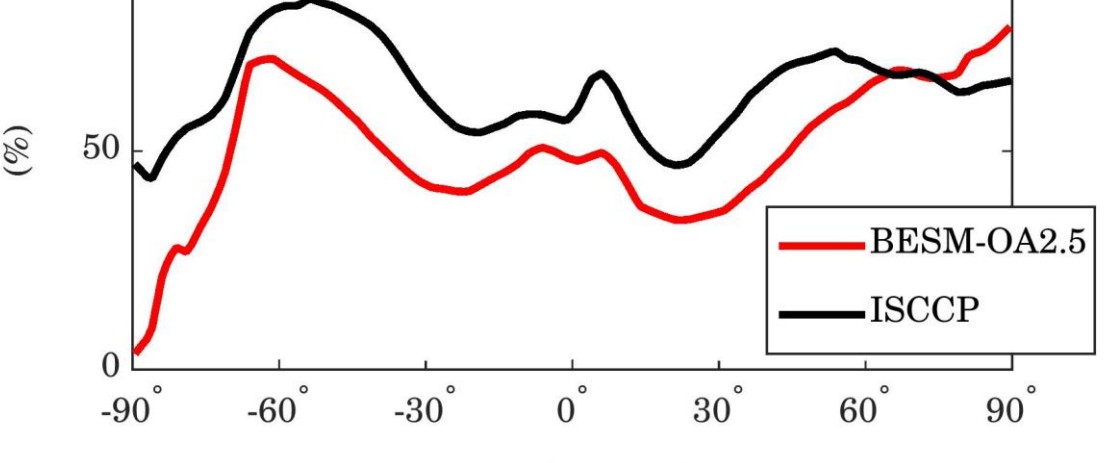

**b) Zonally averaged total cloud cover**

Figure 7 – (a) Spatial map of annual mean total cloud fraction bias of BESM-OA2.5

relative to ISCCP. (b) Zonally averaged total cloud cover for the BESM-OA2.5 and

ISCCP datasets. The periods used were 1971–2000 and 1984–2009 for BESM-OA2.5

and ISCCP, respectively. Units are in percentage.

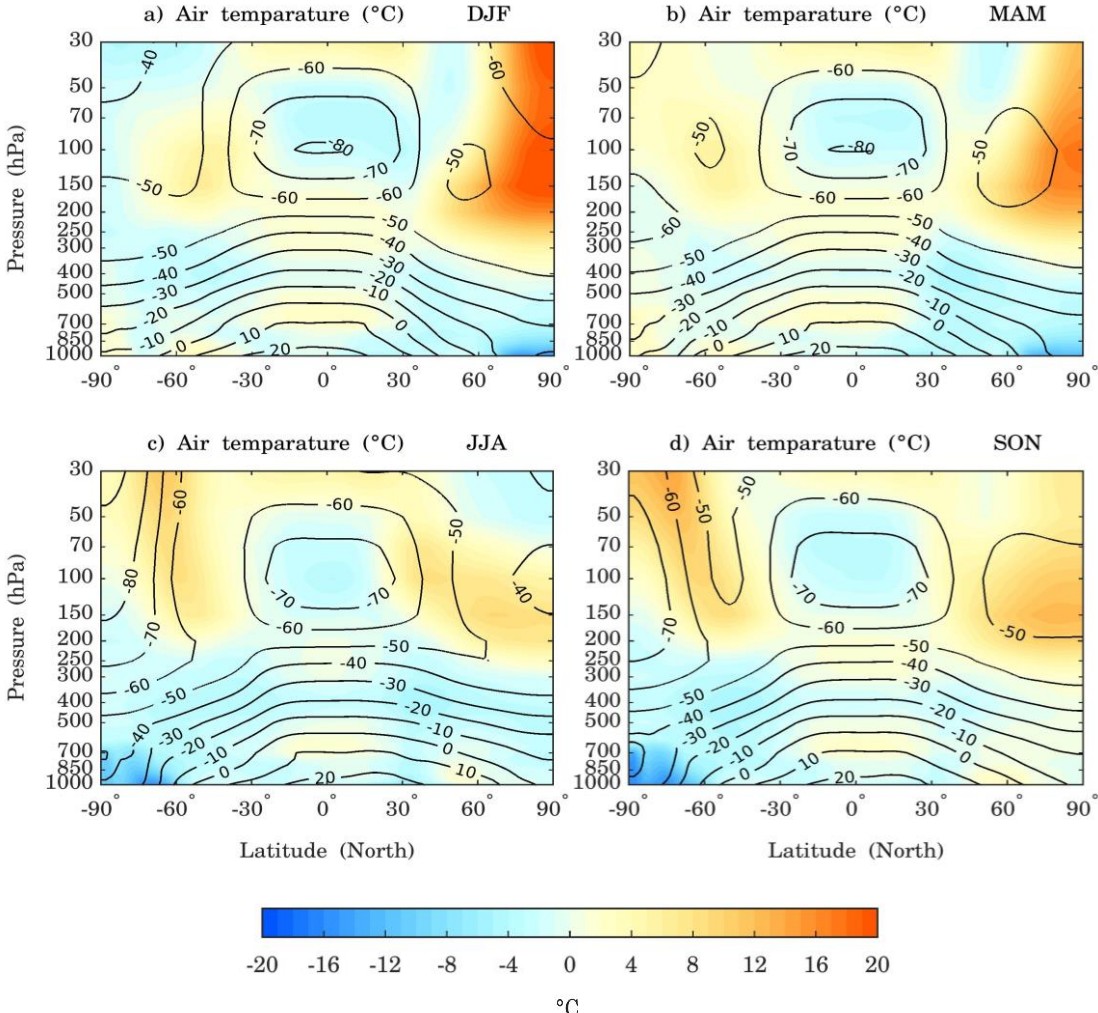

Figure 8 – Contour lines showing the zonally averaged vertical air temperatures for BESM-OA2.5 and the difference between the BESM-OA2.5 and 20CRv2 datasets are shaded in. Both are averaged over the period 1971–2000. The units are in ℃ and the contour interval is 10 ℃.

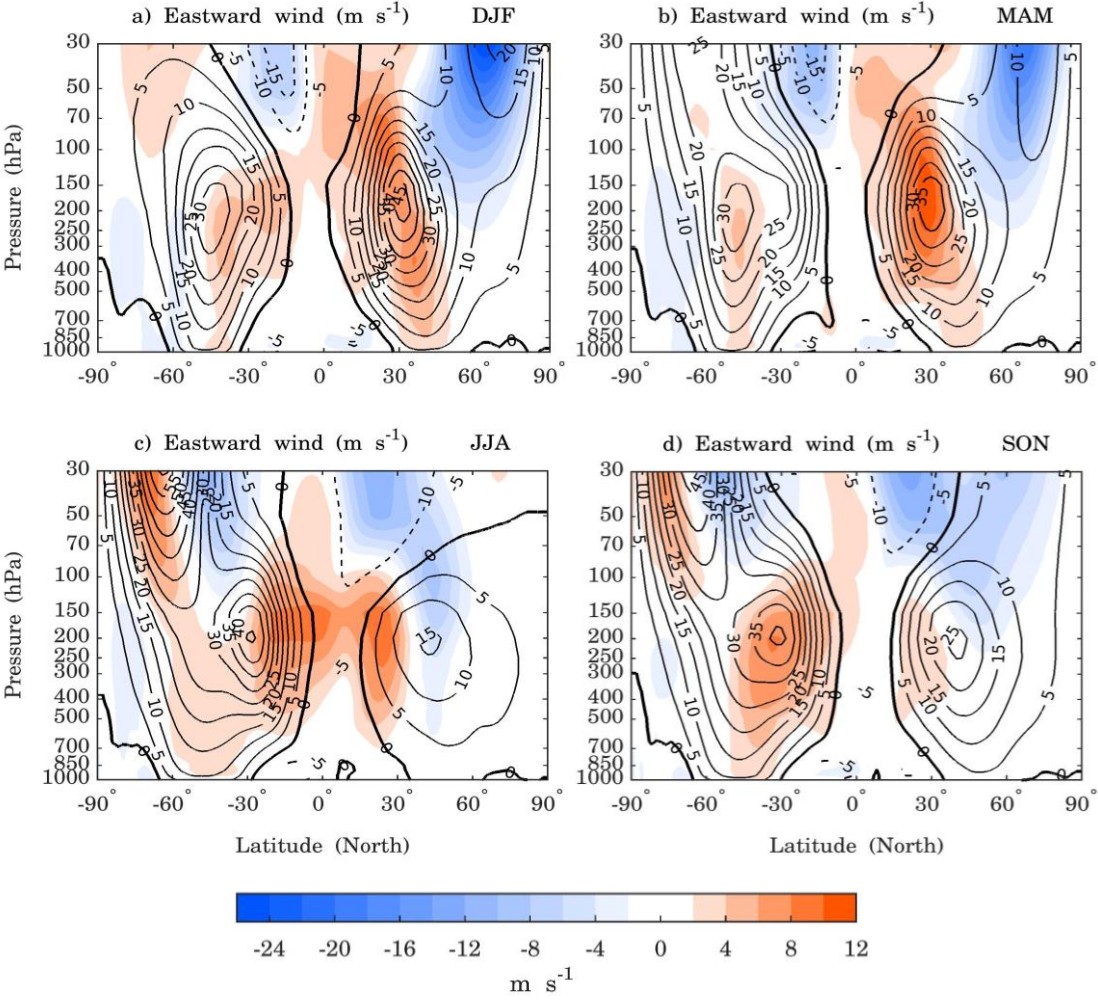

Figure 9 – Contour lines showing the zonally averaged zonal wind for BESM-OA2.5 and the differences between the BESM-OA2.5 and 20CRv2 datasets are shaded in. Both data sets were averaged over the period 1971–2000. The solid contour lines represent eastward zonal wind and the dashed contour lines represent westward zonal wind. The units are in meters per second and the contour interval is 5 m s$^{-1}$, with the zero-contour line highlighted.

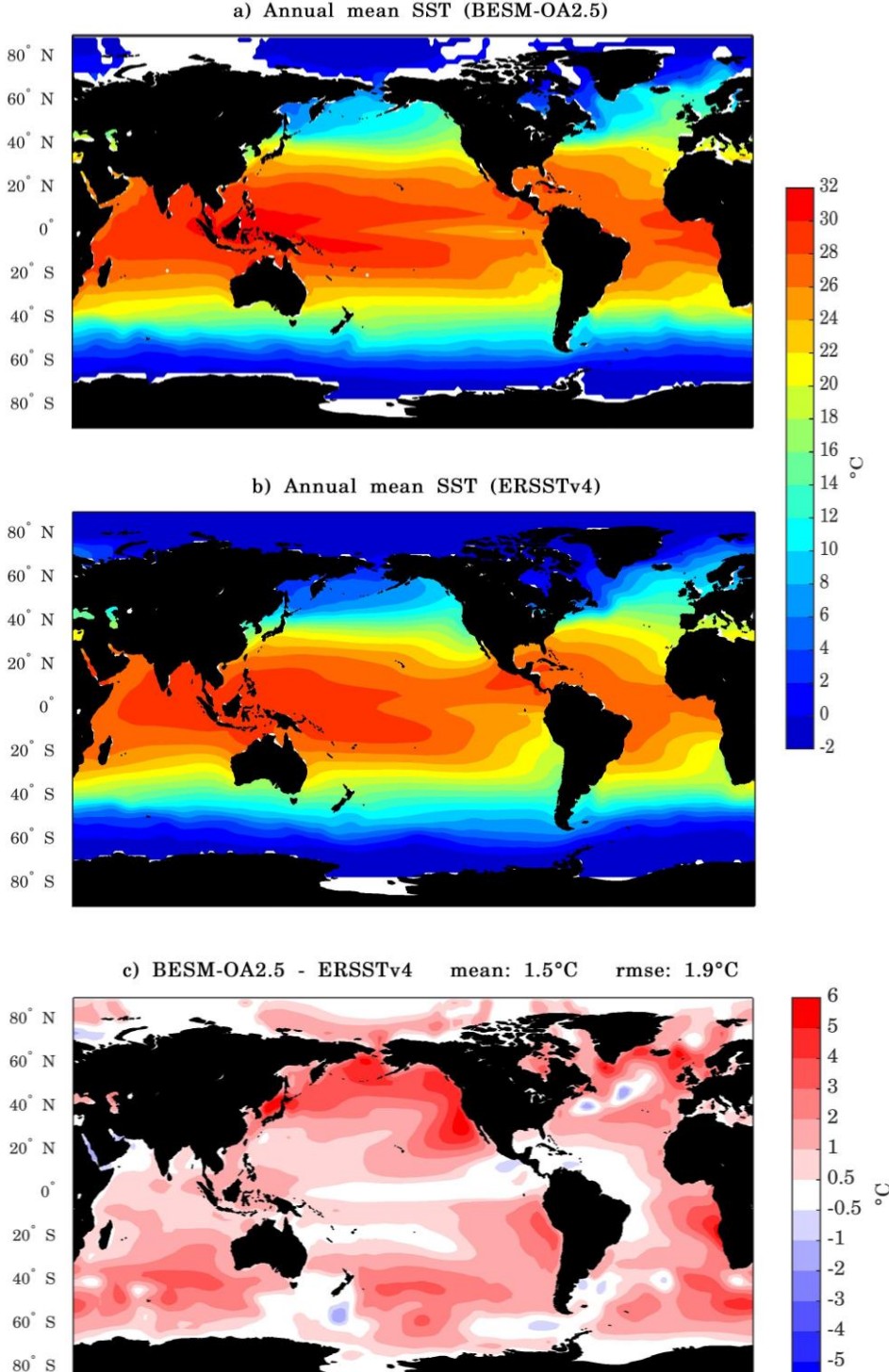

1   Figure 10 – Spatial maps of the annual mean sea surface temperatures generated by (a)

2   BESM-OA2.5 and (b) ERSSTv4 and (c) the bias of BESM-OA2.5 relative to ERSSTv4.

3   The averages were computed over the period 1971–2000. Units are in ℃.

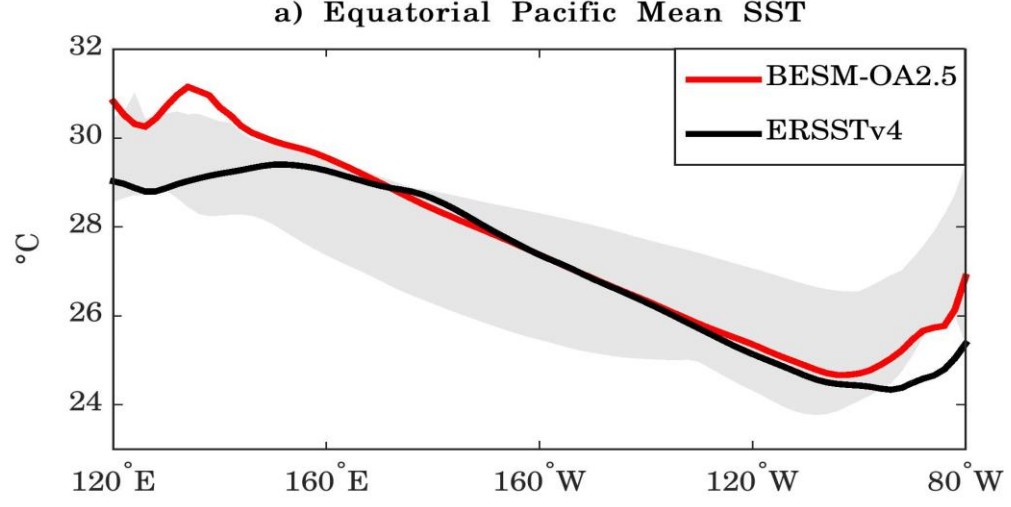

**a) Equatorial Pacific Mean SST**

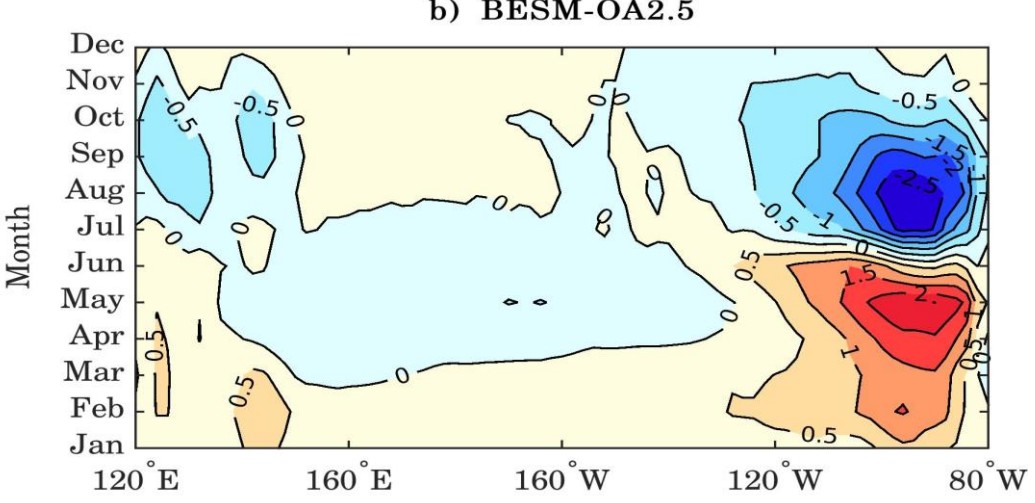

**b) BESM-OA2.5**

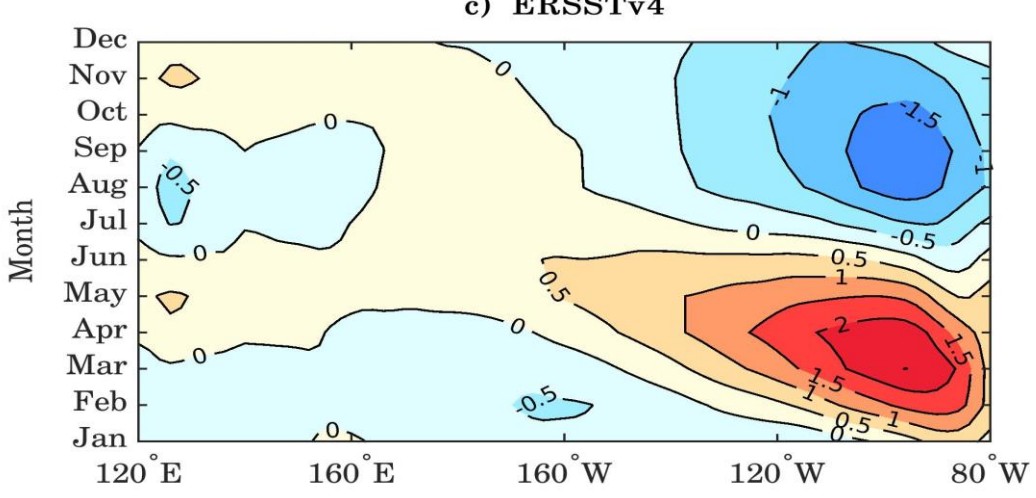

**c) ERSSTv4**

Figure 11 – (a) Mean SSTs along the equator in the Pacific Ocean and the annual cycle
of the equatorial Pacific SST anomalies for (b) BESM-OA2.5 and (c) ERSSTv4. The
equatorial region is defined by averaging over $2\,^{\circ}$ S–$2\,^{\circ}$ N. The BESM-OA2.5 and
ERSSTv4 data were averaged over the period 1971–2000. In (a), the grey shadow
represents the spread of 11 CMIP5 models, which were also averaged over the period
1971–2000. Units are in ℃.

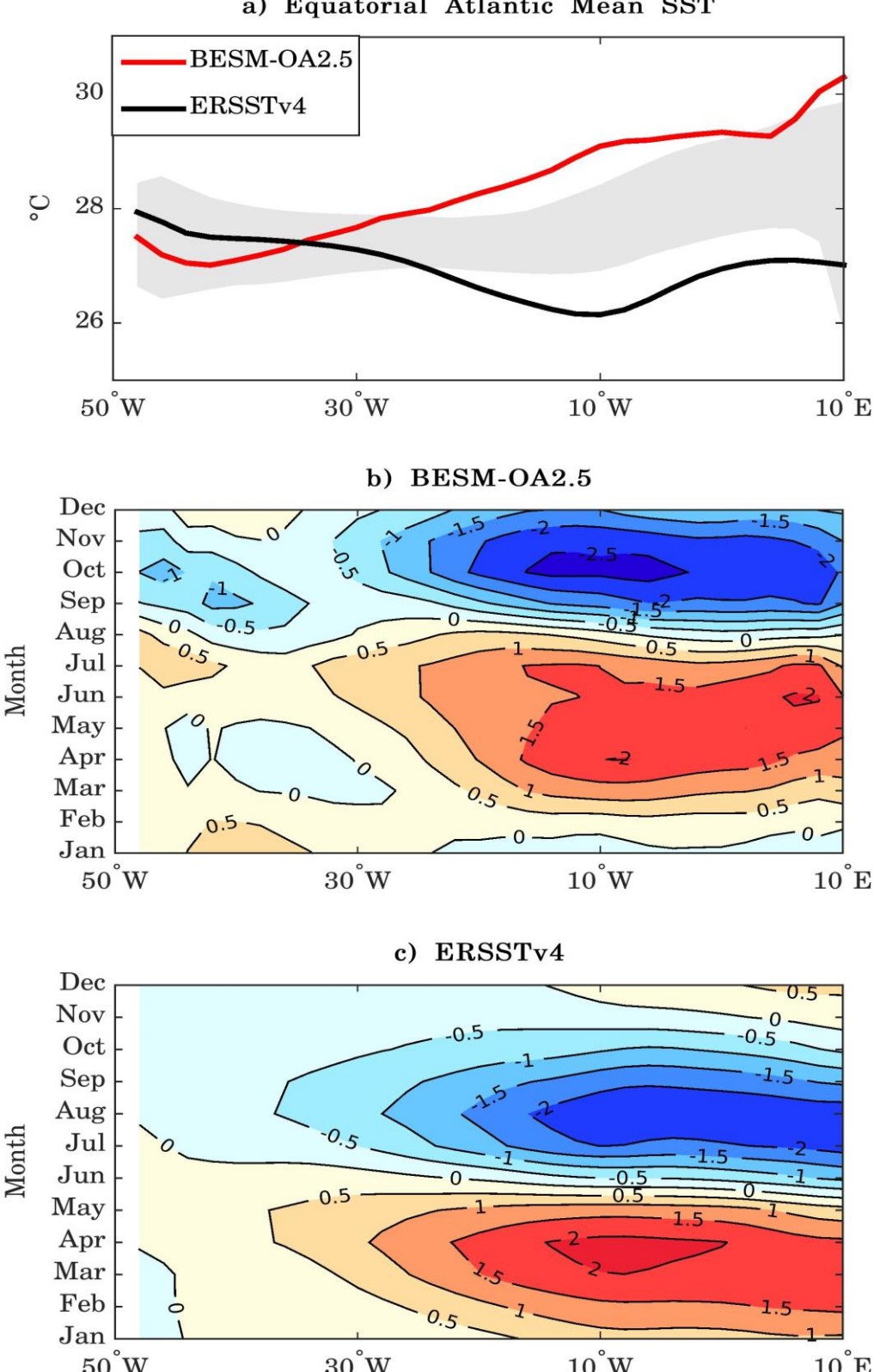

Figure 12 – As Fig. 11, but for the Atlantic Ocean.

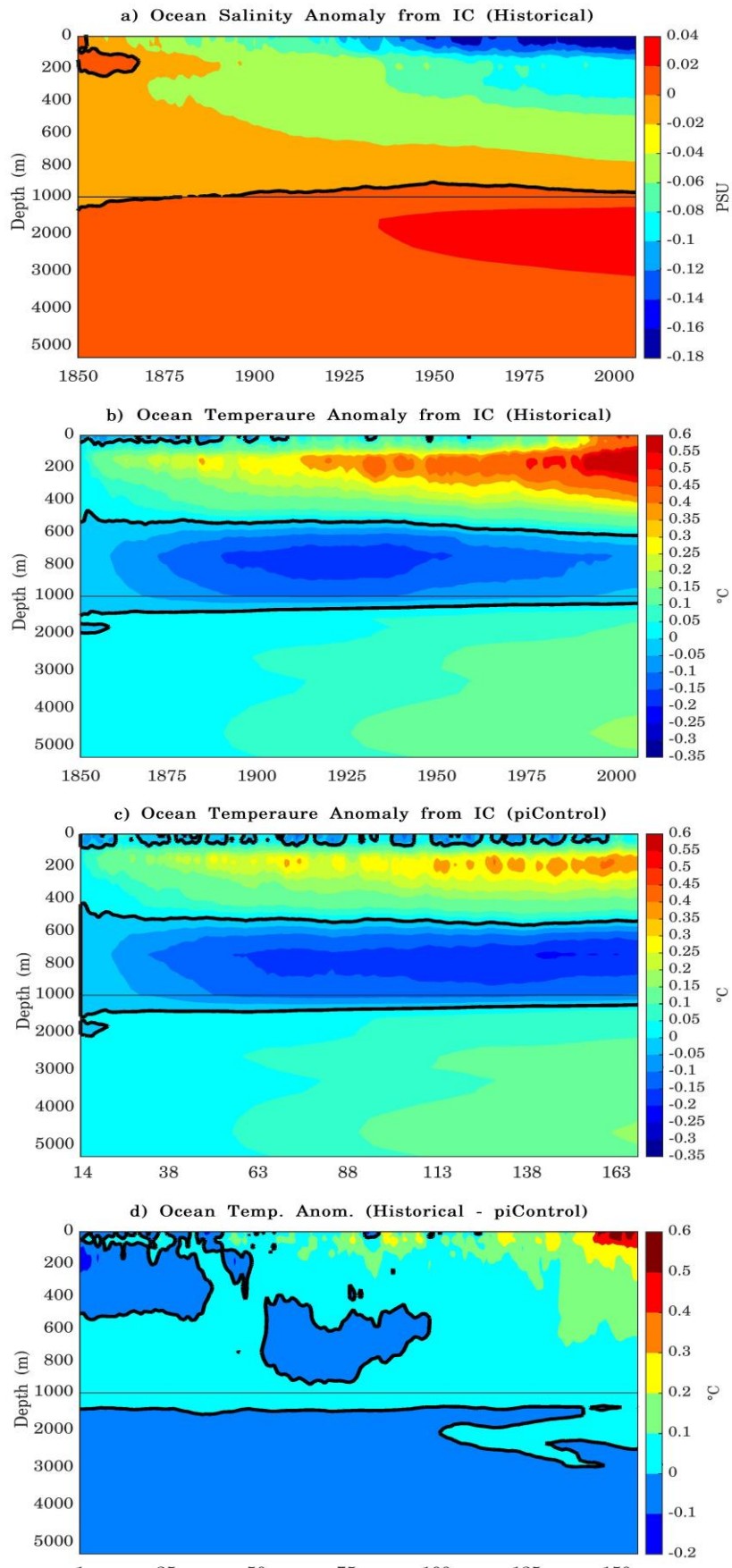

Figure 13 – Depth-time Hovmöller diagrams of the global average ocean (a) salinity
and (b) temperature anomalies from the respective initial conditions (IC). Here, the
initial conditions were taken from the first year for (a, b) historical simulation and from
the 14[th] year for the (c) piControl simulation. The map shown in (d) presents the
difference between the temperature anomalies of the historical simulation relative to the
piControl. The diagrams are based on annual average time series simulated by the
historical simulation over the period 1850–2005 (156 years) and by the piControl
simulation over the period 14–169 years (156 years). The thick black line represents the
zero contours. Note that the vertical scales are different above and below 1000 m.

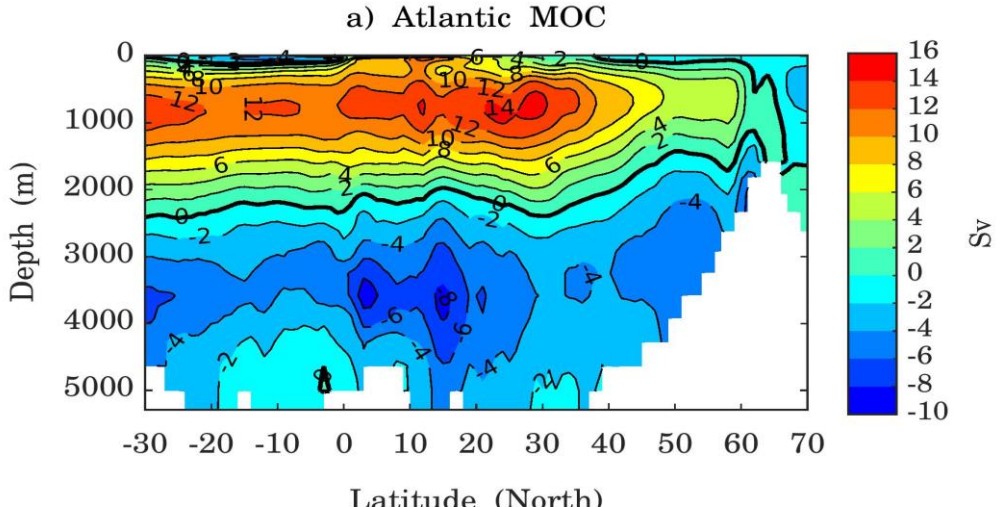

a) Atlantic MOC

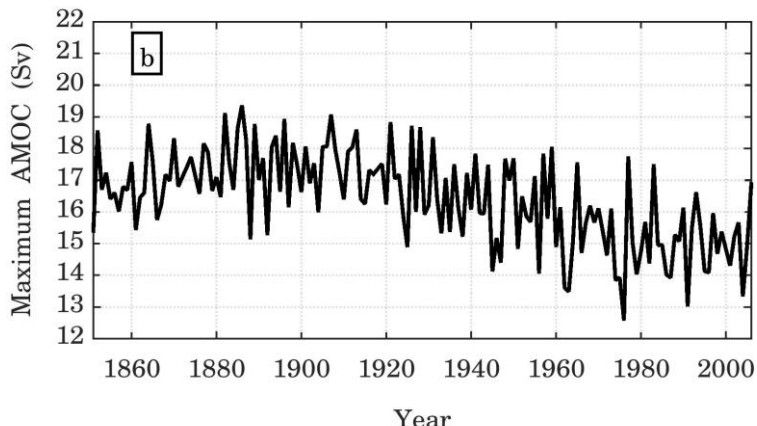

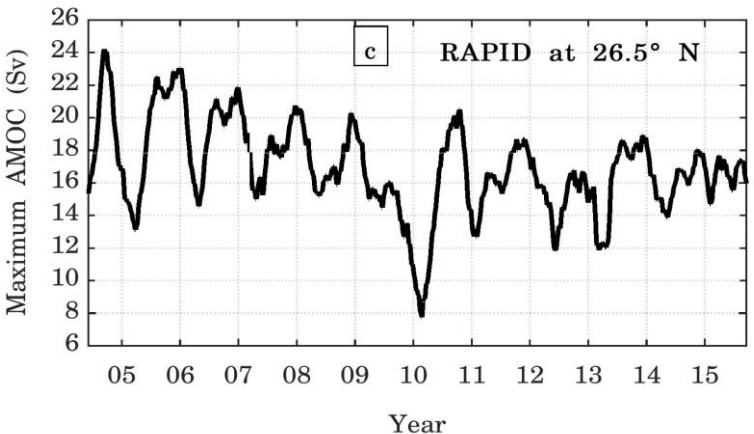

Figure 14 – (a) The Atlantic Meridional Overturning Circulation averaged for the period 1971–2000 and (b) the annual mean maximum AMOC strength time series at latitude 30 °N simulated by BESM-OA2.5 for historical simulation over the period 1850–2005. (c) The graph shows the AMOC time series measured by the RAPID project at 26.5 °N over the period April/2004 to October/2015. The RAPID time series is smoothed by a 3-month running average. Units are in Sverdrup.

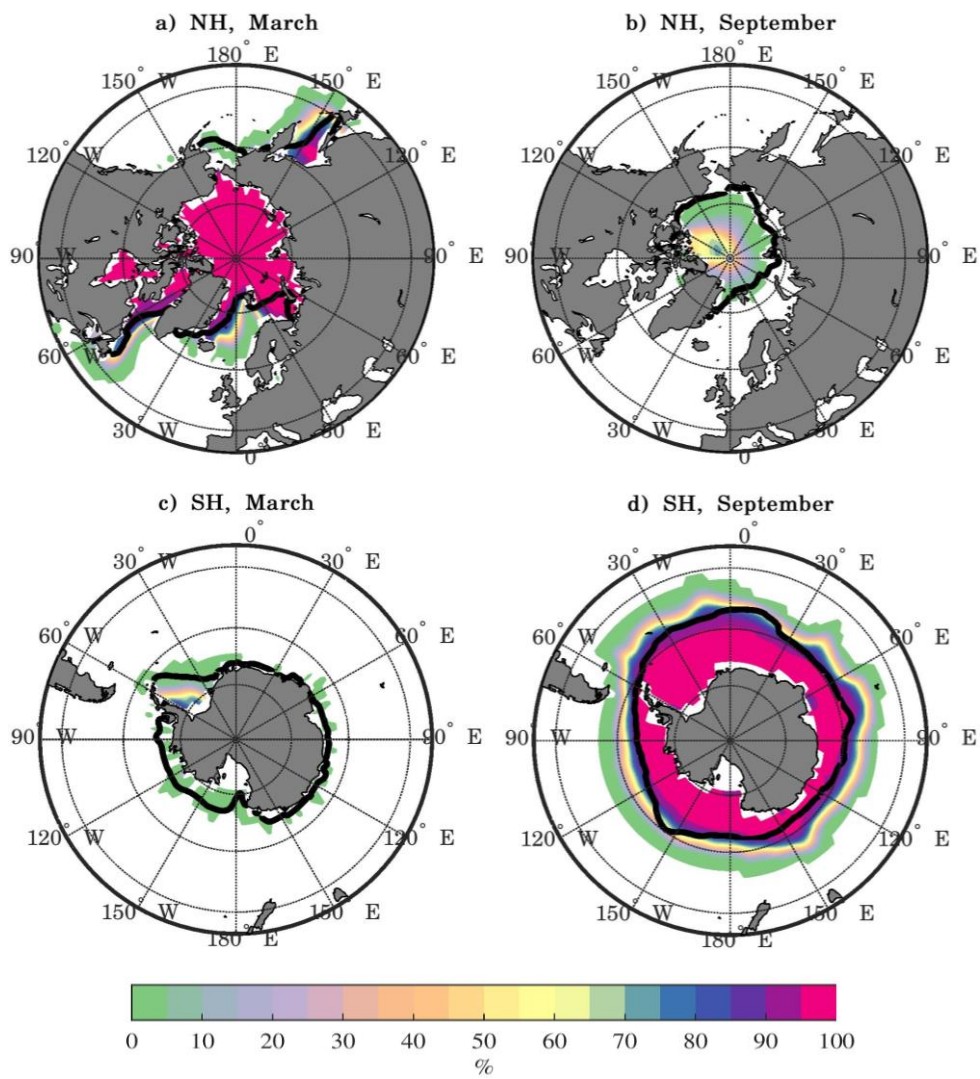

Figure 15 – BESM-OA2.5 mean sea ice concentrations for March (a, c) and September

(b, d) for each hemisphere. The solid black lines show the 15 % mean sea ice

concentration from the 20CRv2 Reanalysis. The average values were computed over the

period 1971–2000 for BESM-OA2.5 and 20CRv2. The concentrations are presented as

percentages.

### a) Pacific SST EOF1 (17.9%) BESM-OA2.5

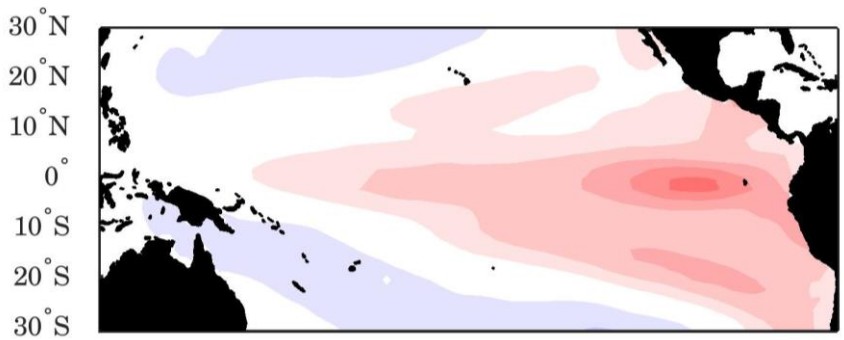

### b) Pacific SST EOF1 (45.0%) ERSSTv4

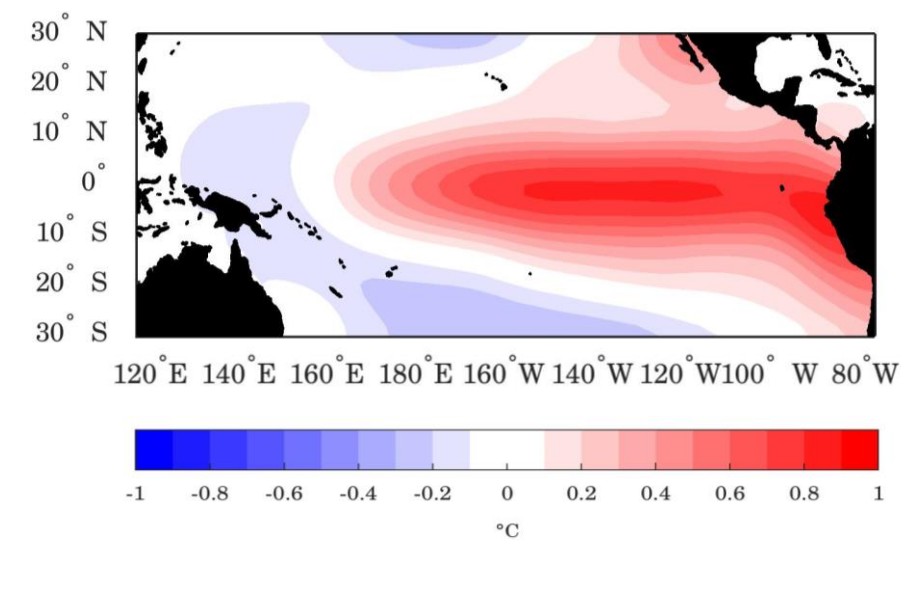

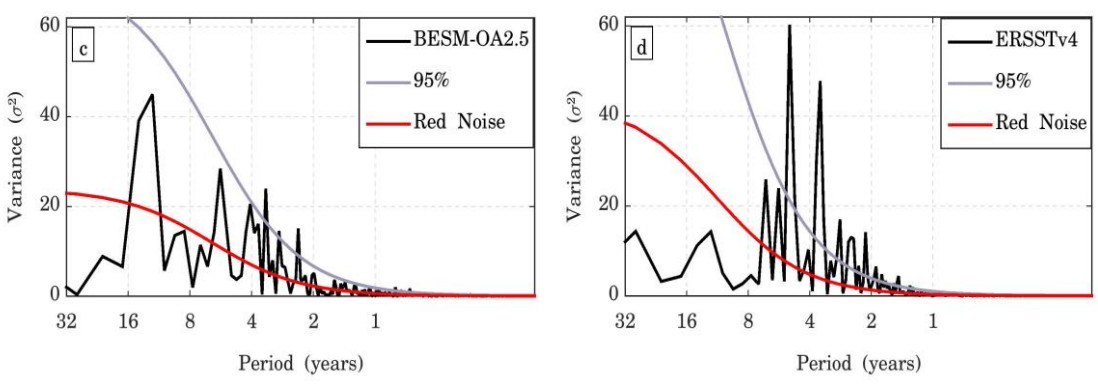

Figure 16 – The leading EOF modes of the detrended monthly SST anomalies over the

Tropical Pacific region (30 ° S–30 ° N; 240 °–70 ° W) for (a) BESM-OA2.5 and (b)

ERSSTv4. The results are shown as the SST anomalies regressed onto the
corresponding normalized PC time series (℃ per standard deviation) over the period
1950–2005. The percentages of the variance explained by each EOF are indicated in the
titles of the figures. The contour interval is 0.1 ℃. Figures (c) and (d) show the power
spectra of the leading joint PC time series of the patterns for BESM-OA2.5 and
ERSSTv4, respectively. The solid red line represents the theoretical red noise spectrum
and the gray line represents the 95 % confidence level.

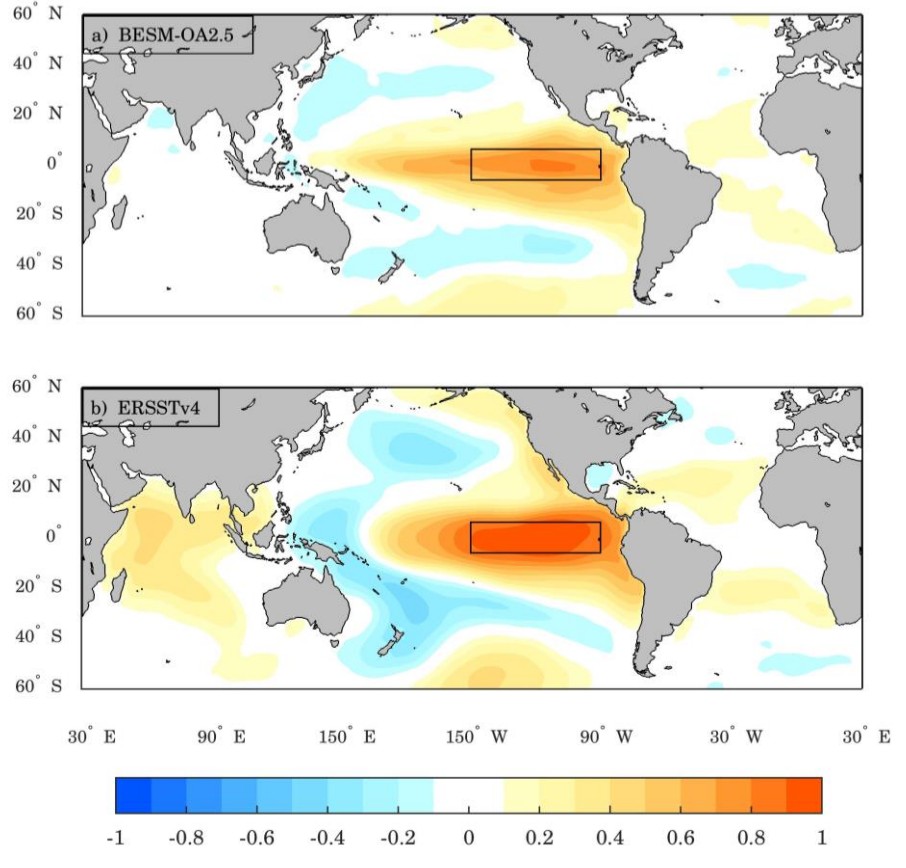

Figure 17 – Spatial maps with the monthly correlations between the Niño-3 index and
the global SST anomalies computed by (a) BESM-OA2.5 and (b) ERSSTv4 over the
period 1900–2005. The anomalies were obtained by subtracting the monthly means for
the entire detrended time series at each grid point. Black rectangles show the Niño-3
index region. Shaded areas are statistically significant at the 95 % confidence level
(based on two tailed Student's t-tests).

## a) AMM jEOF1 (10.7%) BESM-OA2.5

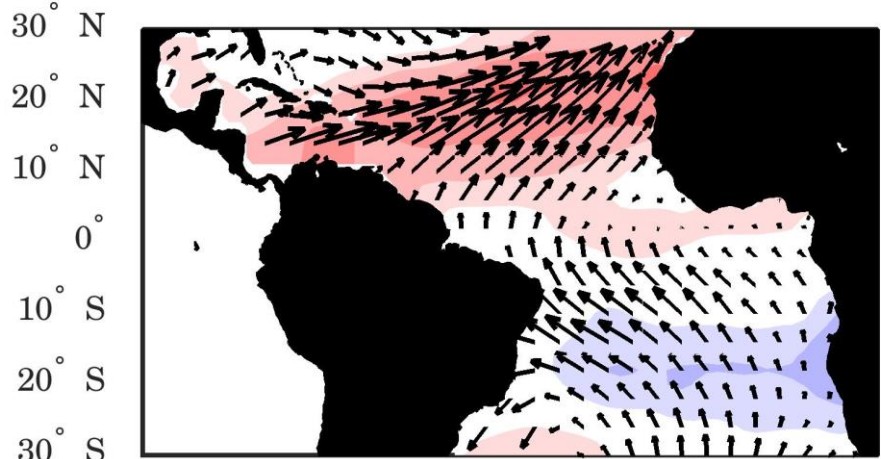

## b) AMM jEOF1 (11.8%) ERSSTv4 (SST), 20CRv2 (Taux,Tauy)

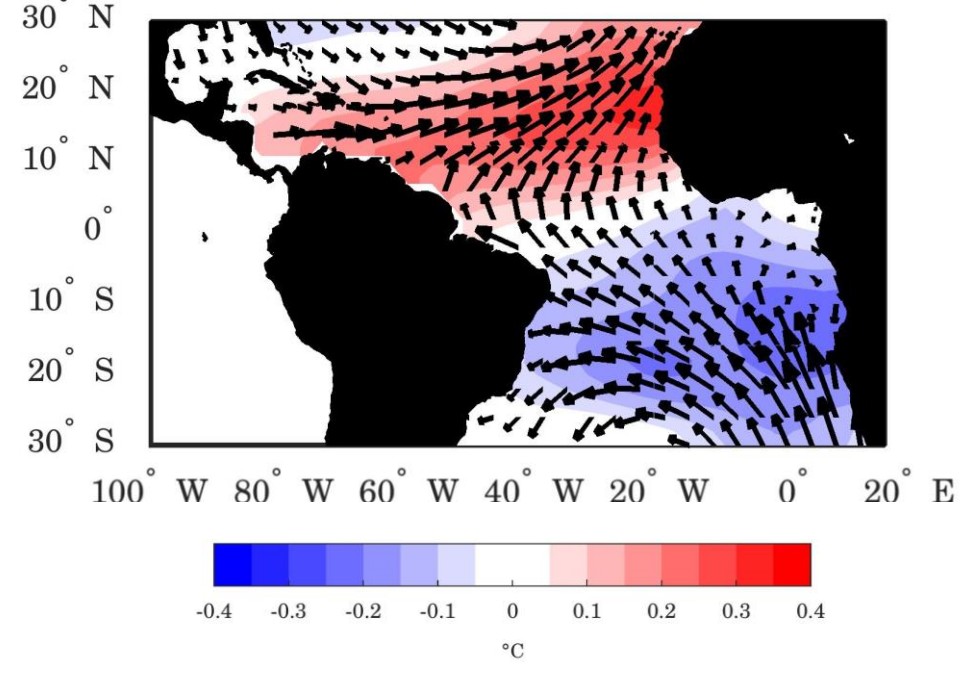

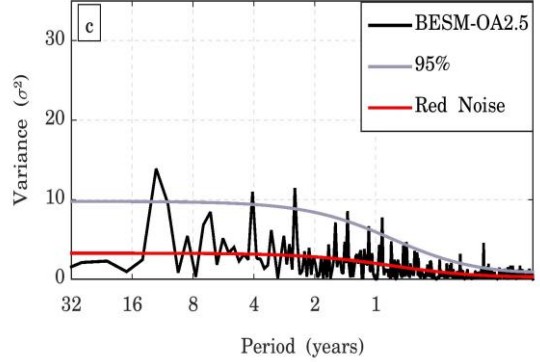

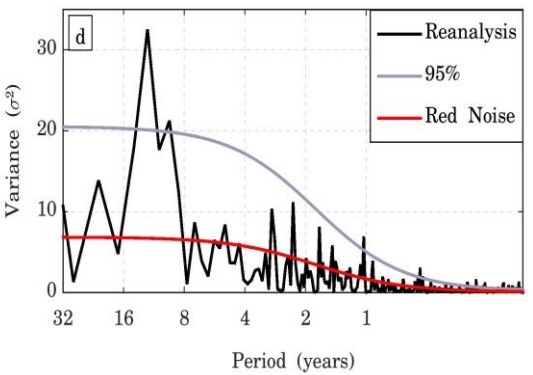

Figure 18 – The leading joint EOF modes of the detrended monthly SST and wind stress
(Taux and Tauy) anomalies for the Tropical Atlantic region (30 °S–30 °N; 100 °W–20 °
E) based on (a) BESM-OA2.5 and (b) from observation (ERSSTv4 and 20CRv2
Reanalysis). The results are shown as the SST anomalies regressed onto the
corresponding normalized PC time series ( °C per standard deviation) and wind stress
anomalies regressed onto the corresponding normalized PC time series (ms$^{-1}$ per
standard deviation) over the period 1950–2005. The percentages of the variance
explained by each EOF are indicated in the titles of the figures. The contour interval is
0.05 °C. Figures (c) and (d) show the power spectra of the leading joint PC time series
of the AMM pattern simulated by BESM-OA2.5 and based onReanalysis, respectively.
The solid red line represents the theoretical red noise spectrum and the gray line
represents the 95 % confidence level.

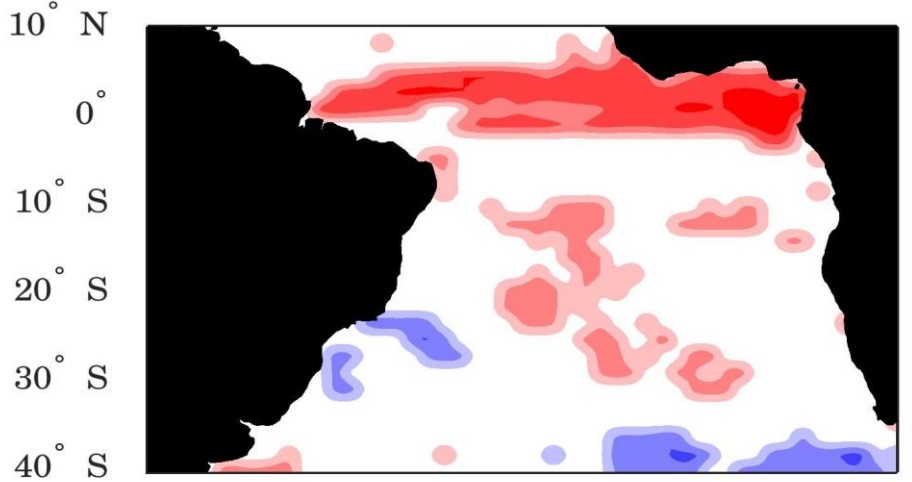

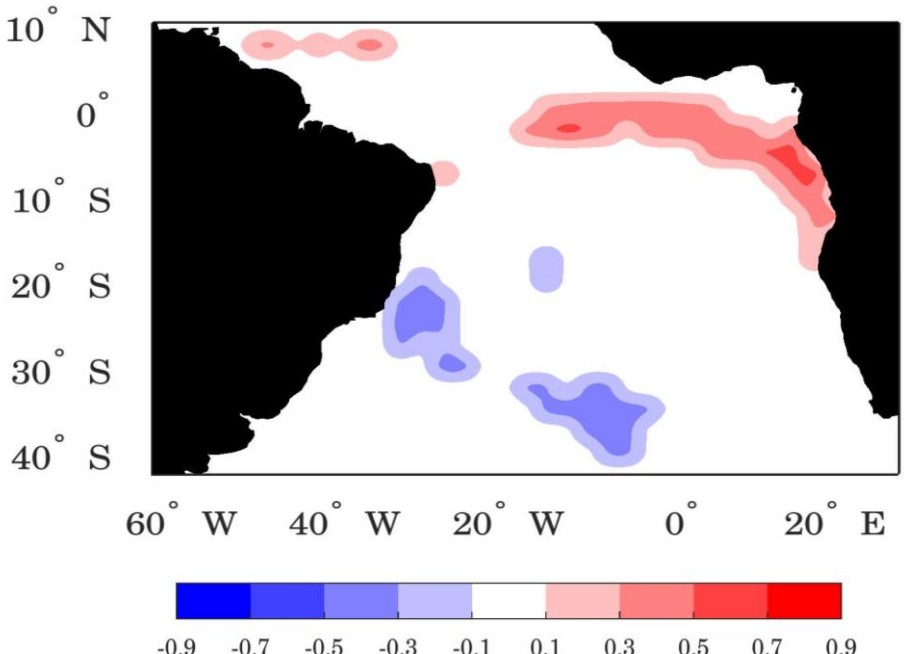

Figure 19 – Spatial maps with the correlation between SST and precipitation (seasonal average DJF) over the South Ocean (40 ° S−10 ° N; 70 ° W−20 ° E) computed by (a) BESM-OA2.5 over the period 1971−2002 and (b) based on Reanalysis over the period 1979−2010. Shaded areas are statistically significant at the 95 % confidence level (based on two tailed Student's t-tests).

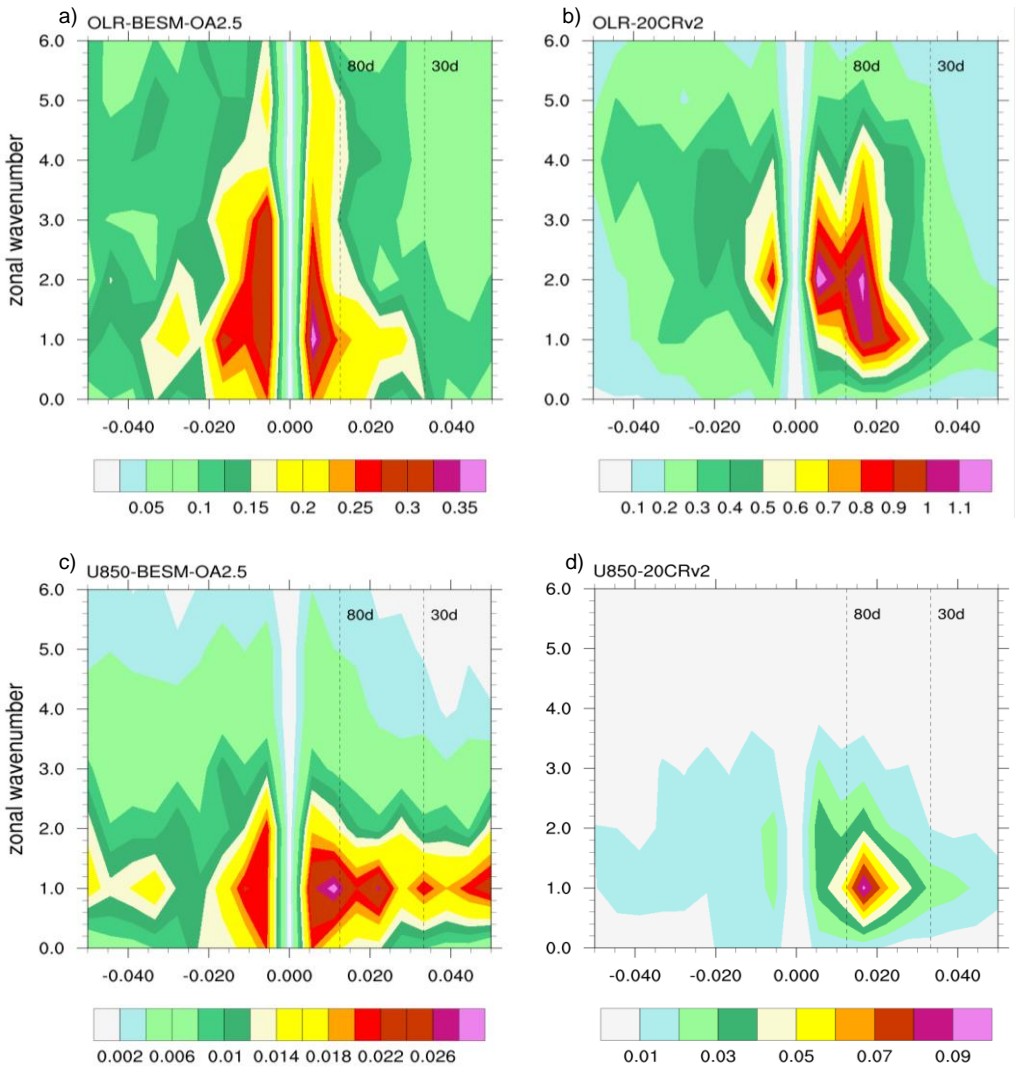

Figure 20 – Wavenumber-frequency power spectra of the tropical (10 ° S–10 ° N) averaged daily outgoing long-wave radiation (OLR) for (a) BESM-OA2.5 and (b) 20CRv2, respectively, and the averaged daily zonal wind component at 850 hPa pressure level (U850) for (c) BESM-OA2.5 and (d) 20CRv2, respectively. The data used were the daily anomalies for the boreal winter (Nov-Apr) over the period 1971–2000. The daily anomalies were obtained by subtracting the climatological daily mean calculated over the period 1971–2000. Individual spectra were calculated for each

1    boreal winter and then averaged over the time period used. Units for the zonal wind

2    (OLR) are $m^{-2}\,s^{-2}$ ($W\,m^2\,s^{-1}$) per frequency interval per wavenumber interval.

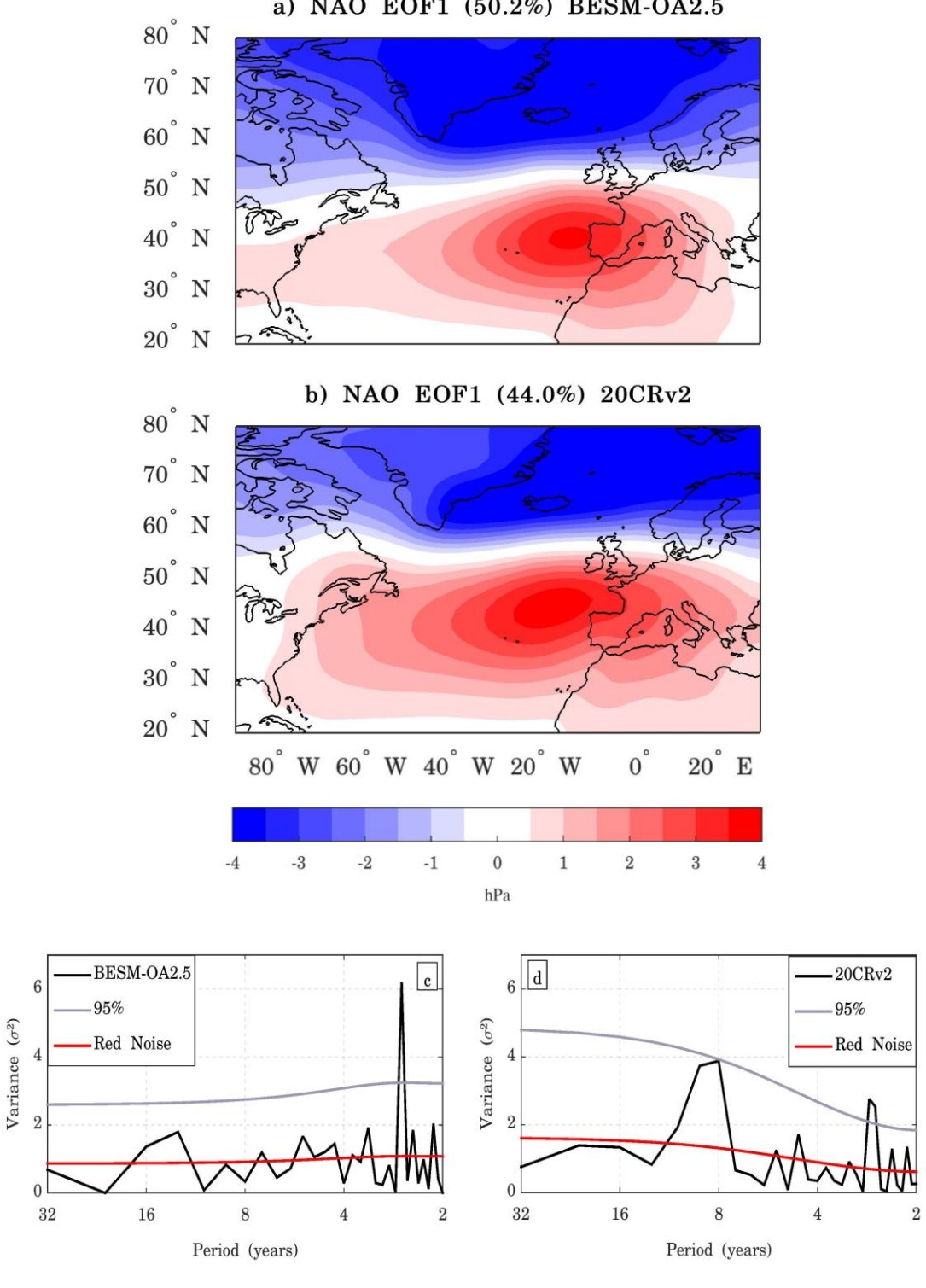

Figure 21 – The leading EOF modes of the boreal winter (DJF) seasonal averaged SLP

anomalies for the Euro-Atlantic region (20 °–80 ° N; 100 ° W−30 ° E) for (a) BESM-

1 OA2.5 and (b) 20CRv2. The results are shown as the SLP anomalies regressed onto the

2 corresponding normalized PC time series (hPa per standard deviation) for the period

3 1950−2005. The percentages of the variance explained by each EOF are indicated in the

4 titles of the figures. The contour interval is 0.5 hPa. Figures (c) and (d) show the power

5 spectra of the leading PC time series of the NAO pattern for BESM-OA2.5 and 20CRv2

6 Reanalysis, respectively. The solid red line represents the theoretical red noise spectrum

7 and the gray line represents the 95 % confidence level.

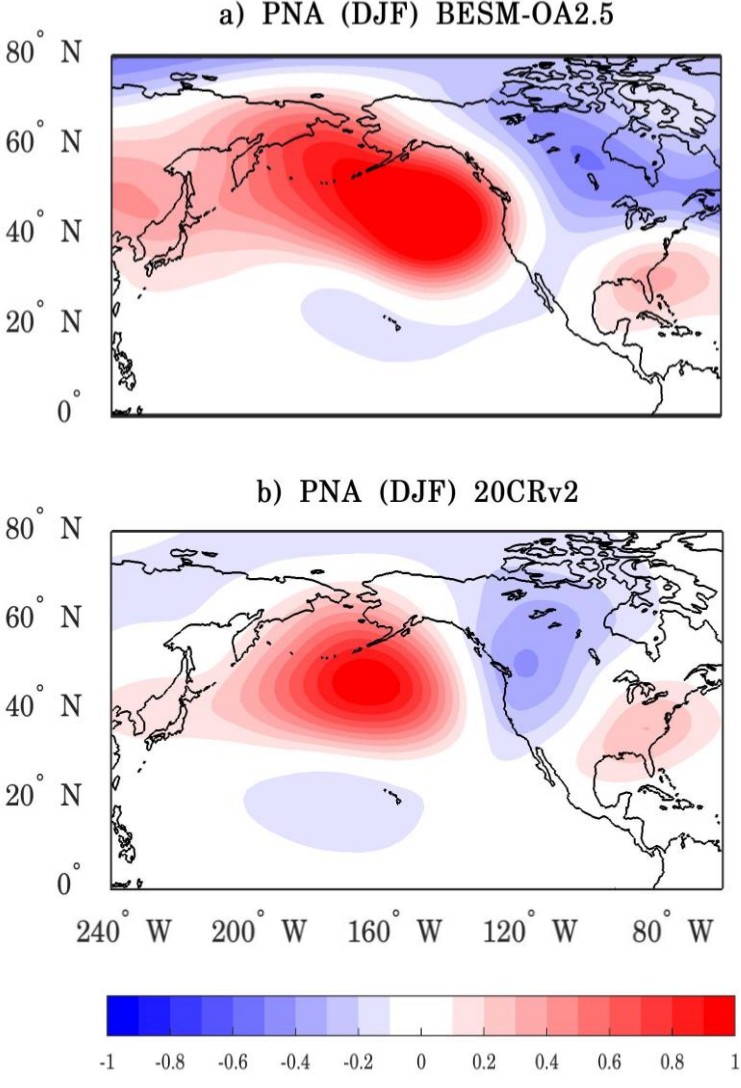

Figure 22 – One-point correlation maps for (a) BESM-OA2.5 and (b) 20CRv2
Reanalysis showing the correlation coefficient of 500 hPa geopotential height based at
45 ° N, 165 ° W and the other grid points. The time series used were from the boreal
winter seasonal (DJF) averaged dataset for the period 1950−2005.

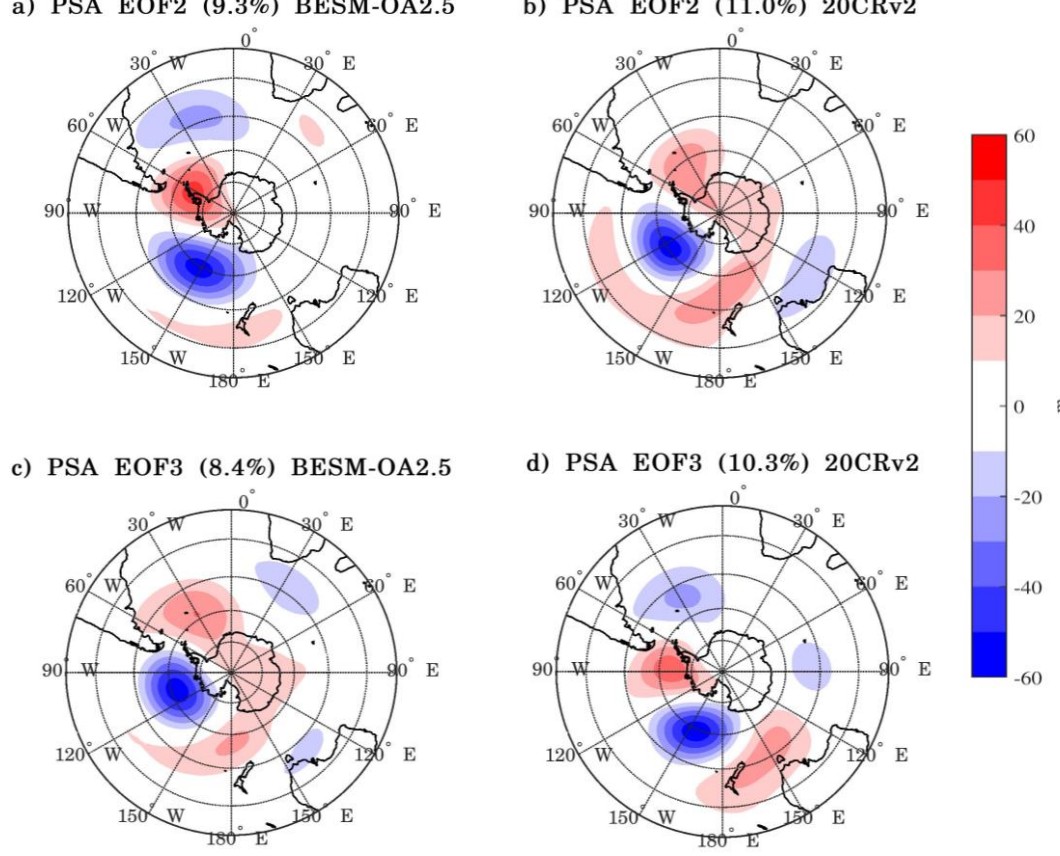

Figure 23 – (a) The second and third EOF modes of the monthly mean 500 hPa
geopotential height field for the Southern Hemisphere (20 °−90 ° S) for BESM-OA2.5
(b) and 20CRv2 Reanalysis. The results are shown as the 500 hPa geopotential height
regressed onto the corresponding normalized PC time series (meters per standard
deviation) over the period 1950−2005. The percentages of the variance explained by
each EOF are indicated in the titles of the figures. The contour interval is 10 m.

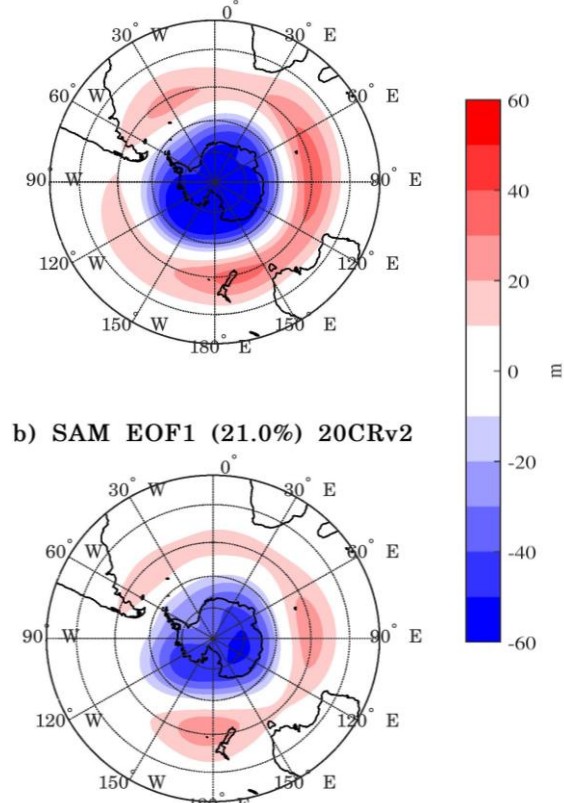

Figure 24 – The leading EOF modes of the monthly mean 500 hPa geopotential height

field for the Southern Hemisphere (20 °–90 ° S) for (a) BESM-OA2.5 and (b) 20CRv2

Reanalysis. The results are shown as the 500 hPa geopotential height regressed onto the

corresponding normalized PC time series (meters per standard deviation) over the

period 1950–2005. The percentages of the variance explained by each EOF are

indicated in the titles of the figures. The contour interval is 10 m.

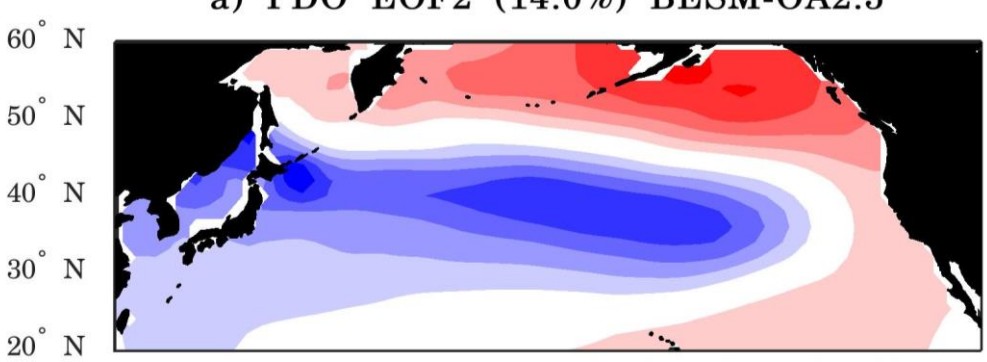

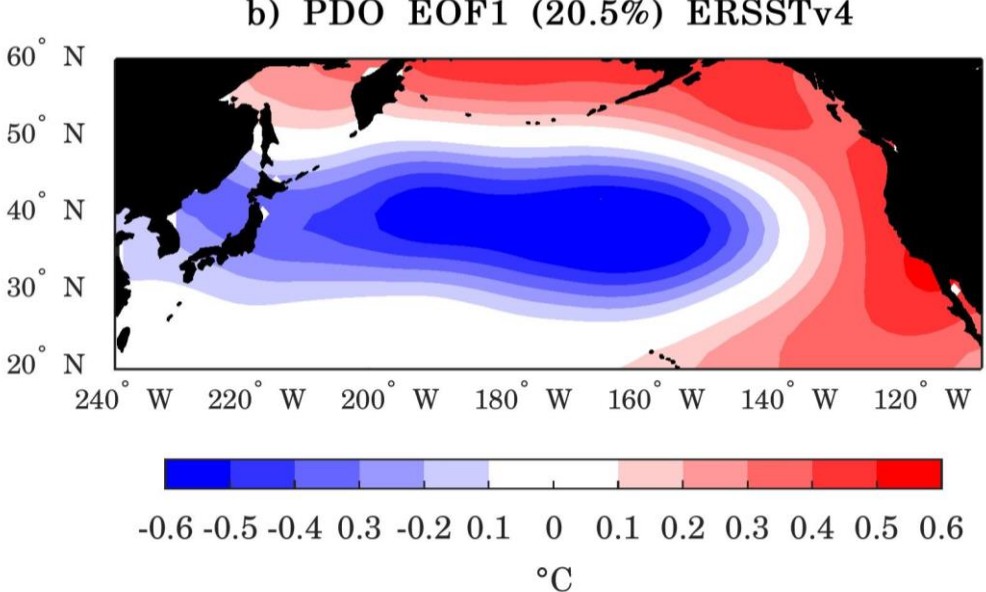

Figure 25 – (a) The second EOF modes of monthly SST anomalies of BESM-OA2.5 and (b) the leading EOF mode of monthly SST anomalies of ERSSTv4, both over the North Pacific Ocean (20 °–60 °N; 240 °–110 °W). The results are shown as the monthly SST anomalies regressed onto the corresponding normalized PC time series (℃ per standard deviation) over the period 1900−2005. The percentages of the variance explained by each EOF are indicated in the titles of the figures. The contour interval is 0.1 ℃.

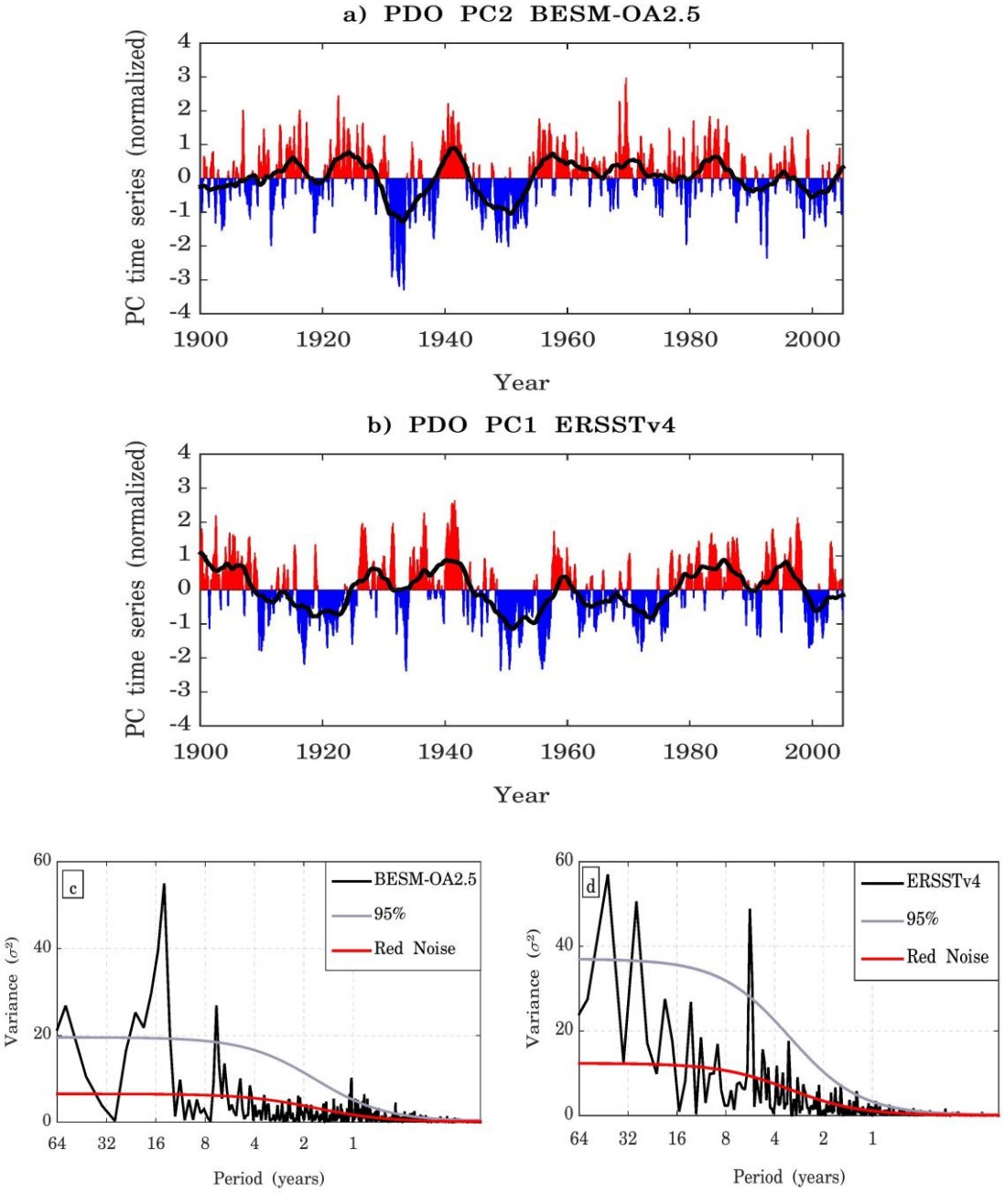

Figure 26 – Normalized second PC time series for (a) BESM-OA2.5 and normalized leading PC time series for (b) ERSSTv4 over the period 1900–2005. The solid black lines show the 5-year running average. Figures (c) and (d) show the power spectra of the second PC time series for BESM-OA2.5 and for the leading PC time series for ERSSTv4, respectively. The solid red line represents the theoretical red noise spectrum and the gray line represents the 95 % confidence level.

| Institute | Model | Simulation | Horizontal resolution (lat×lon) | |
|---|---|---|---|---|
| | | | Atmosphere | Ocean |
| Commonwealth Scientific and Industrial Research Organisation/Bureau of Meteorology (Australia) | ACCESS1.3 | Historical GHG r3i1p1 | 1.25°×1.875° | 300×360 (tripolar) |
| Canadian Centre for Climate Modelling and Analysis (Canada) | CanESM2 | Historical GHG r1i1p1 | 2.7906°×2.8125° | 0.9303°; 1.1407°×1.40625° |
| National Center for Atmospheric Research (USA) | CCSM4 | Historical GHG r1i1p1 | 0.9424°×1.25° | 384×320 (tripolar) |
| Centre National de Recherches Météorologiques/Centre Européen de Recherche et de Formation Avancée en Calcul Scientifique (France) | CNRM-CM5 | Historical GHG r1i1p1 | 1.4008°×1.40625° | 292×362 (tripolar) |
| Geophysical Fluid Dynamics Laboratory (USA) | GFDL-ESM2M | Historical GHG r3i1p1 | 2.0225°×2.5° | 0.3344°; 1°×1° |
| Goddard Institute for Space Studies (USA) | GISS-E2-H | Historical GHG r1i1p1 | 2°×2.5° | 1°×1° |
| Met Office Hadley Centre (UK) | HadGEM2-ES | Historical GHG r1i1p1 | 1.25°×1.875° | 0.3396°; 1°×1° |
| L'Institut Pierre-Simon Laplace (France) | IPSL-CM5A-MR | Historical GHG r1i1p2 | 1.2676°×2.5° | 149×182 (tripolar) |
| Japan Agency for Marine-Earth Science and Technology, Atmosphere and Ocean Research Institute (The University of Tokyo), and National Institute for Environmental Studies (Japan) | MIROC-ESM | Historical GHG r1i1p1 | 2.7906°×2.8125° | 0.5582°; 1.7111°×1.40625° |
| Meteorological Research Institute (Japan) | MRI-CGCM3 | Historical GHG r1i1p1 | 1.12148°×1.125° | 0.5°; 0.5°×1° |
| Bjerknes Centre for Climate Research and Norwegian Meteorological Institute (Norway) | NorESM1-M | Historical GHG r1i1p1 | 1.8947°×2.5° | 384×320 (tripolar) |

2  Table 1 - List of the models from CMIP5 with historical GHG simulations used for the

3  comparison with BESM-OA2.5. Models with higher resolution in the tropical region

1 and decreasing resolution towards the poles have two values for latitude in their

2 respective oceanic resolution columns. For models with oceanic tripolar grids, the

3 number of grid points in each coordinate are given.

