# Peer review of "The Brazilian Earth System Model version 2.5: Evaluation of"

_Geoscientific Model Development, 2018_

## Short Comment (SC1) · 9 Jul 2018

Dear Authors,

please provide additional information on how to obtain the model code. Please at least provide an email-Adress and / or a web site where the licence agreement and contact information can be found.

Best , Astrid Kerkweg
* * *

---

## Author Comment (AC1) · 11 Jul 2018

Dear GMD community,

in order to obtain the source code and data of BESM-OA2.5, please contact Paulo Nobre (paulo.nobre@cptec.inpe.br).

Best regards,

Sandro Veiga
* * *

---

## Referee Comment (RC1) · Anonymous Referee #1 · 20 Jul 2018

The manuscript is a document about Brazilian Earth System Model version 2.5, which is designed for the coming CMIP6 coordinated experiments. Authors comprehensively evaluated the major climatology and climate variabilities through CMIP5 simulation configurations. Considering the importance of the model document, I think the manuscript should be eventually accepted.However, I have to nonetheless point out there is still much room for improvement. My comments are put as below.

1. The model BESM-OA2.5 is an Earth System Model. However, there is no evaluation of model about cloud-aerosol-chemistry or dynamic carbon cycle processes. The unique characters highly related with Earth System Model should been introduced, and some results should been also evaluated.

2. The bias of the double ITCZ is taken as the long standing problems from CMIP3 to

CMIP5. As for CMIP6, some model groups illustrated the encouraging progresses to mitigation these biases. How about BESM-OA2.5? Annual mean SST and precipitation should be also given.

3. As a documentation of BESM-OA2.5, the behaviors of the subseaonal variability (e.g. MJO) and diurnal cycle are highly suggested to be given.

4. Fig5: there are some uncertainty raised from the observation/reanalysis products, therefore, CMAP or the last global precipitation datasets of stations should be added to present the differences.

5. The plans of BESM-OA2.5 for CMIP6 participation are also highly expected.

―――――――――――――――――――――

---

## Referee Comment (RC2) · Anonymous Referee #2 · 25 Jul 2018

The paper documents the present state of a new version of the Brazialian Earth System Model (BESM) that is supposed to be used in the CMIP6 model intercomparison. The authors describe mean state and variability characteristics comparing the simulated results from 20th century observations and reanalyses. They describe the improvement in relation to an earlier version of BESM and highlight some aspects that are of particular importance for South American climate. The general content and aspects covered are well designed and go along with other description papers published for similar models. However, the paper is not mature enough to be published and needs major revisions.

There are some general issues that need to be addressed:

1. There is no description of the pre-industrial control run. In particular, the coupled

model simulation presented here is done after an extremely short spin-up phase (just 13 years after one cycle of CORE forcing!). Experience tells us that models tend to drift after changing from stand-alone conditions to fully coupled. BESM could be an exception, but it could be that many features seen in this paper are due to incomplete adaptation. It should be documented how the 3d ocean temperature and salinity fields evolve. How does the integrated ocean-atmosphere heat flux evolve? Is there any energy imbalance that could (partly) explain the weak warming in the 20th historical simulation? 2. A documentation of a newly developed couple model should include an estimate of climate sensitivity. The analyses for that should be standard procedure. 3. The discussion of the evaluation with observations/reanalyses is descriptive, but does most often not discuss if the biases are acceptable, larger/smaller than in other models, or which consequences come with them. For example, if we assume that including aerosols into the BESM influences the historical simulation in a similar way as in other models (Figure 2), BESM would severely underestimate global warming in the last century. This should be a matter of concern and lead the authors to look for the origin of this discrepancy. Or is the plan that everything will be better in the next generation of BESM,as somehow implied in the conclusions? 4. The quality of the figures should be improved. At least in my pdf version I could hardly decipher axis and contour labels

Minor comments: Abstract, ln 6: "validation" would mean that you have some measures for when a model is valid, better use "evaluation"

Page 3, ln 6 and following text. The authors say that for an ESM there needs to be an interactive biogeochemical module. But it is not explained later, if or what kind of biogeochemistry model is included and if there are other publications planned on this aspect.

Page 4, ln 1: do you mean interactive aerosols and chemistry or just the ability to use them as read-in fields?

[Figure]

Ln 11: "seamless predictions" is it really used for weather predictions (at which resolutions?) and do you have to apply problem-specific parameterizations?

Page 6, ln 18ff. How does the system behave after switching from CORE to fully coupled. I doubt that the spin-up is long enough.

Page 15: SSTs seem to be generally too warm. Is that also true for surface air temperature in the control run or in the beginning of the historical? Wouldn't that call for some tuning exercise, e.g. looking into cloud parameterizations?

Page 17, ln 21ff. How does AMOC look in the control run, any drift?

---

## Author Comment (AC2) · 30 Oct 2018

We thank the valuable comments, whose responses follow:

**1. The model BESM-OA2.5 is an Earth System Model. However, there is no evaluation of model about cloud-aerosol-chemistry or dynamic carbon cycle processes. The unique characters highly related with Earth System Model should been introduced, and some results should been also evaluated.**

Reply:

The model used in this work is an ocean-atmosphere-biosphere coupled model, indicated in its name BESM-OA version 2.5. Therefore, it is not at this version a full Earth System Model, as disclaimed in the manuscript (Page 3 Lines L19-L23). We acknowledge that an Earth System Model is a comprehensive model that includes all Earth system components, including biogeochemical cycles (e.g. dynamic carbon cycle processes) and cloud-aerosol-chemistry processes. Although the main aim of our group is to build up such a model, at the moment the model is a coupled ocean-atmosphere climate model, without the representation of either biogeochemical cycles or cloud-aerosol-chemistry processes. Nonetheless, the name Brazilian Earth System Model was chosen in order to avoid a future change of the model's name on its transition from an ocean-atmosphere coupled model to an Earth System model. For this reason, in the acronym BESM was always added the letters OA, which stand for ocean-atmosphere coupled model. Therefore, at the moment, there is any evaluation of the biogeochemical cycles and their interaction with other Earth system components.

**2. The bias of the double ITCZ is taken as the long standing problems from CMIP3 to CMIP5. As for CMIP6, some model groups illustrated the encouraging progresses to mitigation these biases. How about BESM-OA2.5? Annual mean SST and precipitation should be also given.**

Reply:

As well as other global coupled climate model groups, the BESM development group is working on understanding the causes for the recurrent double ITCZ problem in global coupled models. From the research on cloud physics parameterization we are currently working on, we have detected that the double ITCZ coupled problems emerges in part due to incorrect distribution of upper air sources of mass along the equator. This research is in progress, and part of its results was submitted to Climate Dynamics (Bottino and Nobre, pers. comm.) and is currently under review.

As you have suggested, the figures for the mean fields of SST and precipitation for BESM-OA2.5 and Reanalysis have been included in the manuscript. The annual mean precipitation figure is included on Page 56 and the annual mean SST is included on Page 64.

**3. As a documentation of BESM-OA2.5, the behaviors of the subseasonal variability (e.g. MJO) and diurnal cycle are highly suggested to be given.**

Reply:

All outputs of the simulation were previously defined to be given in daily time scale, therefore it would not be possible to analyze the diurnal cycle. In the case of subseasonal variability, the Madden-Julian Oscillation (MJO) is computed and added to the revised manuscript as a topic of Tropical Variability subchapter (Page 25-27 Lines L16-L2). The MJO figure is presented on Page 80.

**4. Fig5: there are some uncertainties raised from the observation/reanalysis products, therefore, CMAP or the last global precipitation datasets of stations should be added to present the differences.**

Reply:

Thank you for your advice. The difference between BESM-OA2.5 and the CMAP precipitation reanalysis has been computed and is shown below (Fig. 1). The global model's mean biases are similar for GPCP (0.3 mm day$^{-1}$) and CMAP (0.4 mm day$^{-1}$). In the case of the global model's RMSE biases, they are also similar for GPCP (1.4 mm day$^{-1}$) and CMAP (1.5 mm day$^{-1}$). This description has been added to the revised manuscript (Page 13, Lines L13-L16).

[Figure]

Figure 1 – Spatial map of annual mean precipitation bias of BESM-OA2.5 relative to CMAP. The averages values are computed over the periods 1971–2000 and 1979–2008, for BESM-OA2.5 and CMAP, respectively. Units are in mm day$^{-1}$.

**5. The plans of BESM-OA2.5 for CMIP6 participation are also highly expected.**

Reply:

The plans for the BESM-OA2.8 CMIP6 participation consists, in addition to the DEC characterization runs, the Scenario Model Intercomparison Project (ScenarioMIP;

O'Neill et al., 2016), Decadal Climate Prediction Project (DCPP; Boer et al., 2016), and, supercomputer time available at CPTEC, the High Resolution Model Intercomparison Project (HighResMIPv1.0; Haarsma et al., 2016).

Boer, G. J., Smith, D. M., Cassou, C., Doblas-Reyes, F., Danabasoglu, G., Kirtman, B., Kushnir, Y., Kimoto, M., Meehl, G. A., Msadek, R., Mueller, W. A., Taylor, K. E., Zwiers, F., Rixen, M., Ruprich-Robert, Y., and Eade, R.: The Decadal Climate Prediction Project (DCPP) contribution to CMIP6, Geosci. Model Dev., 9, 3751-3777, https://doi.org/10.5194/gmd-9-3751-2016, 2016.

Haarsma, R. J., Roberts, M. J., Vidale, P. L., Senior, C. A., Bellucci, A., Bao, Q., Chang, P., Corti, S., Fučkar, N. S., Guemas, V., von Hardenberg, J., Hazeleger, W., Kodama, C., Koenigk, T., Leung, L. R., Lu, J., Luo, J.-J., Mao, J., Mizielinski, M. S., Mizuta, R., Nobre, P., Satoh, M., Scoccimarro, E., Semmler, T., Small, J., and von Storch, J.-S.: High Resolution Model Intercomparison Project (HighResMIP v1.0) for CMIP6, Geosci. Model Dev., 9, 4185-4208, https://doi.org/10.5194/gmd-9-4185-2016, 2016.

O'Neill, B. C., Tebaldi, C., van Vuuren, D. P., Eyring, V., Friedlingstein, P., Hurtt, G., Knutti, R., Kriegler, E., Lamarque, J.-F., Lowe, J., Meehl, G. A., Moss, R., Riahi, K., and Sanderson, B. M.: The Scenario Model Intercomparison Project (ScenarioMIP) for CMIP6, Geosci. Model Dev., 9, 3461-3482, https://doi.org/10.5194/gmd-9-3461-2016, 2016.

---

## Author Comment (AC3) · 30 Oct 2018

We thank the valuable comments, whose responses follow:

**Major Revisions:**

**1.1 There is no description of the pre-industrial control run.**

Reply:

Our group has prepared two manuscripts describing describing the results using piControl and abrupt 4 regarding the atmospheric component has been submitted to the GMD and is under review for GMDD (Capistrano et al., 2018). The manuscript describing the results regarding the oceanic component is in preparation and will be submitted soon (Nobre et al. 2018). We have included a summary description of the pre-industrial control run in the revised manuscript (Page 8, Lines L20-L23).

In order to advance the results regarding the analysis for the ocean component of the piControl run, we describe some of the analyses of Nobre et al. (2018) below.

Fig. 1 shows the variation of the global averaged (a) near-surface air temperature (SAT), (b) sea surface temperature (SST), (c) sea surface salinity (SSS), (d) net of the radiation at the top of atmosphere (TOA), (e) net of the ocean/atmosphere heat flux and (f) the maximum strength of the Atlantic Meridional Overturning Circulation (AMOC), over 1000 years of simulation. This period includes the 13 years of coupled ocean-atmosphere spin-up used by the Historical simulation and the remaining 156 years of simulation that the piControl runs in parallel with the Historical simulation. The red dash-dot lines show the linear trend of each time series. With the exception of the SSS, all the variables show a fast adjustment to stable conditions during the first 13 years of the piControl simulation, which is the period used as the spin-up period for Historical simulation. With the exception of the SSS, the piControl simulation maintains stable conditions. Over this stable period, the mean SAT is 13.286 ºC with a linear trend of 0.0091 ºC (100 yr)$^{-1}$ and the mean SST is 19.57 ºC with a linear trend of 0.0084 ºC. Although the SAT and SST trends are statistically significant at 95%

level, they are small and comparable to other state-of-the-art models (e.g. NorESM1-M presents a linear trend of 0.0074 ºC $(100 \text{ yr})^{-1}$ and 0.0062 ºC $(100 \text{ yr})^{-1}$ for SAT and SST, respectively, and NESMv3 presents a linear trend of 0.0021 ºC $(100 \text{ yr})^{-1}$ and 0.0073 ºC $(100 \text{ yr})^{-1}$ for SAT and SST, respectively), therefore indicate stable variation for both variables. The mean SSS is 34.07 PSU with a linear trend of -0.017 PSU $(100 \text{ yr})^{-1}$. The SSS linear trend is significantly higher than simulated by NorESM1-M that has a linear trend of -0.00006 PSU $(100 \text{ yr})^{-1}$ and NESMv3 that has a linear trend of -0.008 PSU $(100 \text{ yr})^{-1}$. However, it should be noticed that there is stabilization after 500 years of simulation, since the linear trend for the first 200 years is -0.09 PSU $(100 \text{ yr})^{-1}$ and for the last 500 is -0.004 PSU $(100 \text{ yr})^{-1}$. This SSS long-term adjustment can be related to the ocean drift.

The net radiation at TOA has a mean value of -4.33 W $m^{-2}$ and a linear trend of -0.0090 W $m^{-2}$ $(100 \text{ yr})^{-1}$. The linear trend is comparable to other state-of-the-art models (e.g. NorESM1-M presents a linear trend of -0.0038 W $m^{-2}$ $(100 \text{ yr})^{-1}$ and NESMv3 presents a linear trend of -0.0041 W $m^{-2}$ $(100 \text{ yr})^{-1}$). The negative average value means that the atmosphere is losing heat to the outer space and can be an explanation for the weak warming observed in the Historical simulation (Fig. 2 (manuscript)). Although the negative bias of the net radiation at TOA is significant, there is an important improvement from the previous version (BESM-OA2.3) that presents a negative net radiation at TOA of roughly -20 W $m^{-2}$ (Marcus Bottino, personal communication, 2018). Nevertheless, the mean value of the net radiation at TOA is still away from zero (optimal value), when compared with NorESM1-M that presents an average value of 0.086 W $m^{-2}$ and NESMv3 that presents an average value of 0.17 W $m^{-2}$. The global net of the ocean/atmosphere heat flux has a mean value of 0.94 W $m^{-2}$, which is slightly away from zero (optimal value) than NorESM1-M (0.122 W $m^{-2}$) and NESMv3 (0.31 W $m^{-2}$, for Earth surface). The net of the ocean/atmosphere heat flux presents a linear trend of -0.018 W $m^{-2}$ $(100 \text{ yr})^{-1}$, which is an order of magnitude higher than NorESM1-M (-0.004 W $m^{-2}$ $(100 \text{ yr})^{-1}$) and NESMv3 (-0.006 $(100 \text{ yr})^{-1}$, for Earth surface). The positive values indicate that the ocean is gaining heat from the atmosphere throughout the simulation, but since it has a negative trend, the ocean heat content is diminishing during the integration. The small trends (although statistically significant at 95%) of the net radiation at TOA and net of the ocean/atmosphere heat flux show that the model has achieved stable

conditions for the heat flux after the initial adjustment. The AMOC has a maximum mean value of 13.38 Sv and a linear trend of -0.11 Sv $(100 \text{ yr})^{-1}$ at 28 ºN. However, it can be noticed that there is a sharp decrease of the AMOC in the period 80-170 years and then its strength recoveries. The average AMOC strength measured by the project RAPID at 26.5 ºN is 17.2 Sv (McCarthy et al., 2015), which is slightly higher than simulated by BESM-OA2.5 piControl. The AMOC strength at 26.5 ºN simulated by NorESM1-M and NESMv3 are ~31 Sv and 14.8 Sv respectively. The linear trend for NorESM1-M and NESMv3 are -0.12 Sv and -0.22 Sv respectively. The AMOC negative linear trend simulated by BESM-OA2.5 is very similar to the linear trends given by NorESM1-M and NESMv3. However, this negative linear trend indicates that the model is still drifting throughout the piControl simulation. Despite the surface quantities (except SSS) and heat flux in the air-sea interface and at TOA indicate stable conditions, the model still drifts in the ocean.

[Figure]

Figure 1 – Annual average time series for the global average near surface air temperature (a), sea surface temperature (b), sea surface salinity (c), net of the radiation at TOA (d; positive values indicates that the atmosphere is warming), net of the ocean/atmosphere heat flux (e; positive values indicates that the ocean is warming) and the maximum AMOC (f; at latitude 28° N and depth of 800 m), simulated by the piControl from the beginning of the simulation up to 1000 years. The red dash-dot lines show the linear trend of each time series, starting from the year 14th of the simulation (vertical black dash-dot lines). All the trends are statically significant at 95% of confidence level.

To further evaluate the piControl ocean drift, it is computed the depth-time Hovmöller diagrams of global mean ocean temperature and salinity departures from their respective initial conditions (Fig. 2). By initial conditions we mean the value of the first year after the model adjusts, in this case, the 14[th] year. The ocean warms in the sub-surface waters, between 150 and 350 m depth, and in deeper waters, from 1500 m up to the ocean floor. At 1000 years of simulation, the higher warming occurs below 4000 m depth, where the waters warm ~0.8-0.9 ℃ relative to the initial values (Fig. 2a). There is a cooling of 0.2-0.3 ℃ between 500-1500 m depth that starts to narrow throughout the simulation up to 600 years. From 600 years of simulation the whole global ocean warms with intermittent signals only on the surface. The ocean salinity slightly increases below 1000 m depth up to the ocean floor and at 1000 years of simulation the increase is ~0.06 PSU comparing with the initial values (Fig. 2b). Above 1000 m depth there is a significant freshening of the ocean waters, with the surface waters salinity decreasing up to 0.28 PSU in 1000 years of simulation when compared with initial values. Such results indicate that the global ocean is still drifting from its initial conditions.

[Figure]

Figure 2 – Depth-time Hovmöller diagrams of global average ocean (a) temperature and (b) salinity anomalies from the respective initial conditions (IC). Here the initial conditions are taken from the 14[th] year, therefore after the 13 initial years of the adjustment. The diagrams are based on annual average time series by the piControl simulation over the period 1000 years. The thick black line represents the zero contours. Note that the vertical scales are different above and below 1000 m,

The linear trends are obtained through a linear regression of each respective time series and the statistical significance is tested through the Mann-Kendall test, considering statistically significant at 95% of confidence level. The number of degrees of freedom in the test is adjusted due to the autocorrelation of the time series.

**1.2 In particular, the coupled model simulation presented here is done after an extremely short spin-up phase (just 13 years after one cycle of CORE forcing!). Experience tells us that models tend to drift after changing from stand-alone conditions to fully coupled. BESM could be an exception, but it could be that many features seen in this paper are due to incomplete adaptation.**

Reply:

We agree with your observations, and the spin-up process is better described below. After the ocean stand-alone run forced with CORE fields (71 years), a spin-up run of the fully coupled model in a previous version is done for 100 years. The atmosphere and ocean state at the end of this 100 years long integration is used as the initial conditions for the piControl run. Moreover, land ice albedo and cloud microphysics were changed from the initial spin-up run (exp178) to the piControl simulation. The ocean model configuration remains identical on both versions. Therefore, regardless of the coupled model having being integrated for 100 years, in this case, it is assumed only as spin-up the first 13 years period of the piControl simulation.

We recognize that long spin-ups are desirable. In fact, the deep layers of the ocean are still adjusting after 1000 years. However, through previous tests, it was observed that the surface layers of the ocean model tends to adjust in the first 13 years for surface variables, as SAT, SST and net radiation at TOA. As discussed above (point 1.1), the piControl simulation shows a fast adjustment to stable conditions in the first 13 years of the simulation that can be inferred through the variation of the SAT, SST, the net radiation at TOA and the ocean/atmosphere heat flux. After this fast adjustment the model reaches stable conditions, given by the linear trend of each variable. For this reason, it is assumed that the model was prepared to initiate a coupled Historical simulation.

This information has been included in the revised manuscript (Page 8, Lines L17-L24).

**1.3 It should be documented how the 3d ocean temperature and salinity fields evolve. How does the integrated ocean-atmosphere heat flux evolve? Is there any energy imbalance that could (partly) explain the weak warming in the 20th historical simulation?**

Reply:

This is an important issue. Thank you for comment it. This topic has been included in the revised manuscript (Page 12, Lines L17-L23), as well as the Fig. 3 (Page 55). Similarly to the piControl simulation, the net radiation at TOA has a negative bias and net of the ocean/atmosphere heat flux has a positive bias (Fig. 3). The net radiation at TOA has a mean value of -4.20 W m$^{-2}$ and the net ocean/atmosphere heat flux has a mean value of 1.16 W m$^{-2}$ in the first 50 years. Throughout the simulation, the net radiation at TOA becomes less negative due to the increasing $CO_2$ on the atmosphere and consequential increasing atmospheric heat content. Part of this heat is transferred into the ocean as positive net of the ocean/atmosphere heat flux increasing indicates. Negative values of net radiation at TOA means that the atmosphere is losing heat to the outer space during the simulation, which is likely the reason for the weak warming observed in the Historical simulation (Fig. 2; manuscript).

[Figure]

Figure 3 – Annual average time series for the global average net of the radiation at TOA (a; positive values indicates that the atmosphere is warming) and net of the ocean/atmosphere heat flux (b; positive values indicates that the ocean is warming), simulated by the Historical run over the period 1850-2005 (156 years).

The depth-time Hovmöller diagrams of global mean ocean temperature and salinity departures from their respective initial conditions simulated by the Historical run is

shown in Fig. 4. Here initial conditional means the value of the first year of simulation, in this case, the year 1850. The prominent warming occurs from the surface up to 400 m depth (Fig. 4a). This warming is more significant at the end of the simulation (~0.6 ℃ comparing with initial conditions) and is likely to be related to the global warming of the planet and consequential increasing heat flux from the atmosphere into the ocean. In deeper waters, from 1500 m up to the ocean floor, there is a weaker warming, indicating that the ocean is gaining heat mainly in the upper layers. Between 500-1500 m depth, it is observed a cooling tendency respective to initial conditions. The ocean salinity slightly increases below 1000 m depth and from 1935 the increase reaches 0.04 PSU between 1500 and 3000 m depth compared with the initial values (Fig. 4b). Above 1000 m depth there is a significant freshening of the ocean waters, with the surface waters salinity decreasing up to 0.18 PSU at the end of the simulation. Such tendency can mean that the ocean is still drifting from its initial conditions in the Historical simulation.

This topic has been included in the revised manuscript (Pages 19-20, Lines L17-L28), as well as the Fig. 4 (Page 70).

[Figure]

Figure 4 - Depth-time Hovmöller diagrams of global average ocean temperature and salinity anomalies from the respective initial conditions (IC). Here the initial conditions are taken from the 1[th] year (1850). The diagrams are based on annual average time series simulated by the Historical simulation over the period 1850-2005 (156 years). The thick black line represents the zero contours. Note that the vertical scales are different above and below 1000 m.

**2. A documentation of a newly developed couple model should include an estimate of climate sensitivity. The analyses for that should be standard procedure.**

Reply:

As mentioned in point 1 above, the group has prepared a manuscript that analyzes the piControl and abrupt 4×CO2 simulations. In this manuscript, it is evaluated the climate sensitivity of the model by analyzing the response of the atmosphere in the piControl and abrupt 4×CO2 simulations (Capistrano et al., 2018). For this reason, the climate sensitivity of the model is not addressed in the revised manuscript. Nonetheless, since it is relevant information, a summary has been included in the revised manuscript (Pages 8-9, Lines L24-L4).

Capistrano et al. (2018) estimates that BESM-OA2.5 has an equilibrium climate sensitivity of 2.96 ºC for the abrupt 4×CO2 experiment. This value is within the range from 2.07 to 4.74 ºC that has been computed for 25 CMIP5 models and close to the ensemble averaged value (3.30 ºC).

**3.1 The discussion of the evaluation with observations/reanalyses is descriptive, but does most often not discuss if the biases are acceptable, larger/smaller than in other models, or which consequences come with them.**

Reply:

We agree and discussions comparing BESM biases with other CMIP5 models were included in the manuscript: Page 14 (Lines L2-L13), Page 16 (Lines L18-L22), Page 17 (Lines L8-L10) and Page 17 (Lines L21-L23).

**3.2 For example, if we assume that including aerosols into the BESM influences the historical simulation in a similar way as in other models (Figure 2), BESM would severely underestimate global warming in the last century.**

Reply:

We agree, and a disclaimer has been included in the revised manuscript (Page 12-13, Lines L23-L3).

**3.3 This should be a matter of concern and lead the authors to look for the origin of this discrepancy. Or is the plan that everything will be better in the next generation of BESM, as somehow implied in the conclusions?**

We agree with your observation. However, models can respond in different ways to external forcing, therefore, in the near future, the aim is to carry out a numerical experiment in which the model is forced with observed estimate of aerosol concentration (as read-in field) in order to address to what extension BESM is impacted.

**4. The quality of the figures should be improved. At least in my pdf version I could hardly decipher axis and contour labels**.

Reply:

Thank you for pointing out this issue. All figures have been improved.

**Minor Revisions:**

**Abstract, ln 6: "validation" would mean that you have some measures for when a model is valid, better use "evaluation"**

Reply:

"validation" has been replaced by "evaluation".

**Page 3, ln 6 and following text. The authors say that for an ESM there needs to be an interactive biogeochemical module. But it is not explained later, if or what kind of biogeochemistry model is included and if there are other publications planned on this aspect.**

Reply:

The model used in this work is an ocean-atmosphere-biosphere coupled model, indicated in its name BESM-OA version 2.5. Therefore, the current version of the model is not a full Earth System Model, as disclaimed in the manuscript (Page 3, Lines L20-L23). We acknowledge that an Earth System Model is a comprehensive model that includes all Earth system components, including biogeochemical cycles (e.g. dynamic carbon cycle processes) and cloud-aerosol-chemistry processes. Although the main aim of our group is to build up such a model, at the moment the model is a coupled ocean-atmosphere climate model, without the representation of either biogeochemical cycles or cloud-aerosol-chemistry processes. Nonetheless, the name Brazilian Earth System Model was chosen in order to avoid a future change of the model's name on its transition from an ocean-atmosphere coupled model to an Earth System model. For this reason, in the acronym BESM was always added the

letters OA, which stand for ocean-atmosphere coupled model. Therefore, at the moment, there is any evaluation of the biogeochemical cycles and their interaction with other Earth system components.

For the next version of the model, the group has been working on activate the biogeochemical model (TOPAZ) within the MOM5 in order to simulate biogeochemical cycles in future simulations. Currently, it is running the stand-alone spin-up of the ocean and biogeochemical models. This information has been included in the revised manuscript (Page 4, Lines L2-L4).

**Page 4, ln 1: do you mean interactive aerosols and chemistry or just the ability to use them as read-in fields?**

Reply:

At the moment, the aim is to be included as read-in fields only. For clarity, this information has been included in the revised manuscript (Page 4, Lines L1).

**Ln 11: "seamless predictions" is it really used for weather predictions (at which resolutions?) and do you have to apply problem-specific parameterizations?**

Reply:

Thank you for pointing out this issue. We apologize for the information not being clear. BESM-OA2.5 has not being used for either on short-range and medium-range weather forecasting (<10 days) or climate prediction on a seamless prediction framework. Seamless prediction framework is an aim that the group would like to pursuit in the future. BESM-OA2.3 is used on extended-range weather forecasting (>10 days) and for seasonal prediction.

The sentence about seamless predictions has been deleted: Page 4 (Lines L15-L18).

**Page 6, ln 18ff. How does the system behave after switching from CORE to fully coupled. I doubt that the spin-up is long enough.**

Reply:

This topic is addressed in more details above, in point 1.1 and 1.2 of major revisions.

**Page 15: SSTs seem to be generally too warm. Is that also true for surface air temperature in the control run or in the beginning of the historical? Wouldn't that call for some tuning exercise, e.g. looking into cloud parameterizations?**

Reply:

The surface air temperature (SAT) biases for piControl (Fig. 5) and Historical (Fig. 6) simulations are generally negative over the ocean, with exception of the warm SST biases along the western coast of Africa and the Americas; a common problem of CMIP5 fully coupled model runs (Wang et al., 2014).

Indeed, adjustments of atmospheric model heat fluxes, as those affected by cloud parameterizations is a current object of research in BESM development team.

[Figure]

Figure 5 - Spatial map of annual mean surfasse air temperature (SAT) bias of BESM-OA2.5 (piControl) relative to 20CRv2. The averages values are computed over the periods 971–1000 (for BESM-OA2.5) and 1871–1900 (for 20Crv2). Units are in C.

SAT Bias mean: -1.8 °C    rmse: 4.0 °C

Figure 6 - Spatial map of annual mean surfasse air temperature (SAT) bias of BESM-OA2.5 (Historical) relative to 20CRv2. The averages values are computed over the periods 1871–1900 (for BESM-OA2.5) and 1871–1900 (for 20Crv2). Units are in C.

**Page 17, ln 21ff. How does AMOC look in the control run, any drift?**

Reply:

This topic is addressed in more details above, in point 1.1 of major revisions. There is a drift but is comparable with other state-of-the-art models.

**Bibliography**

Capistrano, V. B., Nobre, P., Tedeschi, R., Silva, J., Bottino, M., da Silva Jr., M. B., Menezes Neto, O. L., Figueroa, S. N., Bonatti, J. P., Kubota, P. Y., Reyes Fernandez, J. P., Giarolla, E., Vial, J., and Nobre, C. A.: Overview of climate change in the BESM-OA2.5 climate model, Geosci. Model Dev. Discuss., https://doi.org/10.5194/gmd-2018-209, in review, 2018.

Jones, G. S., Stott, P. A. and Christidis, N.: Attribution of observed historical near-surface temperature variations to anthropogenic and natural causes using CMIP5 simulations, J. Geophys. Res. Atmos., 118(10), 4001–4024, doi:10.1002/jgrd.50239, 2013.

Wang, C., Zhang, L. and Lee, S.: A global perspective on CMIP5 climate model biases, Nat. Clim. Chang., 4(3), 201–205, doi:10.1038/NCLIMATE2118, 2014.

---

## Author Response (AR1)

Dr. Qiang Wang
Topical Editor
Geoscientific Model Development
October 28th 2018

Manuscript reference No. GMD-2018-91

Dear Dr. Qiang Wang,

Please find attached a revised version of the manuscript, **The Brazilian Earth System Model version 2.5: Evaluation of its CMIP5 historical simulation**, which we would like to submit for publication in Geoscientific Model Development

We appreciate the opportunity to improve the manuscript.

In the following pages are our point-by-point revisions.

- Page 2, Line 6

    "*validated*" has been replaced by "*evaluated*"

- Page 4, Line 1

    It has been included the following text: "*(as read-in fields)*"

- Page 4, Line 2

    It has been included the following text: "*Currently, work has been done on activate the biogeochemical model (TOPAZ) within the MOM5 in order to simulate biogeochemical cycles in future simulations.*"

- Page 4, Line 5

    "*BESM was*" has been replaced by "*The previous version of BESM (BESM-OA2.3) was*"

- Page 4, Line 15

    It has been deleted the following text: "*One of primary conceptual aim in developing and improving BESM's parameterizations is to serve as a model to be used from numerical weather prediction to seasonal forecast up to the projection of climate change scenarios in a seamless framework as proposed by Palmer et al. (2008).*"

- Page 4, Line 19

    "*BESM*" has been replaced by "*BESM-OA2.3*"

- Page 4, Line 19

    It has been included the following text: "*(10-30 days)*"

- Page 4, Line 20

    It has been included the following text: "*(three months)*"

- Page 5, Line 21

    "*to 2 W m$^{-2}$*" has been replaced by "*to -4 W m$^{-2}$*"

- Page 8, Line 2

    "*for 700 years*" has been replaced by "*for 1140 years*"

- Page 8, Line 4

    "*460*" has been replaced by "*for 1000 years*"

- Page 8, Line 16

    It has been included the following text: "*The ocean stand-alone runs for 71 years (13 years period of ocean model spin-up forced by climatological atmospheric fields plus 58 years period forced by interannually varying atmospheric fields). Then a spin-up of the fully coupled model is done for 100 years. The ocean and atmosphere states at the end of this 100 years long integration are used as the initial condition for the piControl simulation. The piControl simulation shows stable conditions after a fast adjustment over the first 13 years of simulation (figure not shown). The analysis of the piControl and 4×CO2 simulations are described in Capistrano el al. (2018) and Nobre et al. (2018, in preparation). Capistrano et al. (2018) estimates that BESM-OA2.5 has an equilibrium climate sensitivity of 2.96 ºC for the abrupt 4×CO2 experiment. This value is within the range from 2.07 to 4.74 ºC that has been*

*computed for 25 CMIP5 models and close to the ensemble averaged value (3.30 ºC).*"

- Page 9, Line 13

It has been included the following text: "*and from the CPC Merged Analysis of Precipitation (CMAP; Xie and Arkin, 1997) with global horizontal resolution of 2.5° × 2.5° (https://www.esrl.noaa.gov/psd/data/gridded/data.cmap.html);*"

- Page 12, Line 4

It has been deleted the following text: "*The reason for absence of this warming tendency in the last decades of the 20th century is not clear.*"

- Page 16, Line 21

It has been included the following text: "*The net radiation at the top of atmosphere (TOA) has a negative bias and net of the ocean/atmosphere heat flux has a positive bias (Fig. 3). The net radiation at TOA has a mean value of -4.20 W m$^{-2}$ and the net ocean/atmosphere heat flux has a mean value of 1.16 W m$^{-2}$ in the first 50 years. Throughout the simulation the net radiation at TOA becomes less negative due to the increasing $CO_2$ on the atmosphere and consequential increasing atmospheric heat content. Part of this heat is transferred into the ocean as positive net of the ocean/atmosphere heat flux increasing indicates. The negative net radiation at TOA and the positive ocean/atmosphere heat flux are likely the reason for the weak warming observed in the Historical simulation (Fig. 2), since the atmosphere is losing heat to the outer space and into ocean during the simulation.*"

- Page 13, Line 1

   "*3*" has been replaced by "*4*"

- Page 13, Line 1

   It has been included the following text: "for (a) BESM-OA2.5, (b) GPCP dataset, and the spatial distribution of annual mean precipitation"

- Page 13, Line 3

   "*biases*" has been replaced by "*bias*"

- Page 13, Line 3

   It has been included the following text: "*(c)*"

- Page 13, Line 3

   It has been included the following text: "*relative to the GPCP dataset and (d) for BESM-OA2.5 relative to the CMAP dataset. The spatial annual mean precipitation are*"

- Page 13, Line 4

   It has been deleted the following text: "*The bias is obtained*"

- Page 13, Line 5

   It has been deleted the following text: "*through the difference with GPCP dataset, in which the*"

- Page 13, Line 5

It has been deleted the following text: "*are computed*"

- Page 13, Line 7

It has been included the following text: "*and CMAP datasets*"

- Page 13, Line 7

[revised manuscript text omitted]

- Page 20, Line 3

It has been deleted the following text: "*4.1.5 Atlantic Meridional Ocean Circulation*"

- Page 20, Line 10

"*12*" has been replaced by "*14*"

- Page 20, Line 12

"*12*" has been replaced by "*14*"

- Page 20, Line 17

   "*12*" has been replaced by "*14*"

- Page 20, Line 18

   "*The same figure also plots*" has been replaced by "*For comparison, Figure 14c plots*"

- Page 21, Line 19

   "*13*" has been replaced by "*15*"

- Page 22, Line 4

   "*13*" has been replaced by "*15*"

- Page 22, Line 5

   "*13*" has been replaced by "*15*"

- Page 22, Line 6

   "*13*" has been replaced by "*15*"

- Page 22, Line 8

   "*14*" has been replaced by "*16*"

- Page 22, Line 18

   "*14*" has been replaced by "*16*"

- Page 23, Line 13

   "*15*" has been replaced by "*17*"

- Page 23, Line 18

    "*15*" has been replaced by "*17*"

- Page 23, Line 22

    "*15*" has been replaced by "*17*"

- Page 24, Line 22

    "*16*" has been replaced by "*18*"

- Page 25, Line 7

    "*16*" has been replaced by "*18*"

- Page 25, Line 8

    "*16*" has been replaced by "*18*"

- Page 25, Line 9

    It has been included the following text: "*4.2.1.4 Madden-Julian Oscillation*"

- Page 25, Line 10

[revised manuscript text omitted]

- Page 50, Line 20

    The following reference has been included: "*Zhang, C.: Madden-Julian Oscillation, Rev. Geophys., 43(2), 1–36, doi:10.1029/2004RG000158, 2005.*"

- Page 53

    Figure 1 has been improved.

- Page 54

    The quality of the Figure 2 has been improved.

- Page 55

    A new figure has been included (Figure 3). It has been suggested by reviewer #2.

- Pages 56

Previous Figure 3 has been improved and completed with more figures, as suggested by reviewer #1. It has been renamed as Figure 4.

- Page 57, Line 1

"*Spatial map of annual mean precipitation bias of BESM-OA2.5 relative to GPCP. The averages values are computed over the periods 1971–2000 and 1979–2008, for BESM-OA2.5 and GPCP, respectively. Units are in mm day$^{-1}$.*" has been replaced by "*Spatial map of annual mean precipitation for (a) BESM-OA2.5, for (b) GPCP, (c) the bias of BESM-OA2.5 relative to GPCP and (d) the bias of BESM-OA2.5 relative to CMAP. The averages values are computed over the periods 1971–2000 (for BESM-OA2.5) and 1979–2008 (for GPCP and CMAP). Units are in mm day$^{-1}$.*"

- Page 58

The quality of previous Figure 4 has been improved and it has been renamed as Figure 5.

- Page 60

The quality of previous Figure 5 has been improved and it has been renamed as Figure 6.

- Page 61

The quality of previous Figure 6 has been improved and it has been renamed as Figure 7.

- Page 62

The quality of previous Figure 7 has been improved and it has been renamed as Figure 8.

- Page 63

The quality of previous Figure 8 has been improved and it has been renamed as Figure 9.

- Page 64

Figure 9 has been improved and completed with more figures, as suggested by reviewer #1. It has been renamed as Figure 10.

Page 65, Line 1

"*surface bias*" has been replaced "*surface temperature for (a) BESM-OA2.5, (b) ERSSTv4 and (c) the bias*"

- Page 66

The quality of previous Figure 10 has been improved and it has been renamed as Figure 11.

- Page 68

The quality of previous Figure 11 has been improved and it has been renamed as Figure 14.

- Page 70

A new figure has been included (Figure 13). It has been suggested by reviewer #2.

- Page 71

The quality of previous Figure 12 has been improved and it has been renamed as Figure 14.

- Page 73

The quality of previous Figure 13 has been improved and it has been renamed as Figure 15.

- Page 75

  The quality of previous Figure 14 has been improved and it has been renamed as Figure 16.

- Page 76

  The quality of previous Figure 15 has been improved and it has been renamed as Figure 17.

- Page 78

  The quality of previous Figure 16 has been improved and it has been renamed as Figure 18.

- Page 80

  A new figure has been included (Figure 19). It has been suggested by reviewer #1.

- Page 82

  The quality of previous Figure 17 has been improved and it has been renamed as Figure 20.

- Page 84

  The quality of previous Figure 18 has been improved and it has been renamed as Figure 21.

- Page 85

  The quality of previous Figure 19 has been improved and it has been renamed as Figure 22.

The quality of previous Figure 20 has been improved and it has been renamed as Figure 23.

The quality of previous Figure 21 has been improved and it has been renamed as Figure 24.

The quality of previous Figure 22 has been improved and it has been renamed as Figure 25.

[revised manuscript text omitted]

---

## Author Response (AR2)

Dr. Qiang Wang
Topical Editor
Geoscientific Model Development
January 20th 2019

Manuscript reference No. GMD-2018-91

Dear Dr. Qiang Wang,

Please find attached a revised version of the manuscript, **The Brazilian Earth System Model version 2.5: Evaluation of its CMIP5 historical simulation**, which we would like to submit for publication in Geoscientific Model Development

We appreciate the opportunity to improve the manuscript.

In the following pages are our point-by-point revisions.

- Page 4, Line 10

"Ocean" has been replaced by "Oceans"

- Page 8, Line 11

[revised manuscript text omitted]

- Page 23, Line 4

"ocean" has been replaced by "Ocean"

- Page 23, Line 12

"15" has been replaced by "16"

- Page 23, Line 13

"15" has been replaced by "16"

- Page 23, Line 14

   "15" has been replaced by "16"

- Page 23, Line 16

   "16" has been replaced by "17"

- Page 23, Line 22

   "ocean" has been replaced by "Ocean"

- Page 24, Line 1

   "ocean" has been replaced by "Ocean"

- Page 24, Line 3

   "16" has been replaced by "17"

- Page 24, Line 6

   "ocean" has been replaced by "Ocean"

- Page 24, Line 21

   "17" has been replaced by "18"

- Page 25, Line 3

   "17" has been replaced by "18"

- Page 25, Line 7

   "17" has been replaced by "18"

- Page 26, Line 8

    "18" has been replaced by "19"

- Page 26, Line 13

    "16" has been replaced by "18"

- Page 26, Line 16

    "18" has been replaced by "19"

- Page 26, Line 17

    "18" has been replaced by "19"

- Page 27, Line 9

    "19" has been replaced by "20"

- Page 27, Line 21

    "19" has been replaced by "20"

- Page 27, Line 23

    "19" has been replaced by "20"

- Page 28, Line 19

    "20" has been replaced by "21"

- Page 28, Line 20

    "20" has been replaced by "21"

- Page 28, Line 23

  "20" has been replaced by "21"

- Page 28, Line 23

  It has been included the following text: "21"

- Page 30, Line 3

  "21" has been replaced by "22"

- Page 30, Line 4

  "21" has been replaced by "22"

- Page 30, Line 6

  "21" has been replaced by "22"

- Page 30, Line 15

  "22" has been replaced by "23"

- Page 30, Line 22

  "22" has been replaced by "23"

- Page 31, Line 17

  "23" has been replaced by "24"

- Page 32, Line 22

  "24" has been replaced by "25"

- Page 33, Line 6

"25" has been replaced by "26"

- Page 33, Line 9

It has been included the following text: "26"

- Page 37, Line 11

It has been included the following text: "MBJ is supported by a grant funded by FAPESP (2018/06204-0)."

- Page 46, Line 24

It has been included the following text: "Menary, M. B., Kuhlbrodt, T., Ridley, J., Andrews, M. B., Dimdore-Miles, O. B., Deshayes, J. et al.: Preindustrial control simulations with HadGEM3-GC3.1 for CMIP6., J. Adv. Model. Earth Syst., 10, 3049–3075, doi:https://doi.org/10.1029/2018MS001495, 2018." It has been suggested by reviewer #2.

- Page 53

Figure 1 has been improved. The revised figure shows the 100 years of coupled spin-up run.

- Page 69

Figure 13 has been improved. It has been suggested by reviewer #2.

- Page 70, Line 1

"Depth-time Hovmöller diagrams of global average ocean temperature and salinity anomalies from the respective initial conditions (IC). Here the initial conditions are taken from the $1^{th}$ year. The diagrams are based on annual average time series simulated by the Historical simulation over the period 1850-2005 (156 years). The thick black line represents the zero contours. Note that the vertical scales are different above and below 1000 m." has been replaced by "Depth-time Hovmöller diagrams of global average ocean (a) salinity and (b) temperature anomalies from the respective initial conditions (IC). Here the initial conditions are taken from the 1th year for (a, b) Historical simulation, and $14^{th}$ year for (c) piControl simulation. (d) presents the difference between the temperature anomalies of Historical relative to piControl. The diagrams are based on annual average time series simulated by the Historical simulation over the period 1850-2005 (156 years) and by piControl simulation over the period 14-169 years (156 years). The thick black line represents the zero contours. Note that the vertical scales are different above and below 1000 m."

- Page 73

A new figure has been included (Figure 15). It has been suggested by reviewer #1.

- Page 73, Line 3

It has been included the following text: "Figure 15 - BESM-OA2.5 mean sea ice concentration for March (a, c) and September (b, d) for each hemisphere. The solid black lines show the 15 % mean sea ice concentration for 20CRv2 Reanalysis. The averages values are computed over the period 1971–2000 for BESM-OA2.5 and 20CRv2. The concentration is presented in percentage."

- Page 74, Line 3

        "15" has been replaced by "16"

- Page 76, Line 3

        "16" has been replaced by "17"

- Page 78, Line 1

        "17" has been replaced by "18"

- Page 79, Line 3

        "18" has been replaced by "19"

- Page 81, Line 3

        "19" has been replaced by "20"

- Page 83, Line 3

        "20" has been replaced by "21"

- Page 85, Line 3

        "21" has been replaced by "22"

- Page 86, Line 3

        "22" has been replaced by "23"

- Page 87, Line 3

     "23" has been replaced by "24"

- Page 88, Line 3

     "24" has been replaced by "25"

- Page 89, Line 2

     "25" has been replaced by "26"

**Anonymous Referee #1**

We thank the valuable comments, whose responses follow:

**Major Revisions:**

**1.1 The CMIP6 historical experiment forcing is released a longtime ago, and the BESM model is targeted at the CMIP6 project, why do not run the BESM with CMIP6 forcing data?**

Reply:

The present study uses data from a simulation forced following the CMIP5 protocol with the objective of evaluating the model version implemented for the CMIP5 project. Presently, our group is working on an updated version of the BESM model to be used for CMIP6.

**1.2 Moreover, the GHG forcing is only one aspect of the historical forcing, what is the consideration to ignore other forcing? It is unfair to compare the GHG-historical simulation to the real-world observation, how to clarify the role of other forcing (e.g. aerosols)?**

Reply:

We agree with your observations. To compare with real-world it is desirable that the historical simulation is forced with observed aerosols concentration and land use changes jointly with GHG forcing. However, in the process of developing a full ESM, comparing the current version of the model (without the effects of aerosols and land use change, for example) with observations is the only possibility at hand. Many centers have evaluated their models piControl simulation against observed and/or Reanalysis (e.g. Swapna et al., 2015 and Menary et al., 2018). We consider that evaluating ours Historical simulation against reanalysis, in this context, is less of a problem than contrasting the piControl run.

**2. The model suffers a large TOA energy imbalance (about -4 W m$^{-2}$) and surface imbalance (about 1.2 W m$^{-2}$) from the beginning of historical simulation (e.g. 1850-1900). The authors should discuss the possible reason and causing of the energy bias. Is the imbalance due to the non-conservation in the AGCM or coupling process?**

Reply:

The AGCM stand-alone run shows a net radiation at TOA of 0.25 W m$^{-2}$ during 20 years of simulation (Fig. 1a). Such radiative imbalance is within the range simulated by different atmospheric models. However, in the coupled simulation, the net radiation imbalance at TOA is amplified up to -4 W m$^{-2}$ (Fig. 1b). The reason for such imbalance is related to higher loss of energy at TOA both from the outgoing long-wave radiation (OLR) and outgoing short-wave radiation (OSR), compared with AGCM stand-alone simulation (Fig. 1c and 1d). In Fig. 1c and 1d, the solid lines represent the coupled model and the dashed lines represent the AGCM. The higher loss of energy through outgoing long-wave radiation is due to the warm SST bias that BESM-OA2.5 suffers (Fig. 10c, manuscript). The higher loss of energy through the outgoing short-wave radiation is potentially due to enhanced cloud formation in the coupled model run.

[Figure]

Figure 1 – Net of the radiation of TOA simulated by (a) stand-alone AGCM for 20 years and (b) BESM-OA2.5 Historical for the first 20 years (1850-1870). (c) and (d) are outgoing long-wave radiation and outgoing short-wave radiation, respectively. In (c) and (d) the solid lines represent the coupled model and the dashed lines represent the AGCM. Units are in W m⁻².

This information has been included in the revised manuscript (Page 12, Lines L16-L18).

**3.1 Figure 2 shows the 2-m air temperature is less response to the GHG forcing in BESM during 1850s-1960s, while it isn't the case for any CMIP5 model. Any explanation for this unique feature?**

Reply:

The net radiation at TOA has a mean value of -4.20 W m$^{-2}$ and the net ocean/atmosphere heat flux has a mean value of 1.16 W m$^{-2}$ in the first 50 years (Fig. 3, manuscript). Throughout the simulation, the net radiation at TOA becomes less negative due to the increasing $CO_2$ on the atmosphere and consequential increasing atmospheric heat content. Part of this heat is transferred into the ocean as indicated by the increasing positive net heat flux into the ocean. Negative values of net radiation flux at TOA means that the atmosphere is losing heat to the outer space during the simulation, which is likely the reason for the weak air temperature response to the GHG forcing observed in the Historical simulation (Fig. 2; manuscript).

**3.2 The simulated SST is much warmer than the ERSST v4 during the evaluation period (Fig. 10), how is the land surface temperature?**

Reply:

The land surface temperature bias for BESM-OA2.5 Historical simulation is generally negative over desert and semi-arid regions (Fig. 2). Such negative bias is noticed in the Arabian Desert, but also in the Sahara, Kalahari, Gobi, Polar Arctic, Patagonia, Sonoran and Australian deserts. The negative bias is also present in the Brazilian semi-arid region. Conversely, over the most vegetated regions the model presents positive bias, as the Amazon, tropical Africa, North America and Europe. Such biases are likely caused by drier air simulated by the model over desert regions, which tends to enhance the latent heat flux from the land surface over desert areas and causing higher cooling effect compared with Reanalysis, particularly during the night. Conversely, in forest regions the excess of air moisture constrains the loss of latent heat flux from the land surface. This enhances the land surface temperature compared with the Reanalysis.

[Figure]

Figure 2 - Spatial map of annual mean land surface skin temperature bias of BESM-OA2.5 relative to ERA-Interim. The averages values are computed over the periods 1971–2000 (BESM-OA2.5) and 1979–2008 (ERA-Interim). Units are in ℃.

**3.3 How is the sea ice simulation under such a warm climate, please provide the figure of sea ice performance?**

Reply:

Figure 3 shows the mean sea ice concentration simulated by BESM-OA2.5 for the end of the winter and the summer seasons for each hemisphere, over the period 1971-2000. The thick black lines represent the 15 % climatological values for the period 1971-2000 given by the 20CRv2 Reanalysis. The sea ice concentration in the Arctic winter is overestimated in the Atlantic, specifically north of the Scandinavia (Fig. 3a). However, in summer the Arctic sea ice is underestimated (Fig. 3b). In the Antarctica summer the model shows a significant underestimation of the sea ice concentration (Fig. 3c). During the Antarctica winter the model generally overestimates the extension of the sea ice concentration over all Southern Ocean (Fig. 3d). Such seasonal sea ice concentration amplitude is likely related to bias radiative net over higher latitudes that the model suffers, which during the winter in each hemisphere tend to generate higher extension of sea ice and during the summer in each hemispheres tend to enhance the sea ice melting compared with the Reanalysis.

[Figure]

Figure 3 - BESM-OA2.5 mean sea ice concentration for March (a, c) and September (b, d) for each hemisphere. The solid black lines show the 15 % mean sea ice concentration for 20CRv2 reanalysis. The averages values are computed over the period 1971–2000 for BESM-OA2.5 and 20CRv2. The concentration is presented in percentage.

This topic has been included in the revised manuscript (Pages 21-22, Lines L24-L12).

The figure has also been included in the revised manuscript (Page 73).

**3.4 It is also necessary to clarify whether the 1.5K SST warming affect the model climate variability or not.**

Reply:

Some important climate variabilities are reasonable well simulated by the model, as NAO, AMM, AMOC, PSA or PNA. Therefore, it is not clear whether the general warm bias that the model suffers has a profound impact on the climate variability simulated by the model. However, we have analyzed the global spatial standard deviation of variables that are influenced by SST, as precipitation, outgoing long-wave radiation (OLR), sea level pressure (SLP) and surface air temperature (SAT).

The global standard deviation of monthly precipitation anomalies simulated by BESM-OA2.5 over the period 1971-2000 is compared with GPCP standard deviation over the period 1979-2010 (Fig. 4). The largest precipitation variability is found in the west equatorial Pacific (Fig 4a). BESM-OA2.5, besides simulating a comparable standard deviation in the west equatorial Pacific, presents spurious large precipitation variability over the Indian Ocean (Fig 4b). The same pattern is observed in the OLR, which indicates an enhanced convection over the Indian Ocean (Fig 5). It is not clear the reasons for such phenomenon, but the Indian Ocean warm bias can enhance the convection in this region. The tropical Atlantic is other region that shows significant differences between the model and Reanalysis. BESM-OA2.5 has a strong variability over the tropical South Atlantic. Global SLP anomalies standard deviation shows no significant difference between the model and Reanalysis (Fig 6). The pattern is reasonably captured, particularly the higher variability over the Aleutian Islands, Iceland and Amundsen Sea (60-70 °S; 90 °W). In the case of SAT, the model generally presents lower SAT anomalies variability, although the pattern is captured by the model (Fig. 7). Thus, the standard deviation of precipitation, ORL, SLP and SAT anomalies do not show significant difference from the Reanalysis, besides the Indian Ocean region, as has been discussed above.

[Figure]

Figure 4 – Standard deviation of monthly precipitation anomalies for (a) GPCP and (b) BESM-OA2.5. The standard deviation values are computed over the periods 1971–2000 (BESM-OA2.5) and 1979-2010 (GPCP). Units are in mm/day.

[Figure]

Figure 5 – Standard deviation of monthly OLR anomalies for (a) 20CRv2 and (b) BESM-OA2.5. The standard deviation values are computed over the period 1971–2000. Units are in W m⁻².

[Figure]

Figure 6 – Standard deviation of monthly SLP anomalies for (a) 20CRv2 and (b) BESM-OA2.5. The standard deviation values are computed over the period 1971–2000. Units are in hPa.

[Figure]

Figure 7 – Standard deviation of monthly SAT anomalies for (a) ERA-Interim and (b) BESM-OA2.5. The standard deviation values are computed over the period 1971–2000. Units are in °C.

**4. As point out by reviewer 2, the coupled model spin-up period is very short. It is unclear, from the manuscript, whether the decadal variability is affected by model spin-up or not. Is the weakening of AMOC strength (Fig. 14) due to the model adjustment?**

Reply:

The AMOC negative linear trend observed in Historical simulation is likely linked to the model`s drifting throughout simulation. Such conclusion is also reinforced by the depth-time Hovmöller diagrams of global mean ocean temperature and salinity departures from their respective initial conditions simulated by the Historical run is shown in Fig. 8. Here initial conditional means the value of the first year of simulation, in this case, the year 1850. The prominent warming occurs from the surface up to 400 m depth (Fig. 8a). This warming is more significant at the end of the simulation (~0.6 °C comparing with initial conditions) and is likely to be related to the global warming of the planet and consequential increasing heat flux from the atmosphere into the ocean. In deeper waters, from 1500 m up to the ocean floor, there is a weaker warming, indicating that the ocean is gaining heat mainly in the upper layers. Between 500-1500 m depth, it is observed a cooling tendency respective to initial conditions. The ocean salinity slightly increases below 1000 m depth and from the year 1935 the increase reaches 0.04 PSU between 1500 and 3000 m depth compared with the initial values (Fig. 8b). Above 1000 m depth there is a significant freshening of the ocean waters, with the surface waters salinity decreasing up to 0.18 PSU at the end of the simulation. Such tendency can mean that the ocean is still drifting from its initial conditions in the Historical simulation. Similar drift of the model is also observed in the piControl simulation for global average ocean temperature and salinity anomalies from the respective initial conditions.

[Figure]

Figure 8 - Depth-time Hovmöller diagrams of global average ocean temperature and salinity anomalies from the respective initial conditions (IC). Here the initial conditions are taken from the 1[th] year (1850). The diagrams are based on annual average time series simulated by the Historical simulation over the period 1850-2005 (156 years). The thick black line represents the zero contours. Note that the vertical scales are different above and below 1000 m.

**5. Some experience on coupled model tuning would be desirable.**

Reply:

Since all simulations of the BESM-OA2.5 have already been performed, this suggestion will be taken into account on future simulations performed with the new version.

**The issue of model drift is still somewhat "put under the carpet". The authors state that the surface quantities assume a stable state after 13 years in the PiControl run and suggest that drift is therefore no issue. However, the more than 1 W/m$^2$ excess heat that the ocean receives over the entire simulation leads to strong heat uptake. The authors have shown that to the reviewer in their response (Fig 2). However, comparing this figure with the Hovmueller plot in the main text (Figure 13 revised version) suggest that a considerable part of that warming is simply drift and not, as the authors claim in the text, a response to global warming.**

**Since the historical and PiControl runs have been run in parallel, a solution would be to include the control run figure. Another possibility would be to discuss the model drift in terms of an integrated quantity, e. g. heat content or steric (thermohaline) sea-level change, where the "real" changes in the historical run is estimated after the control run drift is subtracted.**

Reply:

We agree with your observation. The ocean temperature anomalies above 600 m reaches ~0.6 $^\circ$C in the Historical simulation whereas in the piControl it reaches ~0.4 $^\circ$C. This difference of 0.2 $^\circ$C between the two simulations is likely due to the global warming. Therefore, there is a contribution of the ocean drift and a smaller contribution of the global warming of the ocean temperature increase in the Historical simulation. To properly provide this information we have included a figure showing the differential heating (Historical minus piControl). The improved text has been included in the revised manuscript (Pages 19, Lines L12). The figure has also been improved in the revised manuscript (Page 69).

[Figure]

Figure 1 – Depth-time Hovmöller diagrams of global average ocean temperature anomalies for (a) Historical and (b) piControl simulations from the respective initial conditions (IC), and (c) the difference between Historical and piControl simulations. In the case of the Historical simulation, the initial conditions are taken from the 1[th] year. In the case of the piControl the initial conditions are taken from the 14[th] year, therefore after the 13 initial years of the adjustment. The diagrams are based on annual average time series. The thick black line represents the zero contours. Note that the vertical scales are different above and below 1000 m.

**2. I am also still confused by the description of the spin-up procedure. In the authors' response it is explained that there has been a 100-year spin up (with a slightly different version of the coupled model) "as initial conditions for the piControl". I don't understand, why this is not mentioned in the main text and in the schematic and there is just talk of a 13 year spin-up. Was that done in addition to the 100 years?**

Reply:

We agree with the reviewer's concern. A new figure has been included, depicting the complete spin-up process. Yes, the 13 years spin-up is additional to the 100 years spin-up, with slight differences between the atmospheric model's versions. To explain the spin-up process accurately, the experiments design figure has been improved (Figure 1 revised manuscript; Page 53) and the following text has been included on the experiments design topic in the revised manuscript (Page 8, Lines L11-L20):

"The ocean stand-alone runs for 71 years (13 years period of ocean model spin-up forced by climatological atmospheric fields plus 58 years period forced by interannually varying atmospheric fields). Then a spin-up of the fully coupled model is done for 100 years. The ocean and atmosphere states at the end of these 100 years long integration are used as the initial condition for the piControl simulation. The atmospheric model's versions are slightly different for 100 years spin-up and the piControl run, in the parameterizations of the land ice albedo and cloud microphysics. The Historical simulation uses as initial conditions information of the 14[th] year provided by the piControl simulation. The piControl simulation shows stable conditions after a fast adjustment over the first 13 years of simulation (figure not shown). Therefore, it is assumed that the Historical simulation has a spin-up of 113 years"

**3 Page 7, lns 3ff: How is river runoff and transfer to the ocean treated?**

Reply:

River runoff is treated as a time invariant inflow on a spatially fixed grid along the continental margins, to mass balance ocean precipitation minus evaporation over the ocean area. This is a standard feature of the ocean model configuration as distributed by GFDL.

**4. Page 20, lns 3ff: "appropriate depth": Not sure if that is true, e.g. the zero crossing in the North Atlantic is below 2500 m. (Compare for example, with the recent paper by Menary et al.: https://agupubs.onlinelibrary.wiley.com/doi/abs/10.1029/2018MS001495).**

Reply:

Thank you for this comment. We agree that a more precise description is necessary. The following improved text has been included in the revised manuscript (Page 20, Line L22):

[revised manuscript text omitted]

---

## Author Response (AR3)

Dr. Qiang Wang
Topical Editor
Geoscientific Model Development
February 26th 2019

Manuscript reference No. GMD-2018-91

Dear Dr. Qiang Wang,

Please find attached a revised version of the manuscript, **The Brazilian Earth System Model version 2.5: Evaluation of its CMIP5 Historical Simulation**, which we would like to resubmit for publication in Geoscientific Model Development.

We appreciate the opportunity to improve the manuscript.

Following your suggestion, a complete English editing of the reviewed manuscript was performed by a native English speaker scientist with publishing experience. The English certificate editing certificate is attached. All the changes in the text based on the reviewers' suggestions and on the English editing are marked in green (included text) and red (deleted text) in the "WithChangeTrack" manuscript.

The manuscript English editing involved grammar, phrasing, and punctuation. Much of the editing involved articles and some word order changes to improve the flow and readability. The most substantial feature of the editing was the use of different tenses. It was edited in such a way that operations performed by the most in the past, which led to figures, are now in the past tense - as they happened once in the past. In contrast, when these operations showed or confirmed a feature or ability of the model that is ongoing, it is maintained in the present tense.

In the following pages are our most important point-by-point revisions.

- Page 9, Line 2

Based on the second comment of the reviewer #2 the following text: "The ocean stand-alone runs for 71 years (13 years period of ocean model spin-up forced by climatological atmospheric fields plus 58 years period forced by interannually varying atmospheric fields). Then a spin-up of the fully coupled model is done for 100 years. The ocean and atmosphere states at the end of this 100 years long integration are used as the initial condition for the piControl simulation. The piControl simulation shows stable conditions after a fast adjustment over the first 13 years of simulation (figure not shown)." has been replaced by "The ocean stand-alone ran for 71 years (a 13-year period of ocean model spin-up forced by climatological atmospheric fields plus a 58-year period forced by interannually varying atmospheric fields). Next, a spin-up of the fully coupled model was performed for 100 years. The oceanic and atmospheric states at the end of this 100-year-long integration were used as the initial conditions for the piControl simulation. The versions of the model differ slightly in the 100-year spin-up and the piControl run, in the parameterizations of the land ice albedo and in the cloud microphysics. For its initial conditions, the historical simulation used information about the 14th year provided by the piControl simulation. The piControl simulation showed stable conditions following a fast adjustment over the first 13 years of simulation (figure not shown). Therefore, it is assumed that the historical simulation had a spin-up of 113 years."

- Page 13, Line 18

Based on the second comment of the reviewer #1, the following text has been included: "
[revised manuscript text omitted]

- Page 43, Line 14

  It has been deleted the following reference: "Arakawa, A. and Schubert, W. H.: Interaction of a Cumulus Cloud Ensemble with the Large-Scale Environment, Part I, J. Atmos. Sci., 31(3), 674–701, doi:10.1175/1520-0469(1974)031<0674:IOACCE>2.0.CO;2, 1974." Through a review of the atmospheric model, it has been concluded that the reference is not appropriated.

- Page 50, Line 24

  It has been included the following reference: "Menary, M. B., Kuhlbrodt, T., Ridley, J., Andrews, M. B., Dimdore-Miles, O. B., Deshayes, J. et al.: Preindustrial control simulations with HadGEM3-GC3.1 for CMIP6., J. Adv. Model. Earth Syst., 10, 3049–3075, doi:https://doi.org/10.1029/2018MS001495, 2018." It has been suggested by reviewer #2.

- Page 54, Line 5

  It has been included the following reference: "Tarasova, T. A. and Fomin, B. A.: Solar Radiation Absorption due to Water Vapor: Advanced Broadband Parameterizations, J. Appl. Meteorol., 39(11), 1947–1951, doi:10.1175/1520-0450(2000)039<1947:SRADTW>2.0.CO;2, 2000." Through a review of the

atmospheric model, it has been concluded that the reference is the correct one.

- Page 54, Line 8

It has been deleted the following reference: "Tarasova, T. A., Barbosa, H. M. J. and Figueroa, S. N.: In- corporation of new solar radiation scheme into CPTECGCM. Instituto Nacional de Pesquisas Espaciais Tech. Rep. INPE-14052-NTE/371, 44 pp. [Available online at http://mtc-m15. sid.inpe.br/col/sid.inpe.br/iris%401915/2006/01.16.10.40/doc/publicacao.pdf, 2006." Through a review of the atmospheric model, it has been concluded that the reference is not appropriated.

- Page 55, Line 14

It has been included the following reference: "Webster, S., Brown, A. R., Cameron, D. R. and P.Jones, C.: Improvements to the representation of orography in the Met Office Unified Model, Q. J. R. Meteorol. Soc., 129(591), 1989–2010, doi:10.1256/qj.02.133, 2003." Through a review of the atmospheric model, it has been concluded that the reference is the correct one.

- Page 58

Figure 1 has been improved. The revised figure shows the 100 years of coupled spin-up run. Based on the second comment of the reviewer #2.

- Page 74

Figure 13 has been improved. It has been suggested by reviewer #2 (comment 1).

- Page 75, Line 1

"Depth-time Hovmöller diagrams of global average ocean temperature and salinity anomalies from the respective initial conditions (IC). Here the initial conditions are taken from the 1[th] year. The diagrams are based on annual average time series simulated by the Historical simulation over the period 1850-2005 (156 years). The thick black line represents the zero contours. Note that the vertical scales are different above and below 1000 m." has been replaced by "Depth-time Hovmöller diagrams of the global average ocean (a) salinity and (b) temperature anomalies from the respective initial conditions (IC). Here, the initial conditions were taken from the first year for (a, b) historical simulation and from the 14[th] year for the (c) piControl simulation. The map shown in (d) presents the difference between the temperature anomalies of the historical simulation relative to the piControl. The diagrams are based on annual average time series simulated by the historical simulation over the period 1850–2005 (156 years) and by the piControl simulation over the period 14–169 years (156 years). The thick black line represents the zero contours. Note that the vertical scales are different above and below 1000 m."

- Page 78

A new figure has been included (Figure 15). It has been suggested by reviewer #1 (comment 3.3).

- Page 78, Line 3

It has been included the following text: "BESM-OA2.5 mean sea ice concentrations for March (a, c) and September (b, d) for each hemisphere. The solid black lines show the 15 % mean sea ice concentration from the 20CRv2 Reanalysis. The average values were computed over the period 1971–2000 for BESM-OA2.5 and 20CRv2. The concentrations are presented as percentages."